# Learning to Strategically Acquire Resources in Competition

**Safwan Hossain**[*]
*Harvard University*

**Mirah Shi**[*]
*University of Pennsylvania*

**Andrew Bennett**
*Morgan Stanley*

**Neil Andrew Chriss**
*Fixed Point Advisors*

**Michael Kearns**
*University of Pennsylvania*

**Anderson Schneider**
*Morgan Stanley*

**Yuriy Nevmyvaka**
*Morgan Stanley*

**Reviewed on OpenReview:** `https://openreview.net/forum?id=Z8j6jGVxAZ`

## Abstract

We consider multiple agents competing to acquire some costly divisible resource (*e.g.* shares of a financial asset, compute resources, etc.) over time. Leveraging a standard model for price dynamics, we propose a novel game-theoretic model for this problem, generalizing settings studied in diverse literatures. Our analysis considers different assumptions on the information available to agents. Under partial-information with a common prior (which subsumes complete information as a special case), we establish the existence, uniqueness, and efficient computability of the Bayesian Nash equilibrium (BNE), and bound the price of anarchy. Next and more generally, we consider agents with no common prior learning to act optimally given realistic market feedback from repeated interactions. We provide sufficient conditions on agents doing simultaneous learning dynamics for last-iterate convergence to the BNE. For all settings, we provide simulations based on real financial data to illustrate our theoretical results and offer new insights on strategic behavior in the context of trading and resource acquisition.

## 1 Introduction

Consider multiple traders trying to acquire a position in a stock ahead of an earnings release. If each trader were acting in isolation, they might follow a classical optimal execution strategy – such as that of Almgren & Chriss (2000) – to minimize their effect on price and thus the cost of trading. However, if multiple traders are pursuing their strategies simultaneously, their aggregate activity influences prices and liquidity. This interaction transforms the problem from individual optimization to understanding the inter-agent strategic behavior, where each agent's decisions affect the market environment faced by all. This challenge of acquiring costly resources in competitive, dynamically priced environments extends beyond financial markets. Training an LLM may require securing substantial cloud computing resources within a given time. As spot prices are shaped by aggregate demand across many users (Shastri & Irwin, 2018), firms need to account not only for their own scheduling and budget constraints but also how others interact. Agents in such environments may only have incomplete knowledge of others or the market itself, complicating their strategy.

While several works have attempted to formally capture these strategic perspectives, they do so with several limitations. Chriss (2024a;b;c) consider a complete-information setting, which is generally unrealistic. More recently, Chriss (2025); Kearns & Shi (2025) study extensions that either assume common priors or repeatedly

---

[*]Equal contributors. Safwan Hossain and Mirah Shi interned at Morgan Stanley Machine Learning Research during the course of the project. Correspondence to shossain@g.harvard.edu, mirahshi@seas.upenn.edu, or Andrew.Bennett@morganstanley.com

Table 1: Comparison of our framework with prior work on strategic position building / resource acquisition.

|  | Constraints | Objective | Information | Equilibrium Result |
|---|---|---|---|---|
| Chriss (2024b;a) | Fixed target | Min. acquisition cost | Complete | Exact NE computation |
| Chriss (2024c) | Fixed target + monotonicity, rate bounds | Min. acquisition cost | Complete | Approx. NE computation |
| Chriss (2025) | Fixed target | Min. acquisition cost | Incomplete (common prior) | Exact BNE computation |
| Kearns & Shi (2025) | Fixed target + per-period trade bounds | Min. acquisition cost | Complete | Approx. CCE computation (via simulated no-regret play) |
| **This work** | General convex | Max. general concave utility minus acquisition cost | Incomplete (no prior required) | Convergence to BNE via interaction |

interacting with the same market instance, a rare occurrence. Further, all these works: (1) require agents to acquire a fixed target position, ruling out more general action constraints; (2) do not allow for objectives that agents may wish to optimize alongside acquisition costs; and most importantly (3) do not address the computational and learning challenges that arise when agents act under incomplete information without common priors. The broad scope and practical relevance of this problem necessitates a general framework that can gracefully accommodate diverse constraints and limitations of real-world markets.

As such, this work proposes a general competitive resource acquisition and trading framework for dynamically priced markets. A central feature of our model is that prices respond to the aggregate actions of all participants. Beyond this, our game-theoretic model makes minimal modeling assumptions. Information may be imperfect or asymmetric, and agents need not know the information structure a priori. Moreover, we allow for rich heterogeneity in agents' objectives: agents may target some fixed position or seek to maximize a personalized utility function on their final position, with different agents having different goals. Finally, our framework accommodates action constraints that may be important for practical applications, such as no short selling or granular limits on per-period trades. Together, these accommodations provide a flexible game-theoretic model that captures a broad range of goals, constraints, and information structures relevant to strategic resource acquisition problems.

## 1.1 Our Contribution

- Section 2 formalizes a novel model for this problem, which generalizes on past settings, by allowing for convex constraints, concave value functions, and incomplete information (see Table 1).[1]

- Section 3 considers the partial-information Bayesian setting: each agent only observes their own private information (their "type"), but all agents have common knowledge of the prior distribution over agent types and market parameters. We establish the uniqueness and computability of the Bayesian Nash Equilibrium (BNE), and characterize the price of anarchy of this game.

- Section 4 extends our model to an online learning setting where agents only observe their type and have no knowledge of the distribution over others' types or market parameters. They must instead learn from repeated interactions and choose their strategy conditioned on their realized type and past feedback. In particular, we consider a feedback model that is natural for this setting: instead of perfectly observing the market parameters or others' actions as end-of-round feedback, agents see only the realized price trajectory and can evaluate their counterfactual costs using estimates of the market parameter. This models learning to acquire resources given realistic contextual information. We establish sufficient conditions under which agents simultaneous learning converge to the BNE in the final round.

---

[1]Kearns & Shi (2025) could alternatively be framed as a partial information model with convergence to CCE via interaction. However, unlike our model for interaction, theirs assumes an identical game in every interaction (same target positions, constraints, market parameters, *etc.*), and that players have sufficient information to compute exact best responses.

- In Section 5, we empirically simulate our algorithms based on real market data where possible and offer new insights for both the with-prior and prior-free settings.

## 1.2 Additional Related Work

Most relevant to our paper is the line of work on optimal position building in financial markets that we discussed above (Chriss, 2024b;c;a; 2025; Kearns & Shi, 2025). Our work can be seen as a generalization of these, accommodating constraints and objectives natural in both financial and non-financial settings such as compute markets Shastri & Irwin (2018), which are of growing importance. We provide a detailed discussion of how our model relates to and subsumes the settings of these works in Appendix C.2. A summary comparison is given in Table 1.

More broadly, our model captures standard notions of *market impact* in finance, of which there is a large literature (see Webster (2023); Li et al. (2024) and citations therein for a detailed overview). These works broadly consider how prices change in response to trading (both theoretically and empirically from real markets). In this literature, market impact is often decomposed into permanent and temporary impact (Almgren & Chriss, 2000; Bacry et al., 2015; Moro et al., 2009), which is the same approach that we take. There are also models such as the *propagator model* (Bouchaud et al., 2004; Gatheral & Schied, 2013; Obizhaeva & Wang, 2013) that allow for transient impacts in between these extremes; we do not consider these, but such extensions would be an interesting direction for future work.

Our work relates to the literature on learning in games more broadly. Particularly, the setting in Section 4 is similar in spirit to that of Hartline et al. (2015), who consider a framework for no-regret learning in repeated Bayesian games. That said, our model is outside their framework (as it allows continuous market types), and has some specific structure that we can additionally leverage (strong monotonicity).

Our setup is conceptually related to resource allocation/congestion games, where agents share resources, and the cost of any resource depends on its demand (Rosenthal, 1973). Unlike canonical versions of these, however, we consider time-varying prices that endogenously adjust based on supply and demand. Lastly, while we study competitive resource acquisition in a market setting, a non-market variant has long been studied in the context of *fair division* (Moulin, 2004) for both divisible and indivisible goods. Recent literature here has extended this problem to an online setting (see Aleksandrov & Walsh (2020) for a survey), and often incorporates learning and predictions (Banerjee et al., 2023), spiritually motivating our model in Section 4.

## 2 Model

**Preliminaries** Consider a market of $n$ strategic agents looking to trade (buy/sell) some costly, divisible resource (stock, currency, compute time, *etc.*) over a period of $T$ rounds. In the simplest setting, each strategic agent $i$'s action (also called strategy or trajectory) is a $T$-dimensional vector $\boldsymbol{h}_i$, where $h_{i,t}$ denotes how much they purchased at time $t \in [T]$. Note that $\boldsymbol{h}_i$ is a signed vector, and we conventionally denote positive (negative) values as buying (selling) throughout. Each agent has some set of convex constraints on their allowable actions, represented by a convex feasible set of trajectories $G_i \subseteq \mathbb{R}^T$. For example, if agent $i$ wants to procure at most $V_i$ shares without short selling or over-buying, their constraints are given by $G_i = \{\boldsymbol{h}_i : 0 \leq \sum_{l=1}^{t} h_{i,l} \leq V_i \ \forall t \in [T]\}$. We also assume that each agent has some value function over their strategy, which can capture the value of their final position and/or any preferences on their acquisition schedule. For an agent $i$, we represent this via a bounded concave function[2] $f_i : \mathbb{R}^T \to \mathbb{R}$, with $|f_i(\cdot)| \leq U$. For example, if agent $i$ has a concave value $\phi_i$ on their final acquired position (say they have a private valuation $r_i$ for the asset, so they value owning $x$ shares as $\phi_i(x) = r_i x$) and penalizes selling by some non-negative factor $\zeta$, this function could be $f_i(\boldsymbol{h}_i) = \phi_i(\mathbf{1}^\top \boldsymbol{h}_i) - \zeta \sum_{t=1}^{T} |h_{i,t}| \mathbb{I}\{h_{i,t} < 0\}$, which is clearly concave. In settings like compute markets or optimal trade execution, where the agents' goal is to acquire a fixed target position as cheaply as possible, one can set $f_i = 0$ and include a hard constraint on $\mathbf{1}^\top \boldsymbol{h}_i$ in

---

[2]Concave utility on final position, which corresponds to diminishing marginal utility, is a natural restriction in economics and game theory. See (Mas-Colell et al., 1995; Debreu, 1959).

$G_i$. Lastly, and inspired by the seminal work of Kyle (1985), we allow the market to contain a non-strategic (possibly random) *exogenous* agent, which captures all non-strategic trade flow. The exogenous action is given by a signed vector $\boldsymbol{s} \in \mathbb{R}^T$, where positive values indicate buying.

**Price Model**  Core to understanding how agents interact strategically is modeling how the aggregate demand/supply levels influence resource prices. We assume the following dynamic model for price as affected by agents' trading:

**Assumption 1** (Price Dynamics). *All agents face the same price $p_t$ for each share of the resource traded at time $t$. The price $p_t$ is determined by the total trading trajectories of all agents up to and including time $t$ according to the following equations:*

$$p_t = p_t^w + \beta D_t; \qquad p_t^w = p_{t-1}^w + \alpha D_t$$

*where $p_0 = p_0^w$ is the initial price, $D_t = \sum_{i=1}^n h_{i,t} + s_t$ is the aggregate excess demand at time $t$, and $\alpha > 0$ and $\beta \geq 0$ are some market parameters.*

The dynamic process for $p_t^w$ is a discretization of the Walrasian price dynamics from general equilibrium theory, which posits that prices evolve from an imbalance of supply and demand: $dp_t = \alpha(\text{demand}_t - \text{supply}_t)dt$, where $\alpha > 0$ is a sensitivity factor (Walker, 1987). The additional $\beta\left(\sum_{i=1}^n h_{i,t} + s_t\right)$ term in the execution price $p_t$ accounts for additional costs imposed by market makers who provide liquidity to balance supply and demand, causing temporary deviations from the equilibrium price ($\beta \geq 0$ controls the strengths of this impact). This maps to how prices are modeled within the theory of optimal trade execution, with $\alpha$ and $\beta$ corresponding to permanent and temporary impact coefficients, respectively (Almgren & Chriss, 2000) – see Appendix C.1 for more details. That said, this remains a stylized model that abstracts some of the deeper micro-structures of limit order books – see Section 6 for a discussion on this.

**Market Type**  We refer to the total set of parameters defining the market and pricing dynamics as the *Market Type*, which we denote as the tuple $\boldsymbol{\lambda} = (f_1, \ldots, f_n, p_0, \alpha, \beta, \boldsymbol{s})$.

**Agent Types**  We consider a very general partial information model, where each agent's information is determined by their *type*. As discussed below, this framework subsumes complete information settings, so we do not study those separately. Agent types are formalized as follows.

**Definition 1** (Agent Types). *For each agent $i$, let $\theta_i \in \Theta_i$ denote their* type*, capturing all information known to them before acting. More specifically:*

- *$\Theta_i$ is some compact type space which can be either discrete ($|\Theta_i| = m_i < \infty$) or continuous ($\Theta_i \subseteq \mathbb{R}^{m_i}$).*
- *Player type $\theta_i$ fully determines their constraints, denoted $G_i(\theta_i) \subseteq \mathbb{R}^T$, which we assume to be non-empty, closed, convex, and bounded.*

While we make no assumptions about the types themselves, in practice they can be perceived as the features agents use to understand the market. Importantly, we make no assumptions about how well the private type $\theta_i$ correlates with the market parameters $\boldsymbol{\lambda}$; they may range anywhere from fully informative to completely uninformative, or anywhere in between. More generally, the correlation between agent types and $\boldsymbol{\lambda}$ may be of varying strengths for different agents, naturally capturing information asymmetry that is common in most markets. We also *do not* assume that $\theta_i$ fully specifies the value function $f_i$: instead $f_i$ can depend on uncertain market parameters. Finally, we note that if $\theta_i$ is in fact fully informative of $\boldsymbol{\lambda}$ for every agent, then our model is equivalent to the canonical "complete information" setting.

**Realized Utility**  For fixed market type and trading trajectories, we model the total utility realized by each agent $i$ according to their personal value $f_i$, minus the total cost they incur buying and selling. Formally:

**Definition 2** (Realized Utility). *Let $(\boldsymbol{h}_i, \boldsymbol{h}_{-i})$ denote the trading trajectories of agent $i$ and all other strategic agents respectively. Letting $\boldsymbol{p}(\boldsymbol{h}_i, \boldsymbol{h}_{-i}, \boldsymbol{\lambda}) \in \mathbb{R}^T$ denote the sequence of prices under Assumption 1, agent $i$'s realized utility is:*

$$u_i(\boldsymbol{h}_i; \boldsymbol{h}_{-i}, \boldsymbol{\lambda}) = f_i(\boldsymbol{h}_i) - \boldsymbol{p}(\boldsymbol{h}_i, \boldsymbol{h}_{-i}, \boldsymbol{\lambda})^\top \boldsymbol{h}_i \,.$$

**Regular Strategy Spaces**  Under partial information, we formalize the strategy for each agent as a function $\boldsymbol{h}_i : \Theta_i \to \mathbb{R}^T$, which determines how they would behave under any type realization. For any feasible strategy $\boldsymbol{h}_i$, it must be that $\boldsymbol{h}_i(\theta_i) \in G_i(\theta_i)$ for every $\theta_i \in \Theta_i$. In practice, however, agents may benefit from considering more restricted spaces, especially when type spaces $\Theta_i$ become more complex and we wish to ensure learnability. We therefore consider a more general class of feasible strategy spaces with explicit complexity bounds:

**Definition 3** (Strategy Spaces). *We say that a strategy space $\mathcal{H}_i$ for agent $i$ is $(B, L)$-regular if:*

1. *The strategy space $\mathcal{H}_i$ is feasible: $\boldsymbol{h}_i(\theta_i) \in G_i(\theta_i)$ for all $\theta_i$ in $\Theta_i$, and $\boldsymbol{h}_i \in \mathcal{H}_i$.*
2. *The strategy space $\mathcal{H}_i$ is convex and is a closed subset of the $L_2(\Theta_i, P_{\theta_i}; \mathbb{R}^T)$ function space where $\langle \boldsymbol{h}_i, \boldsymbol{h}_i' \rangle_{\mathcal{H}_i^*} = \mathbb{E}_{\theta_i}[\langle \boldsymbol{h}_i(\theta_i), \boldsymbol{h}_i'(\theta_i) \rangle]$.*
3. *Every strategy $\boldsymbol{h}_i \in \mathcal{H}_i$ is bounded: $\|\boldsymbol{h}_i(\theta_i)\|_2 \leq B$ for all $\theta_i \in \Theta_i$*
4. *Every strategy $\boldsymbol{h}_i \in \mathcal{H}_i$ is Lipschitz: $\|\boldsymbol{h}_i(\theta_i) - \boldsymbol{h}_i(\theta_i')\|_\infty \leq L\|\theta_i - \theta_i'\|_\infty$ for all $\theta_i, \theta_i' \in \Theta_i$.*

**Bayesian Game Instance**  Putting the above together, we can fully characterize an overall *Bayesian Game* instance in terms of a distribution over possible instances, feasible strategies as mappings from player type to trading trajectory, and expected agent utilities, as follows:

**Definition 4** (Bayesian Game Instance). *An instance of our Bayesian game setting is characterized by:*

1. ***Distribution over instances:*** *Let $\mathcal{I} = (\theta_1, \ldots, \theta_n, \boldsymbol{\lambda})$ denote a game instance. We assume a joint distribution $P(\theta_1, \ldots, \theta_n, \boldsymbol{\lambda})$ from which the components of an instance $\mathcal{I}$ are sampled from.*
2. ***Strategy Spaces:*** *Each agent $i$ uses a $(B_i, L_i)$-regular strategy space $\mathcal{H}_i$, for some finite non-negative $B_i$, and some (possibly infinite) non-negative $L_i$.*
3. ***Game Payoff Structure:*** *Each agent $i$'s expected utility given their and others' strategies $(\boldsymbol{h}_i, \boldsymbol{h}_{-i})$ is the expected value of their realized utility over instances $\mathcal{I} \sim P$: $\mathbb{E}_{\theta_i, \theta_{-i}, \boldsymbol{\lambda} \sim P}[u_i(\boldsymbol{h}_i(\theta_i); \boldsymbol{h}_{-i}(\theta_{-i}), \boldsymbol{\lambda})]$.*

Since player strategies are *ex-ante* mappings from their possible types to feasible trading schedules, game play can be ordered as follows: (1) each agent *ex-ante* decides their strategy function $\boldsymbol{h}_i$, (2) a game instance $\mathcal{I} \sim P$ is sampled and the corresponding type information $\theta_i$ is privately revealed to each agent, and (3) each agent executes their trading schedule $\boldsymbol{h}_i(\theta_i) \in G_i(\theta_i)$ and attains the realized utility.

Finally, we comment on what information, if any, agents have about the prior distribution $P$ over the game instances. In Section 3, we consider the case where all agents have common knowledge of $P$, whereas in Section 4 we consider agents having no knowledge of $P$ and must learn via interaction. Although learning from interaction is more realistic, the idealized "common prior" setting allows us to define and analyze concrete notions of equilibria, which we then show in Section 4 can be approximately reached after sufficient rounds of interaction.

## 3  Agents with Common Prior

In this section, we assume that all strategic agents have common knowledge of the joint distribution $P$ over agent and market types. Our focus here is on the characterization of equilibria, in terms of their existence, uniqueness, quality, and computability. This setting can be formally studied in the Bayesian game theory framework, using the standard equilibrium notion:

**Definition 5** (Bayesian Nash Equilibrium). *The strategies $(\boldsymbol{h}_1^{eq}, \ldots, \boldsymbol{h}_n^{eq})$ are in a Bayesian Nash Equilibrium (BNE) if for all agents $i$, all $\boldsymbol{h}_i' \in \mathcal{H}_i$, and all $\theta_i \in \Theta_i$, we have: $\mathbb{E}_{\boldsymbol{\theta}, \boldsymbol{\lambda} \sim P}[u_i(\boldsymbol{h}_i^{eq}(\theta_i); \boldsymbol{h}_{-i}^{eq}(\theta_{-i}), \boldsymbol{\lambda})] \geq \mathbb{E}_{\boldsymbol{\theta}, \boldsymbol{\lambda} \sim P}[u_i(\boldsymbol{h}_i'(\theta_i); \boldsymbol{h}_{-i}^{eq}(\theta_{-i}), \boldsymbol{\lambda})] - \varepsilon$ and $\varepsilon = 0$. When $\varepsilon > 0$, we denote the strategies as being in $\varepsilon$-BNE.*

BNE are the set of joint strategies such that no agent has any *ex-ante* incentive to unilaterally deviate, given others' strategies. Note that if all agent types are perfectly informative of $\boldsymbol{\lambda}$ (i.e. all agents have complete information), then the BNE coincides with the classical notion of Nash Equilibrium.[3]

---

[3]Specifically, under complete information, agents are in BNE if and only if for every possible $\mathcal{I}$ their trajectories given $\mathcal{I}$ are a Nash Equilibrium for the fixed game instance defined by $\mathcal{I}$.

In any case, our definitions and statements so far have been framed with respect to pure (deterministic) strategies. In general, agents may use mixed (randomized) strategies, which begs the question: could mixed strategies appear in BNE? The following lemma immediately establishes that this cannot be the case. We give brief proof sketches of this and other results in this section, with the full details deferred to Appendix A.

**Lemma 1.** *The best response of any agent $i$ is always unique and deterministic – $\boldsymbol{h}_i(\theta_i)$ is unique and deterministic for every type $\theta_i$ – even if others are playing some mixed (possibly correlated) set of strategies.*

This result relies on unrolling the auto-regressive price definition and observing that each agent's utility can be expressed as a strictly concave quadratic function of the joint strategies $h_1(\theta_i), \ldots, h_n(\theta_n)$.

**Existence and Uniqueness of Equilibrium**   We now characterize the existence and uniqueness of the BNE for this game. The following theorem establishes that BNE always exist and are unique.

**Theorem 1.** *Every instance of the Bayesian game (Definition 4) has a unique and deterministic BNE.*

At a high level, the proof of Theorem 1 works by casting the BNE conditions as a variational inequality in the function space $L_2(P)$ – the space of square-integrable functions from $\theta_i$ to $\mathbb{R}^T$ with an inner product written in terms of expectation over $P$. The latter point is key since it means the first order optimality conditions characterizing the BNE can be expressed as a variational inequality in $L_2(P)$. This variational inequality operator can be shown to be strongly monotone, with strong monotonicity constant depending on the distributions of $\alpha$ and $\beta$. This, in combination with the $(B_i, L_i)$-regularity conditions on the strategy spaces $\mathcal{H}_i$, are sufficient to ensure a unique, deterministic BNE.

**Computability of Equilibria**   Treating the BNE computation as an optimization problem, we first note complexity results here are typically stated in terms of iterations or queries to some oracle. Since the BNE conditions can be cast as a strongly monotone variational inequality, by additionally proving Lipschitzness, we can invoke the extra-gradient algorithm to compute this (Korpelevich, 1976). Specifically, assuming oracle access to this operator, we can find a $\varepsilon > 0$ approximate BNE – in the sense that our solution $\hat{x}$ satisfies $||\hat{x} - x^*|| \leq \varepsilon$ in $O\left(\frac{L}{c} \log \frac{1}{\varepsilon}\right)$ queries. We detail this in Corollary 1 in Appendix A. In the case of discrete type-spaces, a more granular bound can be given in terms of calls to the gradient of $f_i$.

**Quality of Equilibria**   Finally, we turn to the characterization of BNE quality. For this, we use the popular *Price of Anarchy* (PoA) notion of Roughgarden (2010), which considers the ratio between maximum possible welfare (sum of all agent utilities) and the welfare at an equilibrium. Since BNE are deterministic and unique, we can give the following simplified formal definition of the PoA:

**Definition 6.** *Let $\boldsymbol{h}_1^{eq}, \ldots, \boldsymbol{h}_n^{eq}$ denote the unique BNE strategies given distribution $P$, and let $welf(\boldsymbol{h}_1, \ldots, \boldsymbol{h}_n, \boldsymbol{\lambda}, \boldsymbol{\theta}) = \sum_{i=1}^n u_i(\boldsymbol{h}_i(\theta_i); \boldsymbol{h}_{-i}(\theta_{-i}), \boldsymbol{\lambda})$ denote the total welfare function. The* Price of Anarchy *ratio is then defined as:*

$$\frac{\sup_{\boldsymbol{h}_1, \ldots, \boldsymbol{h}_n \in \mathcal{H}_1 \ldots \mathcal{H}_n} \mathbb{E}_{\boldsymbol{\theta}, \boldsymbol{\lambda}}[welf(\boldsymbol{h}_1, \ldots, \boldsymbol{h}_n, \boldsymbol{\lambda}, \boldsymbol{\theta})]}{\mathbb{E}_{\boldsymbol{\theta}, \boldsymbol{\lambda}} welf(\boldsymbol{h}_1^{eq}, \ldots, \boldsymbol{h}_n^{eq}, \boldsymbol{\lambda}, \boldsymbol{\theta})}$$

Since the welfare is always non-negative in any non-degenerate case[4] the quantity above is always $\geq 1$, with higher values indicating equilibrium welfare (multiplicatively) far from the optimal. We first consider whether it is possible to bound the PoA for any instance of the Bayesian game. The following theorem shows that this is impossible, as the PoA can be unbounded in some cases.

**Theorem 2.** *For any constants $\alpha > 0, \beta > 0, T \geq 2$, and any $\xi > 0$, there exists a complete information instance where the PoA ratio is at least $\xi$.*

The proof of Theorem 2 is based on an explicit construction where with two traders who strategically trade in opposite directions — one buying and another selling and essentially providing liquidity to the first.

---

[4]if $f_i(\boldsymbol{0}) = 0$, buying/selling nothing guarantees 0 utility to any agent and thus dominates negative utility strategies

As this construction relies on strategic agents trading on both sides, it is natural to ask whether PoA is bounded for game instances where strategic agents all want to trade in the same direction? Such scenarios commonly occur in markets. For example, after an earnings call with earnings above expectations, traders may systematically move their positions in a positive direction. As another example, it may be reasonable to expect that agents in compute markets are all trying to cheaply acquire compute resources. We next establish such a bound, with the proof based on the *smooth games* framework of Roughgarden (2015).

**Theorem 3.** *For any Bayesian game instance $P$ with the following properties:*

- *Agents are constrained to only buy: $h_{i,t}(\theta_i) \geq 0, \forall i, \theta_i, t$ and $s_t \geq 0$.*
- *Each agent $i$'s goal is to build a position $V_i(\theta_i)$ at minimum cost. That is, for all $i$, $\theta_i$, $\sum_t h_{i,t}(\theta_i) = V_i(\theta_i)$ is a constraint and $f_i = 0$.*

*Then for $\gamma = \sup_{\alpha, \beta \in supp(P)} \frac{\alpha}{\alpha + 2\beta}$, the Price of Anarchy is upper bounded by $O(n^2 T^2 \gamma^2)$.*

## 4 Learning to Play without a Common Prior

We now move to a more general and practically motivated setting, where players do not have prior common knowledge of $P$ and instead must learn to act via repeated interaction with the market. Our focus here is to give learning algorithms for agents that ensure convergence to the unique BNE characterized earlier in Section 3.

**Learning Problem Setup:** For this learning setting, all agents interact over $R \in \mathbb{N}^+$ rounds. In each round $r \in [R]$, the interaction follows the same sequence of events as in the Bayesian game outlined in Section 3 — each agent first decides their strategy $\boldsymbol{h}_i^r$, then an instance $\mathcal{I}$ is sampled from $P$ and each agent observes their type $\theta_i$ and correspondingly plays strategy $\boldsymbol{h}_i^r(\theta_i)$. Each agent $i$'s strategy at round $r$, $\boldsymbol{h}_i^r$, can be chosen adaptively based on feedback observed in all previous rounds.

We adopt a realistic feedback model, where at the end of every round, each agent observes the price trajectory of that round — typically public information in markets — and estimates of the market impact coefficients $\alpha^r$ and $\beta^r$ for that round. This can then be used to estimate their cost function. The end-of-round feedback is formalized as follows:

**Assumption 2** (End-of-Round Feedback). *At the end of every round $r$, each agent $i$ observes the realized price trajectory $\{p_t^r\}_{t=0}^T$ and receives estimates $\hat{\alpha}_i^r, \hat{\beta}_i^r$ of market coefficients $\alpha^r, \beta^r$ satisfying $\mathbb{E}[|\hat{\alpha}_i^r - \alpha^r|] \leq \Delta_\alpha$ and $\mathbb{E}[|\hat{\beta}_i^r - \beta^r|] \leq \Delta_\beta$.*

In practice, estimating market parameters can be done via standard regressions — or, more generally, any black-box predictive model — trained on observed market data. We provide one concrete procedure for constructing these estimates in Section 5. Our analysis in this section treats these estimates as outputs of some abstract oracle, and the resulting guarantees depend only on estimation error.

**Doubly-Optimal Learning Task:** A good learning algorithm should guarantee desirable convergence behavior without prior knowledge of $P$. Standard no-regret approaches can ensure that the empirical distribution of actions across rounds converges to a coarse-correlated equilibrium (see Hartline et al. (2015) for the classical result of this kind for Bayesian games). This, however, can be unsatisfying in practice since learners would typically prefer convergence of their actual strategy toward some equilibrium, with the best case being last-iterate convergence to BNE.

Recently, there has been work on *doubly-optimal* algorithms for game-playing that guarantee last-iterate convergence to Nash equilibrium when applied simultaneously by all agents, while ensuring no-regret when other agents behave arbitrarily, which is a useful defensive guarantee (Jordan et al., 2024). Importantly, such algorithms do not require communication between agents, with each agent's action determined by the

feedback from their own cost function. Ideally, we would like to devise a learning algorithm that can obtain such doubly-optimal properties. The existing analysis on doubly-optimal learning does not apply to our setting for two key reasons:

1. Unlike Jordan et al. (2024), we consider Bayesian game instances, where strategies are mappings from types to trajectories. Since types may be continuous, strategies are not generally representable as finite-dimensional vectors, which their results rely upon.

2. Their algorithm and analysis require unbiased stochastic cost function gradients. On the one hand, the realized agent costs for the sampled instance $\mathcal{I} \sim P$ at each round $r$ is an unbiased estimate of the expected cost of that strategy over $P$. However, computing the gradients of these realized costs requires perfect knowledge of $\alpha^r$, $\beta^r$, and aggregate demand $\sum_{j \neq i} \boldsymbol{h}_j + \boldsymbol{s}$. Under Assumption 2, the gradients available to agents can only be computed from estimated market-impact parameters and reconstructed aggregate demand, introducing bias in gradient estimates.

These two limitations create new technical obstacles, which our algorithm and theory address. For the former, our approach is to discretize the type space. For the latter, our approach is to compute stochastic gradient estimates and extend the theory of Jordan et al. (2024) to handle bias in such estimates. We next discuss these two extensions, before presenting our main algorithm and convergence results.

**Type Space Discretization:** We handle continuous type spaces via discretization. Specifically, given a continuous type space $\Theta_i$ and hyperparameter $\delta$, we compute a $\delta$-net $\hat{\Theta}_i$ of $\Theta_i$: for every $\theta_i \in \Theta_i$, there exists a $\hat{\theta}_i \in \hat{\Theta}_i$ such that $||\theta_i - \hat{\theta}_i|| \leq \delta$. We then restrict ourselves to strategies $\hat{\boldsymbol{h}}_i \in \hat{\mathcal{H}}_i \subseteq \mathcal{H}_i$ that are piece-wise constant over the "bins" (the set of $\theta_i \in \Theta_i$ that are closest to a specific $\hat{\theta}_i \in \hat{\Theta}_i$) introduced by the nearest neighbour mapping between $\Theta_i$ and $\hat{\Theta}_i$. Finally, we give a "lifting" argument using Lipschitzness of our strategies to show that convergence of learning under $\hat{\mathcal{H}}_i$ ensures approximate convergence under $\mathcal{H}_i$.

---

**Algorithm 1:** Learning Under Estimated Market Parameters

---

Fix $\delta = 1/\log R$. Let $\hat{\Theta}_i$ be a $\delta$-net of $\Theta_i$. For each $\theta_i \in \Theta_i$, let $\hat{\theta}_i$ denote its nearest neighbor in $\hat{\Theta}_i$

Let $\hat{\mathcal{H}}_i = \{\hat{\boldsymbol{h}}_i : \Theta_i \to \mathbb{R}^T | \hat{\boldsymbol{h}}_i(\theta_i) = \boldsymbol{h}_i(\hat{\theta}_i) \, \forall \boldsymbol{h}_i \in \mathcal{H}_i\}$ be the strategy space that is piecewise constant over each bin given by the discretization

Initialize $\hat{\boldsymbol{h}}_i^1 \in \hat{\mathcal{H}}_i$

Let $z_0 = \frac{1}{\log(R+10)}$

**for** $r = 1, ..., R$ **do**

  Upon observing $\theta_i^r$, compute $\hat{\theta}_i^r$

  Sample $M^r \sim \text{Geometric}(z_0)$

  Let $\eta^{r+1} = \frac{r+1}{\sqrt{1 + \max\{M^1, ..., M^r\}}}$

  Update $\boldsymbol{h}_i^{r+1} = \arg\min_{\hat{\boldsymbol{h}}_i \in \hat{\mathcal{H}}_i} \{(\hat{\boldsymbol{h}}_i - \boldsymbol{h}_i^r)^\top \tilde{\nabla}_i^r + \frac{\eta^{r+1}}{2} \|\hat{\boldsymbol{h}}_i - \boldsymbol{h}_i^r\|^2\}$, where $\tilde{\nabla}_i^r$ is the estimated gradient

---

**Estimated Stochastic Cost Gradient:** In our setting, agents do not observe the realized aggregate demand or the true market-impact parameters; instead, they observe prices and estimate these parameters. We therefore derive a recursive reconstruction of aggregate demand from the price trajectory and the estimated parameters, and use this reconstruction to compute an estimated stochastic gradient. The resulting gradient is biased, with bias controlled by the market-parameter estimation errors. To handle this, we give a new analysis of the algorithm that tolerates some amount of bounded bias, leading to convergence rates that degrade with the estimation error.

Formally, for each agent $i$, let $D_{i,t}^r = \sum_{i=1}^n h_{i,t}^r + s_t^r$ denote the (unobserved) aggregate demand at time $t \in [T]$ of round $r$. Given Assumption 1 it can be shown that $D_{i,t}^r$ can be recursively defined in terms of price increments and $\alpha^r, \beta^r$. Replacing $\alpha^r, \beta^r$ with estimates $\hat{\alpha}_i^r, \hat{\beta}_i^r$ then gives the following recursive estimate for $D_{i,t}^r$:

$$\hat{D}_{i,t}^r = \frac{(p_t^r - p_{t-1}^r) + \hat{\beta}_i^r \hat{D}_{i,t-1}^r}{\hat{\alpha}_i^r + \hat{\beta}_i^r}, \quad \hat{D}_{i,0}^r = 0$$

(See the proof of Theorem 4 for the derivation.) Given this, each agent $i$ can then form an estimate of the aggregate *external* demand schedule: $\hat{d}^r_{-i} = \hat{D}^r_i - \boldsymbol{h}^r_i(\theta^r_i)$, where $\hat{d}^r_{-i}, \hat{D}^r_i \in \mathbb{R}^T$. Then, plugging this in along with the estimates $\hat{\alpha}^r_i$ and $\hat{\beta}^r_i$, each agent $i$ can compute an estimated stochastic gradient of their expected cost at round $r$:

$$\tilde{\nabla}^r_{\boldsymbol{h}_i} = p_0 \boldsymbol{1}_T + \hat{\alpha}^r_i W \boldsymbol{h}^r_i(\theta^r_i) + \hat{\alpha}^r_i W' \hat{d}^r_{-i} + \hat{\beta}^r_i \big(2\boldsymbol{h}_i(\theta_i) + \hat{d}^r_{-i}\big) - \nabla f_i\big(\boldsymbol{h}_i(\theta_i)\big),$$

where $W \in \mathbb{R}^{T \times T}$ is the matrix with $W_{tt} = 2$ for all $t \in [T]$ and 1 everywhere else, and $W' \in \mathbb{R}^{T \times T}$ is the matrix with $W'_{ts} = 1$ for $s \leq t$ and 0 everywhere else.

**Main Algorithm and Learning Result:** Given these ingredients, we present our learning procedure in Algorithm 1, which achieves the following $\varepsilon$-BNE guarantee – i.e. no agent can unilaterally deviate to improve their expected utility by more than $\varepsilon$:

**Theorem 4.** *Let all agents use Algorithm 1 to select strategies. Then, the last iterate strategies $(\boldsymbol{h}^R_1, ..., \boldsymbol{h}^R_n)$ are an $\epsilon$-approximate BNE, for an $\epsilon$ that satisfies, in expectation over the algorithm's randomness:*

$$E[\epsilon] = O\left(\text{poly}(n, T, p_0, B, S, U', \alpha_{\max}, \beta_{\max}, \tfrac{1}{\kappa\hat{\kappa}})\left(\frac{L}{\log R} + \frac{\log^{(m+3)/2}(R)}{\sqrt{R}} + (\Delta_\alpha + \Delta_\beta)\left(1 - \frac{1}{R}\right)\log^2(R)\right)\right)$$

*where $m = \max_i m_i$, $L = \max_i L_i$, and $S$, $U'$, $\kappa$, $\hat{\kappa}$, $\alpha_{\max}$, and $\beta_{\max}$ are constants such that $\sup_{\boldsymbol{s}} \|\boldsymbol{s}\| \leq S$, $\|\nabla f_i(\boldsymbol{h}_i(\theta_i))\| \leq U'$, $\kappa \leq \hat{\alpha}^r_i \leq \alpha_{\max}$, and $\hat{\kappa} \leq \hat{\beta}^r_i \leq \beta_{\max}$, which all hold uniformly over $i$, $r$, and $\theta_i$.*

The proof sketch is as follows. We start by discretizing the strategy space using a $\delta$-net (if they are already not discrete). Next, we extend the key convergence lemma of Jordan et al. (2024) to accommodate gradients biased by some $\Delta$. Then, given a novel formulation of Bayesian games in their framework and the regularity properties of $\hat{\mathcal{H}}_i$, we establish approximate convergence to BNE with respect to the discretized strategy spaces $\hat{\mathcal{H}}_i$, with error dependent on $\Delta_\alpha + \Delta_\beta$. Finally, we extend this to convergence to BNE with respect to the *true* strategy spaces $\mathcal{H}_i$ with an additional $L$-dependent error by leveraging $(B, L)$ regularity. See Appendix B for full proof details.

Estimation errors introduce a term in the equilibrium approximation $\varepsilon$ that grows poly-logarithmically in $R$; while this term is not sublinear, we argue that it does not preclude the guarantee from being meaningful in practice. For fixed horizon $R$, if the estimation error is sufficiently small relative to $R$ – for instance, if the estimates improve over time as more market data is observed — then the overall error vanishes. Lastly and as desired, when $\Delta_\alpha = \Delta_\beta = 0$ (i.e agents have *ex-post* access to true market parameters), our result guarantees convergence to the BNE with vanishing error.

Finally, we note that while Theorem 4 focuses on last-iterate convergence to BNE, a nearly identical analysis also establishes the other doubly-optimal property, *i.e.* that algorithm is approximately no-regret even if other agents behave arbitrarily (and potentially adversarially), which we omit for brevity.

**Practical implementation of Algorithm 1:** We briefly comment on the optimization step in the per-round update of Algorithm 1. For each agent $i$, after discretizing $\Theta_i$ into $\hat{\Theta}_i$, a discretized strategy $\hat{\boldsymbol{h}}_i$ can be represented as a $|\hat{\Theta}_i| \times T$ matrix. Thus, the optimization problem of Algorithm 1 is a quadratic optimization problem over $|\hat{\Theta}_i| \cdot T$ variables:

$$\min_{\hat{\boldsymbol{h}}_i}\{(\hat{\boldsymbol{h}}_i - \hat{\boldsymbol{h}}^r_i)^\top \tilde{\nabla}^r_i + \frac{\eta^{r+1}}{2}\|\hat{\boldsymbol{h}}_i - \hat{\boldsymbol{h}}^r_i\|^2\} \text{ s.t. } \hat{\boldsymbol{h}}_i \in \hat{\mathcal{H}}_i$$

Now, the feasible set $\hat{\mathcal{H}}_i$ is specified by two types of constraints. First, for all $\hat{\theta}_i \in \hat{\Theta}_i$, the trajectory $\hat{\boldsymbol{h}}_i(\hat{\theta}_i)$ must lie in the feasible region given by the convex set $G_i(\hat{\theta}_i)$. Second, strategies $\hat{\boldsymbol{h}}_i$ must be $L_i$-Lipschitz in the $\ell_\infty$ norm. That is, we require for every $\hat{\theta}_i\hat{\theta}'_i \in \hat{\Theta}_i$ and $t \in [T]$:

$$-L_i\|\hat{\theta}_i - \hat{\theta}'_i\|_\infty \leq \hat{h}_{i,t}(\hat{\theta}_i) - h_{i,t}(\hat{\theta}'_i) \leq L_i\|\hat{\theta}_i - \hat{\theta}'_i\|_\infty$$

This yields a set of $2|\hat{\Theta}_i|^2 T$ linear constraints. Therefore, the update step is a standard convex quadratic program. In many natural scenarios, the constraints $G_i$ are *linear* (e.g. inventory constraints, no short selling); in this case, the update step is a quadratic program for which standard solvers can efficiently solve.

## 5 Experimental Evaluations with Real Data

**Empirical Bayesian Game Setup**   While there have been recent works on this topic (see Section 1.2), none of them have considered the problem of real data-based evaluation. Therefore, we now consider the problem of how to construct a Bayesian Game instance for simulated game play that is grounded in publicly available market data. This is meant to illustrate our theoretical results and of relevance for future research in this direction.

A simple approach to this is to estimate distributions of market parameters (*i.e.* $\alpha$, $\beta$, and $s$) from data, and then simulate game play under these parameter distributions. A full Bayesian game instance can then be constructed by augmenting these estimated market parameter distributions with hand-crafted sets of agent types, which can be defined in terms of type-specific constraints $G_i(\theta_i)$, type-specific value functions $f_i(\cdot; \theta_i)$, and how they correlate with market parameters and each other.

Unfortunately, past work exploring market parameter estimation from data has depended on either proprietary (*e.g.* Almgren et al. (2005)) or simulated (*e.g.* Li et al. (2024)) data, which generally depend on a level of market microstructure information that is difficult to discern in publicly available data. Given this, we devised a simple approach that can utilize publicly available forex data from Dukascopy, a Swiss financial services firm, which provides tick-level data of bid/ask prices and volumes[5]. To control for confounding effects, as well as the inherently high signal-to-noise ratio in this data, we pooled all available data for a single market (Canadian Dollar to U.S. Dollar) over the same 3-hour window (9am-12pm ET) on every Tuesday from September 2, 2025 to November 4, 2025. We estimated market parameters from this data as follows:

- Estimating $\alpha$: The permanent impact coefficient $\alpha$ represents the non-transient effect on price due to the imbalance of supply and demand. We first approximate the excess demand by taking the difference between the bid and ask volumes (actual execution data is rarely available publicly). Then we compute the mid-price – the mid-point of the bid and ask prices at each time step. Lastly, we regress the next step (rolling window of 100 ticks) mid-price change based on the average excess volume in the current step, with the regression slope capturing $\alpha$.
- Estimating $\beta$: The temporary impact coefficient is harder to estimate using a simple order-book like this. Since $\beta$ captures the premium that traders pay at execution time due to a large imbalance of buy/sell volume present, we proxy this using the bid-ask spread – the larger this value, the higher amount market makers can charge to execute an order. Excess supply or excess demand, both lead to a higher spread and thus a higher execution cost. We thus predict the ask-bid spread at the next step (rolling window of 100 ticks), based on the absolute volume imbalance at the current step.
- Estimating $s$: We averaged the excess demand at each time of day over the 3-hour period.

We present the results estimating $\alpha, \beta$ in Figures 3 and 4 in Appendix D, along with additional details on our experimental procedure.

We set up a Bayesian game instance with these estimated market parameters and two strategic agents, each with 3 possible types, with $P(\theta_1 = i, \theta_2 = j) = \frac{1}{9}$ for all $i, j \in \{1, 2, 3\}$. For each agent, we imposed type-specific constraints on their final position $(-V_i(\theta_i) \le \mathbf{1}^T \mathbf{h}_i(\theta_i) \le V_i(\theta_i))$, and used value functions defined using type-specific reserve prices on their final position $(f_i(\mathbf{h}_i) = r_i(\theta_i) \mathbf{1}^T \mathbf{h}_i)$. See Figure 1 for details on these parameters. We use a fixed, type-independent value of $s$, gathered as described above. As for the price parameters, $P(\alpha|\theta_1, \theta_2)$ and $P(\beta|\theta_1, \theta_2)$ are Gaussian distributions whose means $\mathbb{E}[\alpha|\theta_1, \theta_2]$ and $\mathbb{E}[\beta|\theta_1, \theta_2]$ are type-dependent. They are within $\pm 1e^{-8}$ and $\pm 1e^{-7}$ of $\alpha^*$ and $\beta^*$ respectively for the left plot and within $\pm 1e^{-8}$ and $\pm 1e^{-6}$ of $\alpha^*$ and $10\beta^*$ for the right plot. That is, the left part of Figure 1 has $\beta$ values centered

---

[5]All raw data and corresponding code will be made publicly available upon publication and are available upon request.

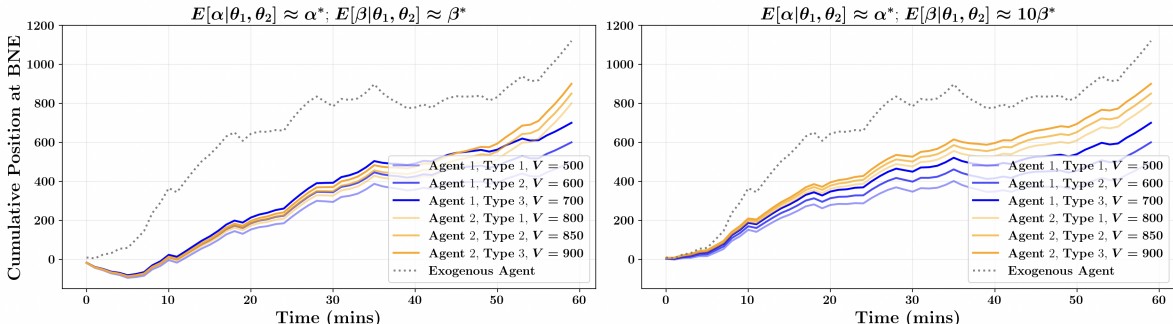

Figure 1: Cumulative position over time for agents under the BNE. The type conditioned expected reserves $\mathbb{E}[r_i|\theta_i]$ are $(1.4, 1.405, 1.41)$ and $(1.415, 1.42, 1.425)$ for agents 1 and 2 respectively. $p_0 = 1.395$ and the type conditioned market parameters $\alpha$ and $\beta$ are gaussian distributed whose means are as given in the subplot title. $\alpha^* = 4.65e^{-7}$ and $\beta^* = 3.25e^{-6}$ are the market parameters estimated from regression. See Appendix D for more details.

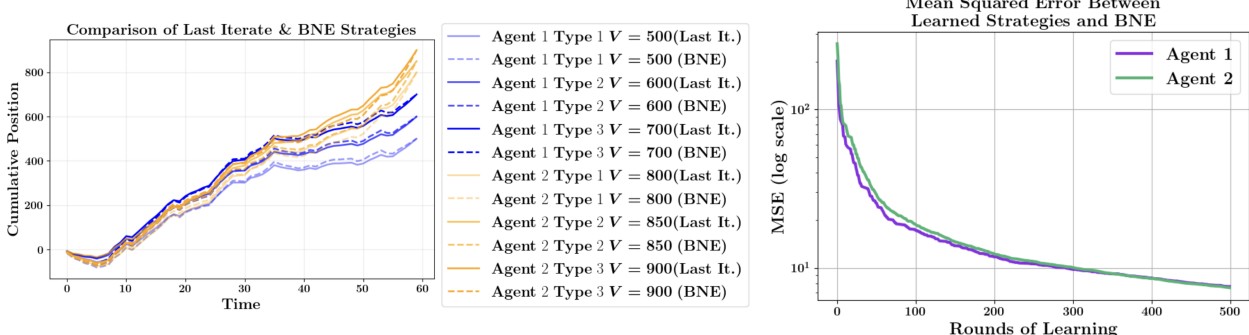

Figure 2: Comparison of Algorithm 1 (over 500 rounds) to exact BNE strategies: (left) we plot the last-iterate strategies returned by Algorithm 1 (solid lines) along with the true BNE (dashed lines) for all agents and types; and (right) we show the convergence in mean-squared error between the strategies from Algorithm 1 and the BNE over the 500 rounds.

and close on the regression predictions; the right part considers $\beta$ values roughly ten times larger. This is to evaluate how the $\alpha$ to $\beta$ ratio affects equilibrium strategies.

**Bayesian Nash Equilibrium Simulations:** We present the equilibrium results in Figure 1 computed using the extra-gradient algorithm Korpelevich (1976). We notice that in the left plot where temporary impact $\beta$ is lower, traders in equilibrium initially take advantage of the (1) high demand from the exogenous party/market aggregate $s$ and (2) high price relative to their reserve. This leads them to start by initially selling to increase profit/utility. As this dampens the price and the aggregate market demand weakens, they switch to buying and building their position and end up close to their upper bound constraint. However, when temporary price impact is higher (right side of the figure), this arbitraging behaviour becomes costly. In this regime, the agent strategies are much more proportional to the market aggregate $s$ as they build toward their final position. This strategy corresponds to the popular *Volume Weighted Average Price (VWAP)* execution algorithm: agents trade proportional to the volume of trade that occurred at that interval. We include a detailed discussion connecting our equilibrium strategy to the VWAP strategy and empirical evaluations of markets that include both types of execution in Appendix E.

**Online Learning Simulations:** We conclude with an empirical investigation on the convergence to equilibrium in the learning setting by simulating the interaction and feedback described in Section 4. We use

the same Bayesian game instance as above and have all agents follow Algorithm 1. To directly measure convergence to the BNE, we assume that in each round, agents observe $\alpha^r, \beta^r \sim P$ as defined above, without additional observation bias (though these parameters themselves are estimates of the unknown true parameters). We show the results of this simulation in Figure 2. On the left we directly compare the final iterate strategies from our learning dynamics with the BNE[6]. We see that these almost exactly overlap, with only very minor discrepancies, which can be explained by noise in the observed market parameters. On the right we plot the convergence of the agent strategies during online learning to the BNE strategies in terms of mean-squared error (MSE), which very rapidly approaches 0, even faster than guaranteed by our theory. Overall, these results validate the technical results in Theorem 4.

## 6  Discussion

This work proposes a general game theoretic model for understanding acquisition and trading of costly resources in competition, a problem with wide-ranging applicability from financial markets to compute/cloud markets. To capture such diverse settings, we start with a standard price model and then accommodate arbitrary convex constraints on the strategy, concave valuation function, and incomplete knowledge of the market and its participants which necessitates learning. We conclude by discussing some practical shortcomings of this model and the corresponding results which may motivate future work:

- In our framework, agents can trade any fractional asset amount at each time step at the endogenously-determined price $p_t$, governed by the linear/quadratic model of Almgren & Chriss (2000). While this model is widely used in the literature, it is stylized from a practical perspective. Real price impact depends also on various market microstructure features. For example, for assets traded in electronic exchanges, agents may need to interact with the corresponding limit order book, and trades may only be possible in particular discrete quantities depending on available bids and offers in the book. Incorporating additional micro-structure, including limit-order books which are widespread (see Li et al. (2024) for a detailed overview), would be a fruitful line of future work. As would extensions to the propagator models of price impact (Bouchaud et al., 2004; Gatheral & Schied, 2013; Obizhaeva & Wang, 2013).

- The learning procedure in Section 4 faces two practical limitations. First, Algorithm 1 handles continuous type spaces by constructing a $\delta$-net and optimizing over the resulting discretized strategy space. This discretization can scale exponentially with the dimension of the type space, giving an exponential dependence in the regret bound and increasing computational expense for high-dimensional type spaces.[7] A direct implementation of our algorithm is thus most practical for low-dimensional spaces. A natural direction for future work is to replace this discretization step with function approximation methods commonly used in online learning. Second, the convergence guarantee depends on the accuracy of the estimated market parameters. As discussed in Section 4, estimation errors contribute a non-vanishing term unless they decay sufficiently quickly. Our results are thus strongest in settings where market impact parameters can be estimated accurately, or where estimation improves with additional observations.

- We assume that agents commit to their full trajectory of actions $\boldsymbol{h}$ *ex-ante*, and cannot adjust dynamically to the behavior of other agents. A richer model would be to consider type-specific *policies* that map history to the action at the next time step $y$, and connect to directions in multi-agent reinforcement learning. That said, our present model can approximate this dynamic interaction by splitting time into many smaller horizons, each of which can be modeled as a separate instance of our game.

- Lastly, our results currently assume all agents are strategic, standard in game-theory. In practice, however, markets contain many types of traders with varying behaviours. In Appendix E we include an initial investigation in this direction and consider a market including both strategic traders and those who trade at a constant rate with respect to volume-weighted time (VWAP), a popular execution algorithm. We empirically analyze this in the complete-information setting and make some interesting observations. While the VWAP traders would benefit from being strategic, the strategic traders would lose out due

---

[6]While Section 4 provide guarantees with respect to the BNE under the true market parameters, here we compare Algorithm 1 to the BNE under the game constructed using our estimated market parameters, since true parameters are unknown.

[7]This is pessimistic and, in practice, we may suffer from some lower intrinsic dimension, see *e.g.* Pestov (2007); Kpotufe (2011); Pope et al. (2021).

to this switch. Interestingly, the overall welfare to the system is often greater when a subset of agents are playing VWAP. This suggests a rich line of future work in rigorously understanding the impacts of such non-strategic actions within this setting. It may also be instructive to take a mechanism design perspective to incentivize behavior that improves overall welfare.

## Broader Impact Statement

This paper studies how multiple strategic agents compete to trade costly, divisible resources under endogenous price dynamics, with applications including financial assets and cloud/compute resources. By advancing game-theoretic and learning-based methods for these settings, this may enable more accurate modeling, simulation, and potentially more efficient resource acquisition strategies. Potential positive societal impacts include improved understanding of strategic effects in these markets, which could inform better market design and help identify conditions that reduce inefficiency or instability. Potential risks are largely indirect and depend on downstream use. These concerns are not unique to our approach and are common to many advances in this area. We view the primary contribution as improving understanding of such systems.

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

## A   Proofs and Details For Section 3

**Lemma 2.** *Suppose each $\theta_i$ is fully informative about all market parameters $\boldsymbol{\lambda}$, thus giving us a complete information instance. Then the utility of agent $i$ for joint strategy $(\boldsymbol{h}_1(\theta_1), \ldots, \boldsymbol{h}_n(\theta_n))$ is strictly concave in their strategy, and is given by: $u_i(\boldsymbol{h}_i(\theta_i); \boldsymbol{h}_{-i}(\boldsymbol{\theta}_{-i}), \boldsymbol{\lambda})$*

$$= f_i(\boldsymbol{h}_i(\theta_i)) - \frac{1}{2}\boldsymbol{h}_i^T(\theta_i)Q_{\alpha,\beta}\boldsymbol{h}_i(\theta_i) - \sum_{j \neq i}(A_{\alpha,\beta}\boldsymbol{h_j}(\boldsymbol{\theta_j}))^T\boldsymbol{h}_i(\theta_i) - \boldsymbol{s}^T A_{\alpha,\beta}\boldsymbol{h}_i(\theta_i) - p_0(\boldsymbol{1}^T\boldsymbol{h}_i(\theta_i))\,,$$

*where $Q_{\alpha,\beta}$ and $A_{\alpha,\beta}$ are $n \times n$ matrices defined in terms of $\alpha$ and $\beta$, and $Q_{\alpha,\beta}$ is symmetric and positive definite.*

*Proof.* Observe that since each $\theta_i$ is fully informative, there is no need to take any expectation over $\boldsymbol{\lambda}$ and the game can be thought of as a complete information setting. Pick an arbitrary realization of $(\theta_1, \ldots, \theta_n)$ and for brevity of notation, let $\boldsymbol{h}_1 \in \mathbb{R}^T, \ldots, \boldsymbol{h}_n \in \mathbb{R}^T$ denote the agent strategies. If the utility is concave in this tuple $(\boldsymbol{h}_1, \ldots, \boldsymbol{h}_n)$ then the claim holds since the choice of types was arbitrary.

We first unroll the auto-regressive nature of the Walrasian price dynamic $p_t^w$. Observe that the following holds:

$$p_1^w = p_0 + \alpha \sum_j h_{j,1} + \alpha s_1 \; ;$$

$$p_2^w = p_0 + \alpha \sum_j h_{j,1} + \alpha s_1 + \alpha \sum_j h_{j,2} + \alpha s_2 \; ; \; \ldots$$

The execution price an agent pays is also influenced by the temporary impact. Combining this with the above, we can write the net cost an agent $i$ faces as follows: $u_i(\boldsymbol{h}_1, \ldots, \boldsymbol{h}_n, \boldsymbol{\lambda})$

$$= f_i(\boldsymbol{h}_i) - \sum_t h_{i,t}p_0 - \alpha\sum_{t=1}^{T}\sum_{\ell=1}^{t} h_{i,t}h_{i,\ell} - \alpha\sum_{t=1}^{T}\sum_{\ell=1}^{t} h_{i,t}(\sum_{j \neq i} h_{j,\ell} + s_\ell) - \beta\sum_{t=1}^{T} h_{i,t}\left(\sum_{j=1}^{n} h_{j,t} + s_t\right)$$

$$= f_i(\boldsymbol{h}_i) - \underbrace{\alpha\sum_{t=1}^{T}\left(h_{i,t}^2 + h_{i,t}\sum_{\ell=1}^{t-1} h_{i,\ell}\right) - \beta\sum_{t=1}^{T} h_{i,t}^2}_{\text{quadratic terms}}$$

$$\underbrace{-\alpha\sum_{t=1}^{T} h_{i,t}\sum_{\ell=1}^{t}\sum_{j \neq i} h_{j,\ell} - \beta\sum_{t=1}^{T} h_{i,t}\sum_{j \neq i} h_{j,t}}_{\text{linear terms} \propto \text{other agent}} \underbrace{-\alpha\sum_{t=1}^{T} h_{i,t}\sum_{\ell=1}^{t} s_\ell - \beta\sum_{t=1}^{T} h_{i,t}s_t}_{\text{linear term} \propto \text{exogenous agent}} - \sum_t p_0 h_{i,t}$$

Focusing on the quadratic terms, it suffices to compute the Hessian, denoted by $Q_{\alpha,\beta}$. Note that $Q_{\alpha,\beta}[t,t] = 2\alpha + 2\beta$. As for the off-diagonal values, these are composed entirely of $\alpha$. Indeed, for any $t_1 \neq t_2$, we have that $Q_{\alpha,\beta}[t_1, t_2] = \alpha$. Next, we consider the linear terms that are proportional to other agents. We wish to express it in the following form: $\sum_{j \neq i}(A_{\alpha,\beta}\boldsymbol{h}_j)^T\boldsymbol{h}_i$. For a given $t$, consider the first of the two linear terms proportional to others. For any $j$, observe that $h_{i,t}$ is multiplied by $\alpha h_{j,1}, \ldots \alpha h_{j,1}$. As for the second term, it multiplies $h_{i,t}$ with $\beta h_{j,t}$. Hence, we conclude that $A_{\alpha,\beta}$ is a lower-triangular matrix, whose diagonals are $\alpha + \beta$ and the remaining values are $\alpha$. As for the linear term with respect to the exogenous agent, it follows a similar pattern, and we can express it as $(A_{\alpha,\beta}\boldsymbol{s})^T\boldsymbol{h}_i$. We thus have the following expression for the matrices $Q_{\alpha,\beta}$ and $A_{\alpha,\beta}$:

$$Q_{\alpha,\beta}[i,j] = \begin{cases} \alpha & \text{if } i < j \\ 2\alpha + 2\beta & \text{if } i = j \\ \alpha & \text{if } i > j \end{cases} \; ; \; A_{\alpha,\beta}[i,j] = \begin{cases} 0 & \text{if } i < j \\ \alpha + \beta & \text{if } i = j \\ \alpha & \text{if } i > j \end{cases} \; ;$$

Since $Q_{\alpha,\beta}$ is a symmetric matrix that can be written as $Q_{\alpha,\beta} = \alpha J + (\alpha + 2\beta)I$ where $J$ is the all 1s matrix and $I$ the identity matrix. Observe that for any $x$, we have that:

$$x^T Q_{\alpha,\beta} x = (\alpha + 2\beta)(x^T I x) + \alpha(x^T J x) = (\alpha + 2\beta)(x^T x) + \alpha(x^T J x) = (\alpha + 2\beta)||x||_2^2 + \alpha\left(\sum_{i=1}^T x_i\right)^2 > 0$$

where the strict inequality holds since the parameters $\alpha, \beta$ are non-negative and $\boldsymbol{x} \neq 0$. In other words, the $Q_{\alpha,\beta}$ matrix is positive definite and thus the utility of each agent, in terms of their own strategy, is a strictly concave function. $\qquad\square$

## A.1 Proof of Lemma 1

*Proof.* In the most general sense, observe that agent $i$'s best response for a realized type $\theta_i$ allows them to play a mixed strategy over all valid strategies: $p_i(\boldsymbol{h}_i|\theta_i)$, where $\boldsymbol{h}_i$ is a vector in $\mathbb{R}^T$ since the probability is already conditioned on $\theta_i$. Suppose the remaining agents are playing some mixed, possibly correlated strategy $\sigma_{-i}$, where $\sigma_{-i}(\boldsymbol{h}_{-i}|\theta_{-i})$ denotes the probability that the remaining agents play strategy $\boldsymbol{h}_{-i} \in G_{-i}(\boldsymbol{\theta}_{-i})$ when their joint type realization is some $\theta_{-i}$. We can then express agent $i$'s best response problem as follows (note that $G_i(\theta_i) \subseteq \mathbb{R}^T$):

$$\mathrm{br}_i(\theta_i, \sigma_{-i}) = \underset{p_i(\cdot|\theta_i)\in\Delta(G_i(\theta_i))}{\arg\max} \int_{\boldsymbol{h}_i} p_i(\boldsymbol{h}_i;\theta_i) \int_{\theta_{-i}} \int_{\boldsymbol{s},\alpha,\beta} P(\theta_{-i},\boldsymbol{s},\alpha,\beta|\theta_i) \int_{\boldsymbol{h}_{-i}} \sigma_{-i}(\boldsymbol{h}_{-i}|\theta_{-i}) u_i(\boldsymbol{h}_i;\boldsymbol{h}_{-i},\boldsymbol{\lambda})$$

noting that in the case of discrete type spaces, $\int_{\boldsymbol{\theta}_{-i}}$ is replaced by $\sum_{\theta_{-i}}$. The linearity of the integral and the fact that $\int_{\boldsymbol{h}_i} p_i(\boldsymbol{h}_1|\theta_i)d\boldsymbol{h}_i = 1$ means that a maximum must exists at a vertex/pure strategy. If multiple pure strategies are optimal, then any linear combination (a mixed strategy) would also be a best-response. However, if there is a unique pure strategy maximizing this, then it means any mixed strategy must be strictly sub-optimal. In other words, it suffices to show that the pure-strategy best-response is unique even when others' play mixed and correlated strategies. This pure best-response problem is given by (again replace the integral with summation for discrete types):

$$\mathrm{br}_i(\theta_i, \sigma_{-i}) = \underset{\boldsymbol{h}_i \in G_i(\theta_i)}{\arg\max} \int_{\theta_{-i},\boldsymbol{\lambda}} P(\theta_{-i},\boldsymbol{\lambda}|\theta_i) \int_{\boldsymbol{h}_{-i}} \sigma_{-i}(\boldsymbol{h}_{-i}|\theta_{-i}) u_i(\boldsymbol{h}_i;\boldsymbol{h}_{-i},\boldsymbol{\lambda})$$

$$= \underset{\boldsymbol{h}_i \in G_i(\theta_i)}{\arg\max} \int_{\theta_{-i},\boldsymbol{\lambda},\boldsymbol{h}_{-i}} P(\theta_{-i},\boldsymbol{\lambda}|\theta_i)\sigma_{-i}(\boldsymbol{h}_{-i}|\theta_{-i})\left[f_i(\boldsymbol{h}_i) - \boldsymbol{h}_i^T Q_{\alpha,\beta}\boldsymbol{h}_i - \sum_{j\neq i}\boldsymbol{h}_j^T A_{\alpha,\beta}\boldsymbol{h}_i - \boldsymbol{s}^T A_{\alpha,\beta}\boldsymbol{h}_i\right]$$

$$= \underset{\boldsymbol{h}_i \in G_i(\theta_i)}{\arg\max} \int_{f_i\in F} f_i(\boldsymbol{h}_i)d\mu(f_i) - \boldsymbol{h}_i^T Q_{\alpha,\beta}\boldsymbol{h}_i$$

$$- \underbrace{\left[\int_{\theta_{-i}}\int_{\boldsymbol{s},\alpha,\beta} P(\theta_{-i},\boldsymbol{s},\alpha,\beta|\theta_i)\int_{\boldsymbol{h}_{-i}}\sigma_{-i}(\boldsymbol{h}_{-i}|\theta_{-i})\sum_{j\neq 1}\boldsymbol{h}_j^T A_{\alpha,\beta} + \boldsymbol{s}^T A_{\alpha,\beta}\right]}_{\boldsymbol{w}^T(\cdot)} \boldsymbol{h}_i$$

$$= \underset{\boldsymbol{h}_i \in G_i(\theta_i)}{\arg\max} f_i^*(\boldsymbol{h}_i) - \boldsymbol{h}_i^T Q_{\alpha,\beta}\boldsymbol{h}_i - \boldsymbol{w}^T(\cdot)\boldsymbol{h}_i$$

where $\mu(f_i)$ is any finite non-negative measure on the function space $F$, and $f^*$ is the result of the integral. The concavity of the function class $F$ and non-negativity of measure $\mu$ ensure that $f^*$ is concave Rockafellar & Wets (2009). Next, we note that $\boldsymbol{w}^T(\cdot)$ is a $T$ dimensional vector that does not depend on the $\boldsymbol{h}_i$. Thus, the objective faced by buyer $i$ is strictly concave (since $Q_{\alpha,\beta}$ is a PD matrix – see Lemma 2) and there is a unique solution. This immediately implies that a mixed strategy will always be a sub-optimal best-response. $\qquad\square$

**Lemma 3.** *For non-negative market parameters $\alpha, \beta$ with at least one of them being positive (i.e. $\alpha > 0$ or $\beta > 0$), define the block $M_{\alpha,\beta}$ as follows (see lemma 2 for definitions of matrix $Q_{\alpha,\beta}$ and $A_{\alpha,\beta}$):*

$$M_{\alpha,\beta} = \begin{bmatrix} Q_{\alpha,\beta} \in \mathbb{R}^{T \times T} & A_{\alpha,\beta} \in \mathbb{R}^{T \times T} & \dots & A_{\alpha,\beta} \in \mathbb{R}^{T \times T} \\ A_{\alpha,\beta} \in \mathbb{R}^{T \times T} & Q_{\alpha,\beta} \in \mathbb{R}^{T \times T} & \dots & A_{\alpha,\beta} \in \mathbb{R}^{T \times T} \\ \vdots & \vdots & \vdots & \vdots \\ A_{\alpha,\beta} \in \mathbb{R}^{T \times T} & A_{\alpha,\beta} \in \mathbb{R}^{T \times T} & \dots & Q_{\alpha,\beta} \in \mathbb{R}^{T \times T} \end{bmatrix} \tag{1}$$

*Then the symmetric component of this matrix $M_{\alpha,\beta}^s = \frac{1}{2}(M_{\alpha,\beta} + M_{\alpha,\beta}^T)$ is positive definite. This implies for any $\boldsymbol{x} \in \mathbb{R}^{nT}$: $\boldsymbol{x}^T M_{\alpha,\beta} \boldsymbol{x} > 0$.*

*Proof.* The matrix $M_{\alpha,\beta}^s$ is an $n \times n$ block matrix with $Q_{\alpha,\beta}$ on the diagonal (since $Q_{\alpha,\beta}$ is symmetric) and all other elements being $A_{\alpha,\beta}^s = \frac{1}{2}(A_{\alpha,\beta} + A_{\alpha,\beta}^T)$. This can be succinctly represented using the Kronecker product (let $J_n$ is an $n \times n$ all 1s matrix and $I_n$ is an $n \times n$ identity matrix):

$$M_{\alpha,\beta}^s = I_n \otimes (Q_{\alpha,\beta} - A_{\alpha,\beta}^s) + J_n \otimes A_{\alpha,\beta}^s \tag{2}$$

Note that the all 1s matrix is positive-definite with one eigenvalue of $n$ and all other eigenvalues 0. Therefore, we can write $\Lambda_{J_n} = U^T J_n U$, where $\Lambda_{J_n} = \text{diag}(n, 0, \dots 0)$. Let $P = U \otimes I_T$, and note that $P^T P = (U^T \otimes I_T)(U \otimes I_T) = U^T U \otimes I_T = I_{nT}$, where we use the mixed product property of Kronecker products. We shall be using $P$ to diagonalize (in the block sense) the matrix $M_{\alpha,\beta}^s$. Specifically, observe that due to the mixed product rule:

$$P^T M_{\alpha,\beta}^s P = P^T (I_n \otimes Q_{\alpha,\beta} - A_{\alpha,\beta}^s) P + P^T (J_n \otimes A_{\alpha,\beta}^s) P$$
$$= (U^T \otimes I_T)(I_n \otimes Q_{\alpha,\beta} - A_{\alpha,\beta}^s)(U \otimes I_T) + (U^T \otimes I_T)(J_n \otimes A_{\alpha,\beta}^s)(U \otimes I_T)$$
$$= (U^T I_n U \otimes I_T (Q_{\alpha,\beta} - A_{\alpha,\beta}^s) I_T) + (U^T J_n U \otimes I_T A_{\alpha,\beta}^s I_T)$$
$$= (I_n \otimes (Q_{\alpha,\beta} - A_{\alpha,\beta}^s)) + (\Lambda_{J_n} \otimes A_{\alpha,\beta}^s)$$

The first summand is a block diagonal matrix with $Q_{\alpha,\beta} - A_{\alpha,\beta}^s$ in each entry, and the second summand is also block diagonal with $nA_{\alpha,\beta}^s$ in the first entry and 0 elsewhere. Therefore, $P^T M_{\alpha,\beta}^s P$ results in a block diagonal matrix $\text{diag}(Q_{\alpha,\beta} + (n-1)A_{\alpha,\beta}^s, Q_{\alpha,\beta} - A_{\alpha,\beta}^s, \dots, Q_{\alpha,\beta} - A_{\alpha,\beta}^s)$. The eigenvalues of $M_{\alpha,\beta}^s$ are the eigenvalues of this block diagonal matrix, which in turn are the eigenvalues of each matrix in the diagonal. Thus, we need to show that $Q_{\alpha,\beta} + (n-1)A_{\alpha,\beta}^s$ and $Q_{\alpha,\beta} - A_{\alpha,\beta}^s$ both have positive eigenvalues. Note that $Q_{\alpha,\beta} = (\alpha + 2\beta)I_T + \alpha J_T$ and $A_{\alpha,\beta}^s = (\frac{\alpha}{2} + \beta)I_T + \frac{\alpha}{2}J_T$. Thus, for any $\boldsymbol{x} \in \mathbb{R}^T$:

$$\boldsymbol{x}^T (Q_{\alpha,\beta} - A_{\alpha,\beta}^s)\boldsymbol{x}^T = \boldsymbol{x}^T \left[ (\tfrac{\alpha}{2} + \beta)I_T + \tfrac{\alpha}{2}J_T \right] \boldsymbol{x} = (\tfrac{\alpha}{2} + \beta)\boldsymbol{x}^T \boldsymbol{x} + \tfrac{\alpha}{2}\left( \sum_{t=1}^T x_t \right)^2 > 0$$

$$\boldsymbol{x}^T (Q_{\alpha,\beta} + (n-1)A_{\alpha,\beta}^s)\boldsymbol{x}^T = \boldsymbol{x}^T \left[ (n+1)(\tfrac{\alpha}{2} + \beta)I_T + (n+1)\tfrac{\alpha}{2}J_T \right] \boldsymbol{x}$$

$$= (n+1)(\tfrac{\alpha}{2} + \beta)\boldsymbol{x}^T \boldsymbol{x} + (n+1)\tfrac{\alpha}{2}\left( \sum_{t=1}^T x_t \right)^2 > 0$$

as long as either $\alpha > 0$ or $\beta > 0$. Since these diagonal matrices are positive definite, they have positive eigenvalues, implying $M_{\alpha,\beta}^s$ has positive eigenvalues and is thus positive definite. $\square$

## A.2 Proof of Theorem 1

*Proof.* We first express the agent best response from a minimization perspective. That is, each agent's best-response for type realization $\theta_i$ is: $\arg\min_{\boldsymbol{h}_i \in G_i} \mathbb{E}_{\boldsymbol{\theta}_{-i}, \boldsymbol{\lambda}|\theta_i}[c_i(\boldsymbol{h}_i, \boldsymbol{h}_{-i}, \boldsymbol{\lambda})]$, where $c_i(\boldsymbol{h}_i, \boldsymbol{h}_{-i}, \boldsymbol{\lambda}) = $

$-u_i(\boldsymbol{h}_i, \boldsymbol{h}_{-i}, \boldsymbol{\lambda})$. Note that by definition, at an $n$ agent BNE with pure strategies for each type, the following must hold:

$$\forall i \in [n], \forall \theta_i \in [k], \forall \boldsymbol{h}' \in \mathcal{H}_i : \mathbb{E}_{\boldsymbol{\theta}_{-i}, \boldsymbol{\lambda}}[c_i(\boldsymbol{h}_i^{eq}(\theta_i), \boldsymbol{h}_{-i}^{eq}(\boldsymbol{\theta}_{-i}), \boldsymbol{\lambda}) - c_i(\boldsymbol{h}', \boldsymbol{h}_{-i}^{eq}(\boldsymbol{\theta}_{-i}), \boldsymbol{\lambda})] \leq 0$$

Since the expected utility/cost is a smooth function and at equilibrium everyone is playing best response, this can equivalently expressed as follows: for any agent and realized type, there can exist no feasible direction at the equilibrium at which the expected cost is decreasing Rockafellar & Wets (2009). This can be expressed as the following variational inequality:

$$\forall i \in [n], \forall \theta_i, \forall \boldsymbol{h}'(\theta_i) \in G_i(\theta_i) \subseteq \mathbb{R}^T : \langle \nabla_{\boldsymbol{h}_i(\theta_i)} \mathbb{E}[c_i(\boldsymbol{h}_i^{eq}(\theta_i); \boldsymbol{h}_{-i}^{eq}(\theta_{-i}), \boldsymbol{\lambda})], (\boldsymbol{h}_i'(\theta_i) - \boldsymbol{h}_i^{eq}(\theta_i)) \rangle_{\mathcal{H}_i} \geq 0$$

Importantly, this characterization is exact even if the derivatives are scaled by a distinct constant. Formally, a set of strategies are at a BNE if and only if the following holds for any choice of $\gamma_{i\ell}$ – we will choose $\gamma_{i,\ell} = P(\theta_i)$, the marginal probability of an agent $i$ being of type $\theta_i$:

$$\forall i \in [n], \forall \theta_i, \forall \boldsymbol{h}'(\theta_i) \in G_i(\theta_i) : \langle \gamma_{i,\theta_i} \nabla_{\boldsymbol{h}_i(\theta_i)} \mathbb{E}_{\boldsymbol{\theta}_{-i}, \boldsymbol{\lambda}}[c_i(\boldsymbol{h}_i^{eq}(\theta_i), \boldsymbol{h}_{-i}^{eq}(\boldsymbol{\theta}_{-i}), \boldsymbol{\lambda})], (\boldsymbol{h}'(\theta_i) - \boldsymbol{h}_i^{eq}(\theta_i)) \rangle_{\mathcal{H}_i} \geq 0 \quad (3)$$

With our choice of scaling $\gamma_{i,\ell}$, and switching the order of gradients and expectation, we have that for any $i, \theta_i$:

$$\gamma_{i,\theta_i} \nabla_{\boldsymbol{h}_i(\theta_i)} \mathbb{E}_{\boldsymbol{\theta}_{-i}, \boldsymbol{\lambda}}[c_i(\boldsymbol{h}_i^{eq}(\theta_i), \boldsymbol{h}_{-i}^{eq}(\boldsymbol{\theta}_{-i}), \boldsymbol{\lambda})]$$

$$= P(\theta_i) \Bigg( \int_{\boldsymbol{\theta}_{-i}} \int_{\boldsymbol{\lambda}} Q_{\alpha,\beta} \boldsymbol{h}_i(\theta_i) P(\boldsymbol{\theta}_{-i}, \boldsymbol{\lambda}|\theta_i) d\boldsymbol{\lambda} d\boldsymbol{\theta}_{-i}$$

$$+ \int_{\boldsymbol{\theta}_{-i}} \int_{\boldsymbol{\lambda}} \Bigg[ \sum_{j \neq i} A_{\alpha,\beta} \boldsymbol{h}_j(\theta_j) + B_{\alpha,\beta} s \Bigg] P(\boldsymbol{\theta}_{-i}, \boldsymbol{\lambda}|\theta_i) d\boldsymbol{\lambda} d\boldsymbol{\theta}_{-i} - \nabla_{\boldsymbol{h}_i(\theta_i)} \int_{\boldsymbol{\lambda}} f_i(\boldsymbol{h}_i(\theta_i)) d\mu(f_i|\theta_i) \Bigg)$$

$$= P(\theta_i) \Bigg[ \int_{\alpha,\beta} Q_{\alpha,\beta} P(\alpha, \beta|\theta_i) d(\alpha,\beta) \Bigg] \boldsymbol{h}_i(\theta_i) + P(\theta_i) \sum_{j \neq i} \int_{\theta_j} \int_{\alpha,\beta} A_{\alpha,\beta} \boldsymbol{h}_j(\theta_j) \int_{\boldsymbol{\theta}_{-(i,j)}} \int_s P(\theta_j, \boldsymbol{\theta}_{-(i,j)}, \boldsymbol{\lambda}|\theta_i)$$

$$+ \underbrace{\int_{\alpha,\beta,s} B_{\alpha,\beta} s P(\boldsymbol{\lambda}, \theta_i) d(\alpha, \beta, s)}_{b_{i,\theta_i}} - P(\theta_i) \nabla_{\boldsymbol{h}_i(\theta_i)} \underbrace{\int_{f_i \in F} f_i(\boldsymbol{h}_i(\theta_i)) d\mu(f_i|\theta_i)}_{f_{i,\theta_i}^*(\boldsymbol{h}_i(\theta_i))}$$

$$= P(\theta_i) \Bigg[ \underbrace{\int_{\alpha,\beta} Q_{\alpha,\beta} P(\alpha,\beta|\theta_i) d(\alpha,\beta)}_{Q_{i,\theta_i}^* \in \mathbb{R}^{T \times T}} \Bigg] \boldsymbol{h}_i(\theta_i) + \sum_{j \neq i} \int_{\theta_j} P(\theta_j, \theta_i) \Bigg[ \underbrace{\int_{\alpha,\beta} A_{\alpha,\beta} P(\alpha, \beta|\theta_j, \theta_i) d(\alpha,\beta)}_{A_{i,\theta_i,j,\theta_j}^* \in \mathbb{R}^{T \times T}} \Bigg] \boldsymbol{h}_j(\theta_j) d\theta_j$$

$$+ b_{i,\theta_i} - P(\theta_i) \cdot \nabla_{\boldsymbol{h}_i(\theta_i)} f_{i,\theta_i}^*(\boldsymbol{h}_i(\theta_i))$$

where in the last transition, we observe: $P(\theta_j, \alpha, \beta|\theta_i) \cdot P(\theta_i) = P(\alpha, \beta, \theta_j, \theta_i) = P(\alpha, \beta|\theta_i, \theta_j) P(\theta_i, \theta_j)$. We also note that $\mu(f_i|\theta_i)$ is a finite non-negative measure on the function space $\mathcal{F}$, and $f_{i,\theta_i}^*$ is the result of the functional integral. The concavity of the function class $\mathcal{F}$ and non-negativity of measure $\mu$ ensure that $f_{i,\theta_i}^*$ is a concave function Rockafellar & Wets (2009). For discrete type spaces, the derivation above would be identical except for the integral over types replaced by sums.

Let $\mathcal{H}_i^*$ denote a function space $L_2(\Theta_i, P; \mathbb{R}^T)$ where $\langle \boldsymbol{h}_i, \boldsymbol{h}_i' \rangle_{\mathcal{H}_i^*} = \mathbb{E}_{\theta_i}[\langle \boldsymbol{h}_i(\theta_i), \boldsymbol{h}_i'(\theta_i) \rangle]$. Observe that $\mathcal{H}_i \subseteq \mathcal{H}_i^*$. Similarly, defining $\mathcal{H} = \prod_{i=1}^n \mathcal{H}_i$ and $\mathcal{H}^* = \prod_{i=1}^n \mathcal{H}_i^*$, $\mathcal{H} \subseteq \mathcal{H}^*$. Observe that we can denote the operator of this variational inequality exactly characterizing the BNE by $F : \mathcal{H} \to \mathcal{H}$. This operator can be decomposed into a linear, non-linear, and constant component: $F = F^{\text{lin}} + F^{\text{nonlin}} + F^{\text{const}}$. We use $\boldsymbol{H} = [\boldsymbol{h}_1, \ldots, \boldsymbol{h}_n] \in \mathcal{H}$ to denote the joint strategies of all players, and $\langle \boldsymbol{H}, \boldsymbol{H}' \rangle_{\mathcal{H}} = \mathbb{E}_{\boldsymbol{\theta}}[\langle [\boldsymbol{h}_1(\theta_1); \ldots; \boldsymbol{h}_n(\theta_n)], [\langle [\boldsymbol{h}_1'(\theta_1); \ldots; \boldsymbol{h}_n'(\theta_n)] \rangle]$.

$$F^{\text{lin}} : [F_i^{\text{lin}}(\boldsymbol{H})](\theta_i) = P(\theta_i) Q_{i,\theta_i}^* \boldsymbol{h}_i(\theta_i) + \sum_{j \neq i} \int_{\theta_j} P(\theta_j, \theta_i) A_{i,\theta_i,j,\theta_j}^* \boldsymbol{h}_j(\theta_j) d\theta_j$$

$$F^{\text{const}} : [F_i^{\text{const}}(\boldsymbol{H})](\theta_i) = b_{i,\theta_i}$$

$$F^{\text{nonlin}} : [F_i^{\text{nonlin}}(\boldsymbol{H})](\theta_i) = -P(\theta_i) \cdot \nabla_{\boldsymbol{h}_i(\theta_i)} f_{i,\theta_i}^*(\boldsymbol{h}_i(\theta_i))$$

We now argue that the operator $F$ is *strongly monotone* on the set $\mathcal{H} = \prod_{i=1}^{n}$. That is, there exists a scaler $c$ such that: $\langle F(\boldsymbol{H}) - F(\boldsymbol{H}'), (\boldsymbol{H} - \boldsymbol{H}')\rangle_{\mathcal{H}} \geq c||\boldsymbol{H} - \boldsymbol{H}'||_{\mathcal{H}}^2$, $\forall \boldsymbol{H}, \boldsymbol{H}' \in \mathcal{H}$. Starting with $F^{\mathrm{nonlin}}$, it is clear that since each $f_{i,\theta_i}^*$ is an integral over concave functions, it is concave function. Thus, $-f_{i,\theta_i}^*$ is a convex function and it is known that the gradient of such functions is a monotone operator. Formally:

$$\langle F^{\mathrm{nonlin}}(\boldsymbol{H}) - F^{\mathrm{nonlin}}(\boldsymbol{H}'), \boldsymbol{H} - \boldsymbol{H}'\rangle_{\mathcal{H}} = -\sum_{i=1}^{n}\int_{\theta_i} P(\theta_i)\langle\nabla_{\boldsymbol{h}_i(\theta_i)} f_{i,\theta_i}^*(\boldsymbol{h}_i(\theta_i)) - \nabla_{\boldsymbol{h}_i'(\theta_i)} f_{i,\theta_i}^*(\boldsymbol{h}_i'(\theta_i))\rangle_{\mathbb{R}^T} d\theta_i \geq 0$$

Turning next to the linear component of the operator, observe we can express this as follows:

$$\langle F^{\mathrm{lin}}(\boldsymbol{H}), \boldsymbol{H}\rangle_{\mathcal{H}} = \sum_{i=1}^{n}\int_{\theta_i}\left[P(\theta_i)\boldsymbol{h}_i^T(\theta_i)Q_{i,\theta_i}^*\boldsymbol{h}_i(\theta_i)d\theta_i + \sum_{j\neq i}\int_{\theta_j} P(\theta_j,\theta_i)\boldsymbol{h}_i^T(\theta_i)A_{i,\theta_i,j,\theta_j}^*\boldsymbol{h}_j(\theta_j)d(\theta_j)\right]$$

$$= \int_{\boldsymbol{\theta}}\int_{\alpha,\beta} P(\boldsymbol{\theta},\alpha,\beta)\left[\sum_{i=1}^{n}\boldsymbol{h}_i^T(\theta_i)Q_{\alpha,\beta}\boldsymbol{h}_i(\theta_i) + \sum_{j\neq i}\boldsymbol{h}_i^T(\theta_i)A_{\alpha,\beta}\boldsymbol{h}_j(\theta_j)\right]d\boldsymbol{\theta}d(\alpha,\beta)$$

$$= \mathbb{E}_{\boldsymbol{\theta},\alpha,\beta}\left[\sum_{i=1}^{n}\boldsymbol{z}_{i,\theta_i}^T Q_{\alpha,\beta}\boldsymbol{z}_{i,\theta_i} + \sum_{i\neq j}\boldsymbol{z}_{i,\theta_i}^T A_{\alpha,\beta}\boldsymbol{z}_{j,\theta_j}\right] = \mathbb{E}_{\theta,\alpha,\beta}[\boldsymbol{z}_{\boldsymbol{\theta}}^T M_{\alpha,\beta}\boldsymbol{z}_{\boldsymbol{\theta}}]$$

where $\boldsymbol{z}_{i,\theta_i} = \boldsymbol{h}_i(\theta_i)$ is a random vector of length $T$ and for a realization $\boldsymbol{\theta}$, $\boldsymbol{z}_{\boldsymbol{\theta}} = [\boldsymbol{z}_{1,\theta_1}, \ldots, \boldsymbol{z}_{n,\theta_n}]^T \in \mathbb{R}^{nT}$ is a concatenation of these $n$ random vectors. Further, $M_{\alpha,\beta}$ is a random matrix which, for any realization of $\alpha, \beta$, is the same as the matrix in Lemma 3, the symmetric component of which we showed to be positive definite; thus, by choosing $c = \lambda_{min}(M_{\alpha,\beta}^s)$ ensures the strong monotonicity condition on the operator $M_{\alpha,\beta}$. Thus, for any $\boldsymbol{z}_{\boldsymbol{\theta}}$ and any realization realization of $(\alpha, \beta)$, there exists a $c_{\alpha,\beta}$ such that $\boldsymbol{z}_{\boldsymbol{\theta}}^T M_{\alpha,\beta}\boldsymbol{z}_{\boldsymbol{\theta}} \geq c_{\alpha,\beta}||\boldsymbol{z}_{\boldsymbol{\theta}}||^2$, when $\boldsymbol{z}_{\boldsymbol{\theta}} \neq \boldsymbol{0}$. Let $c_{min} = \min_{\alpha,\beta}\lambda_{min}(M_{\alpha,\beta}^s)$ the smallest eigenvalue possible in the distribution support of $\alpha, \beta$. We thus have:

$$\langle F^{\mathrm{lin}}(\boldsymbol{H}), \boldsymbol{H}\rangle_{\mathcal{H}} = \mathbb{E}_{\theta,\alpha,\beta}[\boldsymbol{z}_{\boldsymbol{\theta}}^T M_{\alpha,\beta}\boldsymbol{z}_{\boldsymbol{\theta}}] \geq c_{min}\mathbb{E}_{\boldsymbol{\theta}}[||\boldsymbol{z}_{\boldsymbol{\theta}}||_2^2] \tag{4}$$

$$\mathbb{E}_{\boldsymbol{\theta}}[||\boldsymbol{z}_{\boldsymbol{\theta}}||_2^2] = \sum_{i=1}^{n}\int_{\theta_i}||\boldsymbol{h}_i(\theta_i)||_2^2 P(\theta_i)d\theta_i = \sum_{i=1}^{n}||\boldsymbol{h}_i||_{\mathcal{H}_i} = ||\boldsymbol{H}||_{\mathcal{H}} \tag{5}$$

$$\implies \langle F^{\mathrm{lin}}(\boldsymbol{H}), \boldsymbol{H}\rangle_{\mathcal{H}} \geq c_{min}||\boldsymbol{H}||_{\mathcal{H}} \tag{6}$$

We can now prove our claim that the overall operator $F$ is strongly monotone. Since $F^{\mathrm{const}}$ is a constant, it cancels out under subtraction. Due to the monotonicity of $F^{\mathrm{nonlin}}$ and $F^{\mathrm{lin}}$ being linear, we have:

$$\langle F(\boldsymbol{H}) - F(\boldsymbol{H}'), (\boldsymbol{H} - \boldsymbol{H}')\rangle_{\mathcal{H}} = \langle F^{\mathrm{lin}}(\boldsymbol{H} - \boldsymbol{H}'), (\boldsymbol{H} - \boldsymbol{H}')\rangle_{\mathcal{H}} + \langle F^{\mathrm{nonlin}}(\boldsymbol{H}) - F^{\mathrm{nonlin}}(\boldsymbol{H}'), (\boldsymbol{H} - \boldsymbol{H}')\rangle_{\mathcal{H}}$$
$$\geq c_{min}||\boldsymbol{H}||_{\mathcal{H}}$$

Theorem 1.4 of Chapter 3 in Kinderlehrer & Stampacchia (2000) states that when $\mathcal{H}$ is a closed, bounded, convex, non-empty subset of $\mathcal{H}^*$, and $F$ is a strictly monotone and continuous operator, then there exists a unique solution to the variational inequality. As this solution corresponds to the BNE due to Equation 3, it suffices to verify the conditions. It is immediate that $F$ is strictly monotone (strong monotonicity implies strict monotonicity) and continuous since $F^{\mathrm{lin}}$ and $F^{\mathrm{const}}$ are affine and $F^{\mathrm{nonlin}}$ consists of the gradient of a smooth concave function. $\mathcal{H}$ by definition is the set of Lipschitz functions with feasible trajectories, which we assume to be non-empty. Next, $(B, L)$ regularity (see Definition 3) ensures that $||\boldsymbol{h}_i(\theta_i)||_2 \leq B$; $\mathcal{H}$ is thus clearly bounded. It also ensures that $\mathcal{H}_i$ is a closed and convex function class. Thus the product space $\mathcal{H}$ is as well. $\qquad\square$

**Corollary 1.** *Assuming oracle access to the operator $F$, the extra-gradient algorithm Korpelevich (1976) can obtain an $\varepsilon$-approximation to the optimal solution in $O(\frac{L}{c}\log\frac{1}{\varepsilon})$ queries.*

*Proof.* We have shown above that the variational inequality operator defining the BNE is $c$-strongly monotone. Next, we claim it is also $L$-Lipschitz. Since the operator is of the form $F^{\mathrm{lin}} + F^{\mathrm{const}} + F^{\mathrm{nonlin}}$, it

suffices to show Lipschitzness of each term. The constant and linear operator is always lipschitz, with the constant depending on the norm of the operator matrix. Since each $f \in F$ is smooth, $F^{\text{nonlin}}$ is composed of the gradient of some smooth concave functions over a bounded domain. Therefore, this is also Lipschitz, with the constant depending on the Hessian.

Theorem 3.4 of Wadia et al. (2024) states that for any $c$-strongly monotone and $L$-Lipshcitz operator $F$, the extragradient algorithm with step-size $\eta = \frac{1}{2(c+L)}$ converges to the fixed point at a linear rate of $1 - \frac{c}{4L}$. Specifically, it shows that for such an operator $F(x)$ whose unique fixed point is denoted $x^*$, and $x_k$ and $x_{k+1}$ are consecutive iterates of the extra-gradient algorithm, the following holds:

$$||x_{t+1} - x^*||^2 \leq \left(1 - \frac{c}{4L}\right) ||x_t - x^*||^2 \tag{7}$$

where $c < L$ due since (equation 53 in Wadia et al. (2024)):

$$c \leq \frac{\langle F(x) - F(y), x - y \rangle}{||x - y||^2} \leq \frac{||F(x) - F(y)||}{||x - y||} \leq L. \tag{8}$$

To relate this result to a computational framework, observe that running the extra-gradient algorithm requires two calls to the operator $F$ per iteration, and a projection oracle to ensure iterands lie within the feasible region. Assuming such oracle access, we can find a $\varepsilon > 0$ approximate fixed point – in the sense that our solution $\hat{x}$ satisfies $||\hat{x} - x^*|| \leq \varepsilon$ in $O\left(\frac{L}{c} \log \frac{1}{\varepsilon}\right)$ queries. Specifically, due to equation 7, we have that:

$$||x_t - x^*||^2 \leq \left(1 - \frac{c}{4L}\right)^t ||x_0 - x^*||^2 \tag{9}$$

Since the domain is bounded, $||x_0 - x^*|| \leq d$ where $d$ is some constant. Then it suffices to bound $||x_t - x^*||^2 \leq \varepsilon_1^2 ||x_0 - x^*||^2$ and thus:

$$\left(1 - \frac{c}{4L}\right)^t \leq \varepsilon_1^2 \implies t \log \left(1 - \frac{c}{4L}\right) \leq 2 \log \varepsilon_1. \tag{10}$$

Now since $-\log(1 - z) \geq z$ for $z \in [0, 1)$, letting $z = \frac{c}{4L}$, we have that $t \geq \frac{8L}{c} \log(\frac{1}{\varepsilon_1})$. For a given $\varepsilon$, we can choose $\varepsilon_1 = \varepsilon/d$, and we omit the dependence on $d$ as it is a constant. $\qquad \square$

A more granular bound (beyond oracle access to operator $F$) can be given for discrete type spaces. Evaluating the operator herein requires, at most, summation over types and agents pairs, as well as gradient calls to the function $f_i$. Thus for discrete type spaces, we can make a sharper statement: an $\varepsilon$ approximate solution can be obtained in $O\left((nMT)^2 \frac{L}{c} \log \frac{1}{\varepsilon}\right)$ arithmetic operations and queries to the gradient oracle, where $M = \max_i |\Theta_i|$.

### A.3 Proof of Theorem 2

*Proof.* As we consider a complete information setting, we use $\boldsymbol{h}_i \in \mathbb{R}^T$ to denote the strategy of agent $i$ and omit the dependence on $\theta_i$ for the sake of brevity.

Suppose there are $n = 2$ agents and we have some constant values of $\alpha, \beta$ – one can assume, without loss of generality, that $\alpha = \beta = 1$[8]. Let the final position utility for both agents be given by the following linear function: $f_i(\boldsymbol{h}_i) = r_i \sum_t h_{i,t}$, where $r_i$ can be interpreted as the reserve/fair-market price as perceived by agent $i$. Further, the two have box constraints on their cumulative position: $V_i^- \leq \sum_i h_{i,t} \leq V_i^+$. We shall assume the exogenous agent is not present – i.e. $\boldsymbol{s} = \boldsymbol{0}$.

For a positive constant $x$, let the initial price $p_0 = x$ and the reserve prices for the agents be $(r_1 = x, r_2 = x - \varepsilon)$, where $\varepsilon > 0$. We first consider the equilibrium of this game without any constraints. Then each

---

[8]Insofar as $\alpha, \beta$ are constants and not scaling with respect to the $\varepsilon$ all results will hold.

agent's best response is given by:

$$\text{br}_1(\boldsymbol{h}_2) = \arg\max_{\boldsymbol{h}_1} \left\{ -\frac{1}{2}\boldsymbol{h}_1^T Q_{\alpha,\beta}\boldsymbol{h}_1 - (A_{\alpha,\beta}\boldsymbol{h}_2)^T\boldsymbol{h}_1 \right\} \tag{11}$$

$$\text{br}_1(\boldsymbol{h}_2) = \arg\max_{\boldsymbol{h}_2} \left\{ -\varepsilon\boldsymbol{1}^T\boldsymbol{h}_2 - \frac{1}{2}\boldsymbol{h}_2^T Q_{\alpha,\beta}\boldsymbol{h}_2 - (A_{\alpha,\beta}\boldsymbol{h}_1)^T\boldsymbol{h}_2 \right\} \tag{12}$$

Observe that at the equilibrium of this unconstrained game, the gradient of both agents' best responses must be 0. Since this is a quadratic function, the gradient is linear, and the equilibrium can be uniquely specified by the following system of linear equalities:

$$\underbrace{\begin{bmatrix} Q_{\alpha,\beta} & A_{\alpha,\beta} \\ A_{\alpha,\beta} & Q_{\alpha,\beta} \end{bmatrix}}_{\text{Matrix } M \in \mathbb{R}^{2T \times 2T}} \begin{bmatrix} \boldsymbol{h}_1^{eq} \\ \boldsymbol{h}_2^{eq} \end{bmatrix} = \underbrace{\begin{bmatrix} \boldsymbol{0} \\ -\varepsilon \end{bmatrix}}_{\boldsymbol{z} \in \mathbb{R}^{2T}}$$

Recall that the matrices $Q_{\alpha,\beta}, A_{\alpha,\beta}$ are specified using only the terms $\alpha, \beta$. In lemma 2, we noted that $Q_{\alpha,\beta}$ is a positive-definite matrix and thus invertible. The matrix $A_{\alpha,\beta}$ is a lower triangular matrix with $\alpha + \beta$ on the diagonals and is thus also invertible (insofar as $\alpha > 0$ or $\beta > 0$). As such, the matrix $M$ above is invertible and the unconstrained equilibrium strategy is given by $M^{-1}\boldsymbol{z}$. Note that this does not depend on the value of $x$. Further, if $V_i^- \leq -||M^{-1}\boldsymbol{z}||_1$ and $V_i^+ \geq ||M^{-1}\boldsymbol{z}||_1$, then this unconstrained equilibrium is also an equilibrium in the original constrained game. As for the strategy itself, let $m_{ij}$ denote the values of $-M^{-1}$ and note that $m_{ij}$ can be seen as a scaler with respect to $\varepsilon$. Then we have that:

$$h_{1t} = \varepsilon \sum_{j=T}^{2T} m_{t,j} \quad \text{and} \quad h_{2t} = \varepsilon \sum_{j=T}^{2T} m_{T+t,j} \tag{13}$$

Given that the value of the final position is simply the product of the total amount bought and the reserve, the utility of buyer 1 (with reserve $x$) is:

$$u_{eq}^1 = x\varepsilon \underbrace{\sum_{t=1}^{T}\sum_{j=T}^{2T} m_{t,j}}_{\sum_t h_{1t}} - \sum_{t=1}^{T}\left[ \underbrace{\sum_{j=T}^{2T} m_{t,j}\varepsilon\left( x + \alpha\varepsilon\sum_{\tau=1}^{t}\sum_{j=T}^{2T}(m_{\tau,j} + m_{T+\tau,j}) + \beta\varepsilon\sum_{j=T}^{2T} m_{t,j} + m_{T+t,j} \right)}_{\text{price } p_t} \right]$$

$$= \left| \sum_{t=1}^{T}\sum_{j=T}^{2T} m_{t,j}\varepsilon^2\left( \alpha\sum_{\tau=1}^{t}\sum_{j=T}^{2T}(m_{\tau,j} + m_{T+\tau,j}) + \beta\sum_{j=T}^{2T}(m_{t,j} + m_{T+t,j}) \right) \right| = \Theta(\varepsilon^2)$$

where the absolute value in the second line follows, since utility at equilibrium will always be non-negative (the agents not trading would get utility 0, so utility at equilibrium must be at least 0). A similar analysis leads us to show that the utility of the second agent (with reserve $x - \varepsilon$) is also bounded by $\Theta(\varepsilon^2)$, allowing us to conclude that the welfare at equilibrium is $O(\varepsilon^2)$. Formally:

$$u_2^{eq} = \left| -\varepsilon^2\sum_{t=1}^{T}\sum_{j=T}^{2T} m_{T+t,j} - \sum_{t=1}^{T}\sum_{j=T}^{2T} m_{T+t,j}\varepsilon^2\left( \alpha\sum_{\tau=1}^{t}\sum_{j=T}^{2T}(m_{\tau,j} + m_{T+\tau,j}) + \beta\sum_{j=T}^{2T}(m_{t,j} + m_{T+t,j}) \right) \right|$$

We now turn to characterizing the optimal welfare of this instance. For some $\delta > 0$ (to be specified later), consider the following trajectories for each buyer (recall positive values mean buying):

$$\boldsymbol{h}_1 = [x, x, 0, \ldots, 0] \quad \text{and} \quad \boldsymbol{h}_2 = [-x - \delta, -x - \delta, 0, \ldots, 0] \tag{14}$$

Insofar as $V_i^+ \geq 2x$ and $V_i^- \leq -2x - \delta$, the trajectories above are feasible. Under this strategy, it suffices to consider the prices at rounds $t = 1, 2$, for which we have that: $p_1 = x - \alpha\delta - \beta\delta$ and $p_2 = x - 2\alpha\delta - \beta\delta$.

Then the utilities for each buyer is given by:

$$u_1 = x \cdot 2x - x(x - \alpha\delta - \beta\delta) - x(x - 2\alpha\delta - \beta\delta) = 3\alpha\delta x + 2\beta\delta x$$
$$u_2 = (x - \varepsilon)(-2x - \delta) + (x + \delta)(x - \alpha\delta - \beta\delta) + (x + \delta)(x - 2\alpha\delta - \beta\delta)$$
$$= 2\varepsilon x + 2\delta\varepsilon - 3\alpha\delta x - 3\alpha\delta^2 - 2\beta\delta x - 2\beta\delta^2$$
$$\implies u_1^{opt} + u_2^{opt} \geq 2\varepsilon x + 2\delta\varepsilon - 3\alpha\delta^2 - 2\beta\delta^2 = 2x\varepsilon + 2\delta\varepsilon - (3\alpha + 2\beta)\delta^2$$

This gives a concave quadratic (in the unspecified parameter $\delta$) lower bound on the optimal utility. Maximizing it means choosing a $\delta$ such that the gradient is 0:

$$\delta = \frac{\varepsilon}{3\alpha + 2\beta} \implies u_1^{opt} + u_2^{opt} \geq 2x\varepsilon + \frac{2\varepsilon^2}{3\alpha + 2\beta} - \frac{\varepsilon^2}{3\alpha + 2\beta} = 2x\varepsilon + \frac{\varepsilon^2}{3\alpha + 2\beta} = \Theta(x\varepsilon)$$

From here, it is evident that for any constants $\alpha, \beta$ and $x$, we can construct an $\varepsilon > 0$ parametrized instance $\mathcal{I}_\varepsilon$ with box constraints $V_i^- \leq \min(-||M^{-1}\boldsymbol{z}||, -2x - 2\delta)$ and $V_i^+ \geq \max(||M^{-1}\boldsymbol{z}||, 2x)$ with the aforementioned $\delta = \frac{\varepsilon}{3\alpha + 2\beta}$ such that:

$$\text{PoA}(\mathcal{I}_\varepsilon) = \frac{U_{opt}(\mathcal{I}_\varepsilon)}{U_{eq}(\mathcal{I}_\varepsilon)} \geq \frac{\Omega(\varepsilon)}{O(\varepsilon^2)} = \Omega\left(\frac{1}{\varepsilon}\right) \to \infty \quad \text{as } \varepsilon \to 0 \tag{15}$$

$\square$

## A.4 Proof of Theorem 3

*Proof.* We first note that if each agent has a hard constraint $V_i(\theta_i)$, then regardless of their strategy (which is conditioned on $\theta_i$), their value $f(\cdot)$ will be the same under that realization. Thus, it suffices to consider the objective of each agent $i$ in such Bayesian instances as minimizing their expected cost and ignoring $f(\cdot)$ - defined as follows:

$$\bar{c}_i(\boldsymbol{h}_i, \boldsymbol{h}_{-i}) \triangleq \mathbb{E}_{\boldsymbol{\theta}, \boldsymbol{\lambda}}[c_i(\boldsymbol{h}_i(\theta_i), \boldsymbol{h}_{-i}(\theta_{-i}), \boldsymbol{\lambda})] = \mathbb{E}_{\boldsymbol{\theta}, \boldsymbol{\lambda}}\left[\frac{1}{2}\boldsymbol{h}_i(\theta_i)^T Q_{\alpha,\beta}\boldsymbol{h}_i(\theta_i) + \sum_{j \neq i}\boldsymbol{h}_i^T(\theta_i)A_{\alpha,\beta}\boldsymbol{h}_j(\theta_j) + \boldsymbol{h}_i^T(\theta_i)A_{\alpha,\beta}\boldsymbol{s}\right]$$

Observe that for a complete strategy profile $\boldsymbol{H} = (\boldsymbol{h}_1, \ldots, \boldsymbol{h}_n)$, the total cost is given by: $\sum_{i=1}^n \bar{c}_i(\boldsymbol{h}_i, \boldsymbol{h}_{-i})$ (the PoA is the ratio between total equilibrium cost and the least total cost strategy). We next make use of two key facts. First is the standard AM-GM inequality and the second is a restatement of the smooth games framework proposed by Roughgarden (2015) which holds for any cost-minimization game.

**Fact 1.** *For any positive integers $a, b$ and for any $\varepsilon > 0$: $2ab \leq \varepsilon a^2 + \frac{1}{\varepsilon}b^2$ (by AM-GM Inequality).*

**Definition 7** (Roughgarden (2015)). *For any two valid and individually rational strategy profile $\boldsymbol{H}^* = (\boldsymbol{h}_1^*, \ldots, \boldsymbol{h}_n^*)$ and $\boldsymbol{H} = (\boldsymbol{h}_1, \ldots, \boldsymbol{h}_n)$ of a cost-minimization game, the game is* smooth *if there exists constants $\lambda > 0$ and $\mu < 1$ such that:*

$$(LHS) \quad \sum_i \bar{c}_i(\boldsymbol{h}_i^*, \boldsymbol{h}_{-i}) \leq \lambda \sum_i \bar{c}_i(\boldsymbol{H}^*) + \mu \sum_i \bar{c}_i(\boldsymbol{H}) \quad (RHS) \tag{16}$$

Due to the linearity of expectation, one way of showing equation 16 holds is by proving that for every realization of $\theta = (\theta_1, \ldots, \theta_n)$ and $\boldsymbol{\lambda}$:

$$(\text{LHS}) \quad \sum_i c_i(\boldsymbol{h}_i^*, \boldsymbol{h}_{-i}, \boldsymbol{\lambda}) \leq \lambda \sum_i c_i(\boldsymbol{H}^*, \boldsymbol{\lambda}) + \mu \sum_i c_i(\boldsymbol{H}, \boldsymbol{\lambda}) \quad (\text{RHS}) \tag{17}$$

Pick any such realization and, for ease of notation, let $\boldsymbol{H}^* \in \mathbb{R}^{n \times T}, \boldsymbol{H} \in \mathbb{R}^{n \times T}$ denote the joint strategies used by the agents on this realization, with $\boldsymbol{h}_i \in \mathbb{R}^T$ denoting the specific strategy of agent $i$. Next, we recall that the matrix $A_{\alpha,\beta}$ in the cost function is lower triangular. We define $A_{\alpha,\beta}^s = \frac{1}{2}(A_{\alpha,\beta} + A_{\alpha,\beta}^T)$ as the

symmetric version of this matrix. Observe that under the definition of matrix $A_{\alpha,\beta}$ and $Q_{\alpha,\beta}$, we have that $A_{\alpha,\beta}^s = \frac{1}{2}Q_{\alpha,\beta}$. In addition, let $A_{\alpha,\beta}^a = A_{\alpha,\beta} - A_{\alpha,\beta}^s$ denote the remaining (skew-symmetric) component. Finally, we define $\tilde{A}_a = Q_{\alpha,\beta}^{-1/2}A_{\alpha,\beta}^a Q_{\alpha,\beta}^{-1/2}$, and $\kappa_A = ||\tilde{A}_a||_2$.

Let $c_{total}(\boldsymbol{H},\boldsymbol{\lambda}) = \sum_i c_i(\boldsymbol{H},\boldsymbol{\lambda})$. Further, let $\boldsymbol{z}_i = Q_{\alpha,\beta}^{1/2}\boldsymbol{h}_i$ and $\boldsymbol{z}^* = Q_{\alpha,\beta}^{1/2}\boldsymbol{h}_i^*$. Since $Q_{\alpha,\beta}$ is symmetric and positive definite, we note that $Q_{\alpha,\beta}^{1/2}$ is symmetric. Then observe that:

$$
\begin{aligned}
c_{total}(\boldsymbol{H}) &= \sum_i \boldsymbol{h}_i^T A_{\alpha,\beta}\boldsymbol{s} + \sum_i \tfrac{1}{2}\boldsymbol{h}_i^T Q_{\alpha,\beta}\boldsymbol{h}_i + \sum_{(i\neq j)} \boldsymbol{h}_i^T A_{\alpha,\beta}\boldsymbol{h}_j \\
&= \sum_i \boldsymbol{h}_i^T A_{\alpha,\beta}\boldsymbol{s} + \sum_i \tfrac{1}{2}\boldsymbol{h}_i^T Q_{\alpha,\beta}\boldsymbol{h}_i + \sum_{i<j} \boldsymbol{h}_i^T A_{\alpha,\beta}\boldsymbol{h}_j + \boldsymbol{h}_j^T A_{\alpha,\beta}\boldsymbol{h}_i \\
&= \sum_i \boldsymbol{h}_i^T A_{\alpha,\beta}\boldsymbol{s} + \tfrac{1}{2}\sum_{i=1}^n ||\boldsymbol{z}_i||_2^2 + \sum_{i<j} \boldsymbol{h}_i^T(A_{\alpha,\beta}+A_{\alpha,\beta}^T)\boldsymbol{h}_j = \boldsymbol{h}_i^T A_{\alpha,\beta}\boldsymbol{s} + \tfrac{1}{2}\sum_{i=1}^n ||\boldsymbol{z}_i||_2^2 + \sum_{i<j} \boldsymbol{h}_i^T Q_{\alpha,\beta}\boldsymbol{h}_j \\
&= \sum_i \boldsymbol{h}_i^T A_{\alpha,\beta}\boldsymbol{s} + \tfrac{1}{2}\sum_{i=1}^n ||\boldsymbol{z}_i||_2^2 + \sum_{i<j} \boldsymbol{z}_i^T \boldsymbol{z}_j
\end{aligned}
$$

Importantly, since all strategy vectors $\boldsymbol{h}_i$ are positive and $\boldsymbol{s}$ is positive, we can state the following for any two joint strategies $\boldsymbol{H}^*$ and $\boldsymbol{H}$:

$$
c_{total}(\boldsymbol{H}^*,\boldsymbol{\lambda}) \geq \sum_i \boldsymbol{h}_i^{*T}A_{\alpha,\beta}\boldsymbol{s} + \tfrac{1}{2}\sum_{i=1}^n ||\boldsymbol{z}_i^*||_2^2 \quad\text{and}\quad c_{total}(\boldsymbol{H},\boldsymbol{\lambda}) \geq \tfrac{1}{2}\sum_{i=1}^n ||\boldsymbol{z}_i||_2^2 \tag{18}
$$

As for the (LHS), we observe the following:

$$
\begin{aligned}
\text{LHS} &= \sum_i \boldsymbol{h}_i^{*T}A_{\alpha,\beta}\boldsymbol{s} + \tfrac{1}{2}\sum_{i=1}^n \boldsymbol{h}_i^{*T}Q_{\alpha,\beta}\boldsymbol{h}_i^* + \sum_{i=1}\sum_{i\neq j} \boldsymbol{h}_i^{*T}A_{\alpha,\beta}\boldsymbol{h}_j \\
&= \sum_i \boldsymbol{h}_i^{*T}A_{\alpha,\beta}\boldsymbol{s} + \tfrac{1}{2}\sum_{i=1}^n ||\boldsymbol{z}_i^*||^2 + \sum_{i=1}\sum_{j\neq i} \boldsymbol{h}_i^{*T}(A_{\alpha,\beta}^s + A_{\alpha,\beta}^a)\boldsymbol{h}_j \\
&= \sum_i \boldsymbol{h}_i^{*T}A_{\alpha,\beta}\boldsymbol{s} + \tfrac{1}{2}\sum_{i=1}^n ||\boldsymbol{z}_i^*||^2 + \tfrac{1}{2}\sum_{i=1}^n\sum_{j\neq i} \boldsymbol{h}_i^{*T}Q_{\alpha,\beta}\boldsymbol{h}_j + \sum_{i=1}\sum_{j\neq i} \boldsymbol{z}_i^{*T}Q_{\alpha,\beta}^{-1/2}A_{\alpha,\beta}^a Q_{\alpha,\beta}^{-1/2}\boldsymbol{z}_j \\
&= \sum_i \boldsymbol{h}_i^{*T}A_{\alpha,\beta}\boldsymbol{s} + \tfrac{1}{2}\sum_{i=1}^n ||\boldsymbol{z}_i^*||^2 + \tfrac{1}{2}\sum_{i=1}^n\sum_{j\neq i} \boldsymbol{z}_i^{*T}\boldsymbol{z}_j + \sum_{i=1}^n\sum_{i\neq j} \boldsymbol{z}_i^{*T}\tilde{A}_a \boldsymbol{z}_j \\
&\leq \sum_i \boldsymbol{h}_i^{*T}A_{\alpha,\beta}\boldsymbol{s} + \tfrac{1}{2}\sum_{i=1}^n ||\boldsymbol{z}_i^*||^2 + \tfrac{1}{2}\sum_{i=1}^n\sum_{j\neq i} ||\boldsymbol{z}_i^*||\cdot||\boldsymbol{z}_j|| + \sum_{i=1}^n\sum_{i\neq j} ||\boldsymbol{z}_i^*||\cdot||\boldsymbol{z}_j||\cdot\kappa_A \\
&\leq \sum_i \boldsymbol{h}_i^{*T}A_{\alpha,\beta}\boldsymbol{s} + \tfrac{1}{2}\sum_{i=1}^n ||\boldsymbol{z}_i^*||^2 + (\tfrac{1}{2}+\kappa_A)\sum_{i=1}^n\sum_{j\neq i} ||\boldsymbol{z}_i^*||\cdot||\boldsymbol{z}_j|| \\
&\leq \sum_i \boldsymbol{h}_i^{*T}A_{\alpha,\beta}\boldsymbol{s} + \tfrac{1}{2}\sum_{i=1}^n ||\boldsymbol{z}_i^*||^2 + (\tfrac{1}{2}+\kappa_A)\sum_{i=1}^n\sum_{j\neq i} \left(\tfrac{\varepsilon}{2}||\boldsymbol{z}_i^*||^2 + \tfrac{1}{2\varepsilon}||\boldsymbol{z}_j||^2\right) \quad\text{(due to Fact 1)} \\
&\leq \sum_i \boldsymbol{h}_i^{*T}A_{\alpha,\beta}\boldsymbol{s} + \tfrac{1}{2}\sum_{i=1}^n ||\boldsymbol{z}_i^*||^2 + (\tfrac{1}{2}+\kappa_A)\sum_{i=1}^n\sum_{j\neq i} \left(\tfrac{\varepsilon}{2}||\boldsymbol{z}_i^*||^2 + \tfrac{1}{2\varepsilon}||\boldsymbol{z}_j||^2\right) \\
&\leq \sum_i \boldsymbol{h}_i^{*T}A_{\alpha,\beta}\boldsymbol{s} + \tfrac{1}{2}\underbrace{[1+\varepsilon(0.5+\kappa_A)(n-1)]}_{\lambda>1}\sum_{i=1}^n ||\boldsymbol{z}_i^*||^2 + \tfrac{1}{2}\underbrace{\frac{(0.5+\kappa_A)(n-1)}{\varepsilon}}_{\mu}\sum_{i=1}^n ||\boldsymbol{z}_i||^2 \\
&\leq \lambda\left[\sum_i \boldsymbol{h}_i^{*T}A_{\alpha,\beta}\boldsymbol{s} + \tfrac{1}{2}\sum_{i=1}^n ||\boldsymbol{z}_i^*||^2\right] + \mu\tfrac{1}{2}\sum_{i=1}^n ||\boldsymbol{z}_i||^2 \leq \lambda c_{total}(\boldsymbol{H}^*) + \mu c_{total}(\boldsymbol{H}) = \text{RHS}
\end{aligned}
$$

We note that for smooth games, $\mu < 1$, and we thus need to set $(n-1)(0.5 + \kappa_A) < \varepsilon$. From Roughgarden (2015), we know that PoA in smooth games is upper bounded by $\frac{\lambda}{1-\mu}$. Letting $a = (0.5 + \kappa_A)$, we have that PoA as a function of $\varepsilon$ is:

$$\text{PoA}_\varepsilon = \frac{\varepsilon(1 + \varepsilon a(n-1))}{\varepsilon - a(n-1)} \tag{19}$$

By taking the derivative and solving for $\varepsilon$ to get the critical point, we have that :

$$\varepsilon^* = (n-1)a + \sqrt{(n-1)^2 a^2 + 1} \leq (n-1)a + 1 + \sqrt{(n-1)^2 a^2} \leq 2a(n-1) + 1.$$

Letting $b = a(n-1)$ – and thus $\varepsilon^* = 2b + 1$ – we can plug in this expression of $\varepsilon^*$ to our PoA. We get:

$$\text{PoA} = \frac{(2b+1)(1 + b(2b+1))}{2b + 1 - b} = \frac{(2b+1)(2b^2 + b + 1)}{(b+1)} = \frac{4b^3 + 4b^2 + 3b + 1}{b+1}$$

$$= 4b^2 + 3 - \frac{2}{b+1} \leq 4b^2 + 3 \leq 4a^2(n-1)^2 + 3 \leq (1 + 2\kappa_A)^2 (n-1)^2 + 3$$

Lastly, we note that: $||A^a_{\alpha,\beta}||_2 \leq \sqrt{||A^a_{\alpha,\beta}||_1 \cdot ||A^a_{\alpha,\beta}||_\infty} = (T-1)\frac{\alpha}{2}$. Then we have that:

$$\kappa_A = ||Q^{-1/2}_{\alpha,\beta} A_{\alpha,\beta} Q^{-1/2}_{\alpha,\beta}||_2 \leq ||Q^{-1/2}_{\alpha,\beta}||^2_2 \, ||A^a_{\alpha,\beta}||_2 \leq \frac{T-1}{2} \frac{\alpha}{\alpha + 2\beta} = \frac{T-1}{2}\gamma \tag{20}$$

Note that since $\kappa_A$ depends on the market parameters, which are sampled, we choose $\gamma = \sup \frac{\alpha}{\alpha+2\beta}$ (the supremum is over the support of $\alpha, \beta$ to obtain an upper bound that holds uniformly on all samples). Plugging everything in, we have that:

$$\text{PoA} \leq (1 + \gamma(T-1))^2 (n-1)^2 + 3 = O(n^2 T^2 \gamma^2) \tag{21}$$

$\square$

# B  Proofs and Details for Section 4

In what follows we primarily use the cost notation $c_i(\cdot)$ (recall that $c_i(\boldsymbol{h}_i(\theta_i); \boldsymbol{h}_{-i}(\theta_{-i}), \boldsymbol{\lambda}) = -u_i(\boldsymbol{h}_i(\theta_i); \boldsymbol{h}_{-i}(\theta_{-i}), \boldsymbol{\lambda})$). Unless otherwise specified, we use $\|\cdot\|$ throughout to denote the $\ell_2$ norm. We will often use the notion of a "population loss."

**Definition 8** (Population Loss). *Let $\boldsymbol{h}^r_i$ denote the strategy of agent $i$ in round $r$. Then, the expected loss for each agent $i$ in round $r$ is a function $\ell^r_i : \mathcal{H}_i \to \mathbb{R}$ given by:*

$$\ell^r_i(\boldsymbol{h}_i) = \mathbb{E}_{\mathcal{I} \sim P}[c^r_i(\boldsymbol{h}_i)] = \mathbb{E}_{\boldsymbol{\theta}, \boldsymbol{\lambda} \sim P}[-u_i(\boldsymbol{h}_i(\theta_i); \boldsymbol{h}^r_{-i}(\theta_{-i}), \boldsymbol{\lambda})].$$

**Lemma 4.** *For all $i \in [n]$, let $V_i(\boldsymbol{h}) = \nabla_{\boldsymbol{h}_i} \ell_i(\boldsymbol{h}_i, \boldsymbol{h}_{-i}; P) = \nabla_{\boldsymbol{h}_i} \mathbb{E}_{\boldsymbol{\theta}, \boldsymbol{\lambda} \sim P}[c_i(\boldsymbol{h}_i(\theta_i), \boldsymbol{h}_{-i}(\theta_{-i}), \boldsymbol{\lambda})] \in \mathbb{R}^{k_i \times T}$ and $V(\boldsymbol{h}) = (V_1(\boldsymbol{h}), ..., V_n(\boldsymbol{h})) \in \mathbb{R}^{n \times k \times T}$. The operator $V$ is $m$-strongly monotone, i.e. $\langle V(\boldsymbol{h}') - V(\boldsymbol{h}), \boldsymbol{h}' - \boldsymbol{h}\rangle \geq m\|\boldsymbol{h}' - \boldsymbol{h}\|^2$ for all $\boldsymbol{h}, \boldsymbol{h}'$, where $m$ is the strong monotonicity constant of Theorem 1. Consequently, for all $i$, $\ell_i(\boldsymbol{h}_i, \boldsymbol{h}_{-i}; P)$ is $m$-strongly convex in $\boldsymbol{h}_i$.*

*Proof.* Recall that Theorem 1 shows that the operator $W(\boldsymbol{h}) \in \mathbb{R}^{n \times k \times T}$, defined by:

$$W_i(\theta_i)(\boldsymbol{h}) = \Pr(\theta_i) \cdot \nabla_{\boldsymbol{h}_i(\theta_i)} \mathbb{E}_{\theta_{-i}, \boldsymbol{\lambda} \sim P|\theta_i}[c_i(\boldsymbol{h}_i(\theta_i), \boldsymbol{h}_{-i}(\theta_{-i}), \boldsymbol{\lambda})] \in \mathbb{R}^T$$

in the entry corresponding to agent $i$ and type $\theta_i$, is $m$-strongly monotone for some positive $m$.

Now, for every $i$, we can write:

$$V_i(\boldsymbol{h}) = \nabla_{\boldsymbol{h}_i}\left(\sum_{\theta_i} \Pr(\theta_i) \cdot \mathbb{E}_{\theta_{-i}, \boldsymbol{\lambda} \sim P|\theta_i}[c_i(\boldsymbol{h}_i(\theta_i), \boldsymbol{h}_{-i}(\theta_{-i}), \boldsymbol{\lambda})]\right)$$

Fix a agent $i$ and a type $\theta_i^*$. Since each agent has finitely many types, we can write $V$ as a vector of size $nkT$, where the entry of $V$ corresponding to agent $i$ and type $\theta_i^*$ is:

$$
\begin{aligned}
V_i(\theta_i^*)(\boldsymbol{h}) &= \nabla_{\boldsymbol{h}_i(\theta_i^*)} \left( \sum_{\theta_i} \Pr(\theta_i) \cdot \mathbb{E}_{\theta_{-i}, \boldsymbol{\lambda} \sim P|\theta_i} [c_i(\boldsymbol{h}_i(\theta_i), \boldsymbol{h}_{-i}(\theta_{-i}), \boldsymbol{\lambda})] \right) \\
&= \nabla_{\boldsymbol{h}_i(\theta_i^*)} \left( \Pr(\theta_i^*) \cdot \mathbb{E}_{\theta_{-i}, \boldsymbol{\lambda} \sim P|\theta_i^*} [c_i(\boldsymbol{h}_i(\theta_i^*), \boldsymbol{h}_{-i}(\theta_{-i}), \boldsymbol{\lambda})] \right) \\
&\qquad\qquad\qquad \text{(since all terms not involving } \theta_i^* \text{ can be treated as constants)} \\
&= \Pr(\theta_i^*) \cdot \nabla_{\boldsymbol{h}_i(\theta_i^*)} \mathbb{E}_{\theta_{-i}, \boldsymbol{\lambda} \sim P|\theta_i^*} [c_i(\boldsymbol{h}_i(\theta_i^*), \boldsymbol{h}_{-i}(\theta_{-i}), \boldsymbol{\lambda})] \\
&= W_i(\theta_i^*)(\boldsymbol{h})
\end{aligned}
$$

Thus $V = W$, and $V$ is $m$-strongly monotone, i.e. $\langle V(\boldsymbol{h}') - V(\boldsymbol{h}), \boldsymbol{h}' - \boldsymbol{h} \rangle \geq m\|\boldsymbol{h}' - \boldsymbol{h}\|^2$ for all $\boldsymbol{h}, \boldsymbol{h}'$. For every $i$, $m$-strong convexity then follows by definition, by considering $\boldsymbol{h}, \boldsymbol{h}'$ that are the same in all coordinates except $i$. $\qquad\square$

## B.1 Jordan et al. (2024) Algorithm Details

Here we present the algorithm of Jordan et al. (2024) and state its guarantees.

---

**Algorithm 2:** AdaOGD (Algorithm 1 of Jordan et al. (2024))

---

**Input:** Strategy space $\mathcal{H}_i$, cost function $\ell_i$

Initialize $\boldsymbol{h}_i^1 \in \mathcal{H}_i$

Let $z_0 = \frac{1}{\log(R+10)}$

**for** $r = 1, ..., R$ **do**

$\quad$ Sample $M^r \sim \text{Geometric}(z_0)$

$\quad$ Let $\eta^{r+1} = \frac{r+1}{\sqrt{1 + \max\{M^1, ..., M^r\}}}$

$\quad$ Update $\boldsymbol{h}_i^{r+1} = \arg\min_{\boldsymbol{h}_i \in \mathcal{H}_i} \{ (\boldsymbol{h}_i - \boldsymbol{h}_i^r)^\top \tilde{\nabla}_i^r + \frac{\eta^{r+1}}{2} \|\boldsymbol{h}_i - \boldsymbol{h}_i^r\|^2 \}$

---

**Theorem 5** (Theorem 3.7 of Jordan et al. (2024)). *Consider a game $\mathcal{G}$ among $n$ agents, each with a convex and bounded action set $\mathcal{H}_i \subseteq \mathbb{R}^{d_i}$ and a cost function $\ell_i : \prod_{i=1}^n \mathcal{H}_i \to \mathbb{R}$ satisfying: (i) $\ell_i(\boldsymbol{h}_i, \boldsymbol{h}_{-i})$ is continuous in $(\boldsymbol{h}_i, \boldsymbol{h}_{-i})$ and continuously differentiable in $\boldsymbol{h}_i$; (ii) $\nabla_{\boldsymbol{h}_i} \ell_i(\boldsymbol{h}_i, \boldsymbol{h}_{-i})$ is continuous in $(\boldsymbol{h}_i, \boldsymbol{h}_{-i})$; (iii) $\|\boldsymbol{h} - \boldsymbol{h}'\| \leq D$ for all $\boldsymbol{h}, \boldsymbol{h}' \in \prod_{i=1}^n \mathcal{H}_i$; and (iv) $\mathcal{G}$ is $m$-strongly monotone. Suppose at every round $r \in [R]$, each agent observes an unbiased and bounded gradient $\tilde{\nabla}_{\boldsymbol{h}_i^r}^r$ satisfying $\mathbb{E}[\tilde{\nabla}_{\boldsymbol{h}_i^r}^r | \boldsymbol{h}^r] = \nabla_{\boldsymbol{h}^r} \ell_i(\boldsymbol{h}_i^r, \boldsymbol{h}_{-i}^r)$ and $\mathbb{E}[\|\tilde{\nabla}_{\boldsymbol{h}_i}^r\|^2 | \boldsymbol{h}^r] \leq M$. Then, if all agents run Algorithm 2, the final iterate satisfies:*

$$
\mathbb{E}\left[\|\boldsymbol{h}^R - \boldsymbol{h}^*\|^2\right] \leq O\left( \frac{D^2 M (1 + \exp(1/(m^2 \log R))) \log(nR) \log^2(R)}{R} \right)
$$

*where $\boldsymbol{h}^*$ is the Nash equilibrium of $\mathcal{G}$, i.e. for all $i \in [n]$, for all $\boldsymbol{h}_i \in \mathcal{H}_i$, $\ell_i(\boldsymbol{h}_i^*, \boldsymbol{h}_{-i}^*) \leq \ell_i(\boldsymbol{h}_i, \boldsymbol{h}_{-i}^*)$.*

## B.2 Algorithm 2 Under Biased Gradient Observations

We first show how the guarantees of Algorithm 2 change under possibly biased gradient observations. To do so, we extend a key lemma in Jordan et al. (2024) to accommodate for biased gradients.

**Lemma 5** (Extended version of Lemma 3.10 in Jordan et al. (2024)). *Consider a game $\mathcal{G}$ among $n$ agents, each with a convex and bounded action set $\mathcal{H}_i \subseteq \mathbb{R}^{d_i}$ and a cost function $\ell_i : \prod_{i=1}^n \mathcal{H}_i \to \mathbb{R}$ satisfying $\|\boldsymbol{h} - \boldsymbol{h}'\| \leq D$ for all $\boldsymbol{h}, \boldsymbol{h}' \in \prod_{i=1}^n \mathcal{H}_i$ and $\mathcal{G}$ is $m$-strongly monotone. Suppose at every round $r \in [R]$, each agent observes a gradient estimate $\tilde{\nabla}_{\boldsymbol{h}_i^r}^r$ satisfying $\|\mathbb{E}[\tilde{\nabla}_{\boldsymbol{h}_i^r}^r | \boldsymbol{h}^r] - \nabla_{\boldsymbol{h}_i^r} \ell_i(\boldsymbol{h}_i^r, \boldsymbol{h}_{-i}^r)\| \leq \Delta$ and $\mathbb{E}[\|\tilde{\nabla}_{\boldsymbol{h}}^r\|^2 | \boldsymbol{h}^r] \leq M$.*

*Then, if all agents run Algorithm 2, letting $\boldsymbol{h}^\star$ denote the (unique) Nash equilibrium, we have*

$$\sum_{i=1}^n \eta_i^R \, \mathbb{E}\big[\|\boldsymbol{h}_i^R - \boldsymbol{h}_i^\star\|^2 \,\big|\, \{\eta_i^r\}_{i\in[n],\, r\in[R]}\big] \le \sum_{i=1}^n \eta_i^1 \|\boldsymbol{h}_i^1 - \boldsymbol{h}_i^\star\|^2 + D^2 \sum_{r=1}^{R-1} \max\Big\{0,\ \max_{i\in[n]}\big(\eta_i^{r+1} - \eta_i^r - 2m\big)\Big\} \quad (22)$$

$$+ M \sum_{r=1}^{R-1}\Big(\max_{i\in[n]} \frac{1}{\eta_i^{r+1}}\Big) + 2D\sqrt{n}\,\Delta\,(R-1).$$

*Proof.* We follow the proof of Lemma 3.10 in Jordan et al. (2024). Fix $r \in \{1, \dots, R-1\}$. We will condition throughout on the entire step-size realization $\{\eta_i^r\}_{i\in[n],\, r\in[R]}$ but suppress this conditioning in the notation to reduce clutter. Algorithm 2 performs, for each agent $i$, a projected gradient step of the form

$$\boldsymbol{h}_i^{r+1} \in \arg\min_{\boldsymbol{h}_i \in \mathcal{H}_i} \left\{ \left\langle \tilde{\nabla}_{\boldsymbol{h}_i^r}^r,\ \boldsymbol{h}_i - \boldsymbol{h}_i^r \right\rangle + \frac{\eta_i^{r+1}}{2} \|\boldsymbol{h}_i - \boldsymbol{h}_i^r\|^2 \right\}.$$

By the first-order optimality condition, for any $\boldsymbol{h}_i \in \mathcal{H}_i$,

$$\left\langle \tilde{\nabla}_{\boldsymbol{h}_i^r}^r,\ \boldsymbol{h}_i^{r+1} - \boldsymbol{h}_i \right\rangle \le \frac{\eta_i^{r+1}}{2}\Big( \|\boldsymbol{h}_i^r - \boldsymbol{h}_i\|^2 - \|\boldsymbol{h}_i^{r+1} - \boldsymbol{h}_i\|^2 - \|\boldsymbol{h}_i^{r+1} - \boldsymbol{h}_i^r\|^2 \Big).$$

Rearranging and using the identity $\langle \tilde{\nabla}_{\boldsymbol{h}_i^r}^r,\ \boldsymbol{h}_i^r - \boldsymbol{h}_i \rangle = \langle \tilde{\nabla}_{\boldsymbol{h}_i^r}^r,\ \boldsymbol{h}_i^r - \boldsymbol{h}_i^{r+1} \rangle + \langle \tilde{\nabla}_{\boldsymbol{h}_i^r}^r,\ \boldsymbol{h}_i^{r+1} - \boldsymbol{h}_i \rangle$, we obtain

$$\eta_i^{r+1}\big(\|\boldsymbol{h}_i^{r+1} - \boldsymbol{h}_i\|^2 - \|\boldsymbol{h}_i^r - \boldsymbol{h}_i\|^2\big) \le 2\left\langle \boldsymbol{h}_i - \boldsymbol{h}_i^r,\ \tilde{\nabla}_{\boldsymbol{h}_i^r}^r \right\rangle + \frac{1}{\eta_i^{r+1}}\|\tilde{\nabla}_{\boldsymbol{h}_i^r}^r\|^2 + (\eta_i^{r+1} - \eta_i^r)\|\boldsymbol{h}_i^r - \boldsymbol{h}_i\|^2, \quad (23)$$

where we used the standard inequality $2\langle \boldsymbol{h}_i^r - \boldsymbol{h}_i^{r+1},\ \tilde{\nabla}_{\boldsymbol{h}_i^r}^r \rangle - \eta_i^{r+1}\|\boldsymbol{h}_i^{r+1} - \boldsymbol{h}_i^r\|^2 \le \frac{1}{\eta_i^{r+1}}\|\tilde{\nabla}_{\boldsymbol{h}_i^r}^r\|^2$.

Now set $\boldsymbol{h}_i = \boldsymbol{h}_i^\star$ in equation 23 and sum over $i \in [n]$. Writing $\tilde{\nabla}_{\boldsymbol{h}^r}^r := (\tilde{\nabla}_{\boldsymbol{h}_1^r}^r, \dots, \tilde{\nabla}_{\boldsymbol{h}_n^r}^r)$ and using $\|\boldsymbol{h}^r - \boldsymbol{h}^\star\|^2 = \sum_i \|\boldsymbol{h}_i^r - \boldsymbol{h}_i^\star\|^2$, we get

$$\sum_{i=1}^n \eta_i^{r+1}\big(\|\boldsymbol{h}_i^{r+1} - \boldsymbol{h}_i^\star\|^2 - \|\boldsymbol{h}_i^r - \boldsymbol{h}_i^\star\|^2\big) \le 2\left\langle \boldsymbol{h}^\star - \boldsymbol{h}^r,\ \tilde{\nabla}_{\boldsymbol{h}^r}^r \right\rangle + \sum_{i=1}^n \frac{1}{\eta_i^{r+1}}\|\tilde{\nabla}_{\boldsymbol{h}_i^r}^r\|^2$$

$$+ \sum_{i=1}^n (\eta_i^{r+1} - \eta_i^r)\|\boldsymbol{h}_i^r - \boldsymbol{h}_i^\star\|^2. \quad (24)$$

Using $\sum_i \frac{1}{\eta_i^{r+1}}\|\tilde{\nabla}_{\boldsymbol{h}_i^r}^r\|^2 \le \big(\max_{i\in[n]} \frac{1}{\eta_i^{r+1}}\big) \sum_i \|\tilde{\nabla}_{\boldsymbol{h}_i^r}^r\|^2 = \big(\max_{i\in[n]} \frac{1}{\eta_i^{r+1}}\big) \|\tilde{\nabla}_{\boldsymbol{h}^r}^r\|^2$, and taking conditional expectation given $\boldsymbol{h}^r$ (and the step sizes), we obtain from equation 24

$$\sum_{i=1}^n \eta_i^{r+1} \, \mathbb{E}\big[\|\boldsymbol{h}_i^{r+1} - \boldsymbol{h}_i^\star\|^2 \,\big|\, \boldsymbol{h}^r\big] - \sum_{i=1}^n \eta_i^{r+1}\|\boldsymbol{h}_i^r - \boldsymbol{h}_i^\star\|^2 \le 2\left\langle \boldsymbol{h}^\star - \boldsymbol{h}^r,\ \mathbb{E}[\tilde{\nabla}_{\boldsymbol{h}^r}^r \mid \boldsymbol{h}^r] \right\rangle$$

$$+ \Big(\max_{i\in[n]} \frac{1}{\eta_i^{r+1}}\Big) \, \mathbb{E}\big[\|\tilde{\nabla}_{\boldsymbol{h}^r}^r\|^2 \mid \boldsymbol{h}^r\big]$$

$$+ \sum_{i=1}^n (\eta_i^{r+1} - \eta_i^r)\|\boldsymbol{h}_i^r - \boldsymbol{h}_i^\star\|^2. \quad (25)$$

Next, define the game operator $V(\boldsymbol{h}) := (\nabla_{\boldsymbol{h}_1}\ell_1(\boldsymbol{h}), \dots, \nabla_{\boldsymbol{h}_n}\ell_n(\boldsymbol{h}))$ and the conditional bias vector

$$b^r := \mathbb{E}[\tilde{\nabla}_{\boldsymbol{h}^r}^r \mid \boldsymbol{h}^r] - V(\boldsymbol{h}^r) = \big(\mathbb{E}[\tilde{\nabla}_{\boldsymbol{h}_1^r}^r \mid \boldsymbol{h}^r] - \nabla_{\boldsymbol{h}_1}\ell_1(\boldsymbol{h}^r),\ \dots,\ \mathbb{E}[\tilde{\nabla}_{\boldsymbol{h}_n^r}^r \mid \boldsymbol{h}^r] - \nabla_{\boldsymbol{h}_n}\ell_n(\boldsymbol{h}^r)\big).$$

By assumption, each block satisfies $\|b_i^r\| \le \Delta$, hence

$$\|b^r\|^2 = \sum_{i=1}^n \|b_i^r\|^2 \le n\Delta^2 \qquad \Rightarrow \qquad \|b^r\| \le \sqrt{n}\,\Delta. \quad (26)$$

Using this decomposition in the first inner product on the right-hand side of equation 25,

$$\langle \boldsymbol{h}^\star - \boldsymbol{h}^r, \, \mathbb{E}[\tilde{\nabla}_{\boldsymbol{h}^r}^r \mid \boldsymbol{h}^r] \rangle = \langle \boldsymbol{h}^\star - \boldsymbol{h}^r, \, V(\boldsymbol{h}^r) \rangle + \langle \boldsymbol{h}^\star - \boldsymbol{h}^r, \, b^r \rangle.$$

Since $\mathcal{G}$ is $m$-strongly monotone and $\boldsymbol{h}^\star$ is a Nash equilibrium, we have $V(\boldsymbol{h}^\star) = 0$ and therefore

$$\langle \boldsymbol{h}^r - \boldsymbol{h}^\star, \, V(\boldsymbol{h}^r) - V(\boldsymbol{h}^\star) \rangle \geq m \|\boldsymbol{h}^r - \boldsymbol{h}^\star\|^2 \qquad \Rightarrow \qquad \langle \boldsymbol{h}^\star - \boldsymbol{h}^r, \, V(\boldsymbol{h}^r) \rangle \leq -m \|\boldsymbol{h}^r - \boldsymbol{h}^\star\|^2.$$

Moreover, by Cauchy–Schwarz, the diameter bound $\|\boldsymbol{h}^r - \boldsymbol{h}^\star\| \leq D$, and equation 26,

$$\langle \boldsymbol{h}^\star - \boldsymbol{h}^r, \, b^r \rangle \leq \|\boldsymbol{h}^\star - \boldsymbol{h}^r\| \, \|b^r\| \leq D\sqrt{n}\,\Delta.$$

Combining the last three displays yields

$$\langle \boldsymbol{h}^\star - \boldsymbol{h}^r, \, \mathbb{E}[\tilde{\nabla}_{\boldsymbol{h}^r}^r \mid \boldsymbol{h}^r] \rangle \leq -m \|\boldsymbol{h}^r - \boldsymbol{h}^\star\|^2 + D\sqrt{n}\,\Delta. \tag{27}$$

Plugging equation 27 and $\mathbb{E}[\|\tilde{\nabla}_{\boldsymbol{h}^r}^r\|^2 \mid \boldsymbol{h}^r] \leq M$ into equation 25, taking expectation over $\boldsymbol{h}^r$, and using $\|\boldsymbol{h}^r - \boldsymbol{h}^\star\|^2 \leq D^2$, we obtain

$$\sum_{i=1}^n \eta_i^{r+1} \mathbb{E}\big[\|\boldsymbol{h}_i^{r+1} - \boldsymbol{h}_i^\star\|^2\big] - \sum_{i=1}^n \eta_i^r \mathbb{E}\big[\|\boldsymbol{h}_i^r - \boldsymbol{h}_i^\star\|^2\big] \leq \sum_{i=1}^n (\eta_i^{r+1} - \eta_i^r - 2m) \mathbb{E}\big[\|\boldsymbol{h}_i^r - \boldsymbol{h}_i^\star\|^2\big]$$
$$+ M\left(\max_{i\in[n]} \frac{1}{\eta_i^{r+1}}\right) + 2D\sqrt{n}\,\Delta. \tag{28}$$

Moreover,

$$\sum_{i=1}^n (\eta_i^{r+1} - \eta_i^r - 2m) \mathbb{E}\big[\|\boldsymbol{h}_i^r - \boldsymbol{h}_i^\star\|^2\big] \leq D^2 \max\Big\{0, \, \max_{i\in[n]} \big(\eta_i^{r+1} - \eta_i^r - 2m\big)\Big\}.$$

Substituting this into equation 28 and summing over $r = 1, \ldots, R-1$ yields the following, completing the proof:

$$\sum_{i=1}^n \eta_i^R \mathbb{E}\big[\|\boldsymbol{h}_i^R - \boldsymbol{h}_i^\star\|^2\big] \leq \sum_{i=1}^n \eta_i^1 \|\boldsymbol{h}_i^1 - \boldsymbol{h}_i^\star\|^2 + D^2 \sum_{r=1}^{R-1} \max\Big\{0, \, \max_{i\in[n]} \big(\eta_i^{r+1} - \eta_i^r - 2m\big)\Big\}$$
$$+ M \sum_{r=1}^{R-1} \left(\max_{i\in[n]} \frac{1}{\eta_i^{r+1}}\right) + 2D\sqrt{n}\,\Delta\,(R-1)$$

$\square$

We can now state the guarantees of Algorithm 2 under biased gradient observations.

**Theorem 6** (Extended version of Theorem 3.7 in Jordan et al. (2024)). *Consider a game $\mathcal{G}$ among $n$ agents, each with a convex and bounded action set $\mathcal{H}_i \subseteq \mathbb{R}^{d_i}$ and a cost function $\ell_i : \prod_{i=1}^n \mathcal{H}_i \to \mathbb{R}$ satisfying: (i) $\ell_i(\boldsymbol{h}_i, \boldsymbol{h}_{-i})$ is continuous in $(\boldsymbol{h}_i, \boldsymbol{h}_{-i})$ and continuously differentiable in $\boldsymbol{h}_i$; (ii) $\nabla_{\boldsymbol{h}_i}\ell_i(\boldsymbol{h}_i, \boldsymbol{h}_{-i})$ is continuous in $(\boldsymbol{h}_i, \boldsymbol{h}_{-i})$; (iii) $\|\boldsymbol{h} - \boldsymbol{h}'\| \leq D$ for all $\boldsymbol{h}, \boldsymbol{h}' \in \prod_{i=1}^n \mathcal{H}_i$; and (iv) $\mathcal{G}$ is $m$-strongly monotone. Suppose that at every round $r \in [R]$, each agent $i$ observes a gradient estimate $\tilde{\nabla}_{\boldsymbol{h}_i^r}^r$ satisfying $\|\mathbb{E}[\tilde{\nabla}_{\boldsymbol{h}_i^r}^r \mid \boldsymbol{h}^r] - \nabla_{\boldsymbol{h}_i^r}\ell_i(\boldsymbol{h}_i^r, \boldsymbol{h}_{-i}^r)\| \leq \Delta$ and $\mathbb{E}[\|\tilde{\nabla}_{\boldsymbol{h}}^r\|^2 \mid \boldsymbol{h}^r] \leq M$. Then, letting $\boldsymbol{h}^\star$ denote the (unique) Nash equilibrium of $\mathcal{G}$, if all agents run Algorithm 2 for $R$ rounds, the final iterate satisfies*

$$\mathbb{E}\big[\|\boldsymbol{h}^R - \boldsymbol{h}^\star\|^2\big] \leq O\left(\frac{D^2 M\big(1 + \exp(1/(m^2 \log R))\big)\,\log(nR)\,\log^2(R)}{R} + \left(1 - \frac{1}{R}\right)\sqrt{n}\Delta \log(nR)\log(R)\right)$$

*Proof.* The proof closely follows the proof of Theorem 3.7 in Jordan et al. (2024). Since $\|\boldsymbol{h} - \boldsymbol{h}'\| \leq D$ for all $\boldsymbol{h}, \boldsymbol{h}' \in \mathcal{H}$ and for all $i$, $\eta_i^{r+1} = \frac{r+1}{\sqrt{1+\max\{M_i^1, \ldots M_i^r\}}}$ as in Algorithm 2, we have:

$$\sum_{i=1}^n \eta_i^1 \|\boldsymbol{h}_i^1 - \boldsymbol{h}_i^\star\|^2 \leq D^2$$

and:

$$\eta_i^{r+1} - \eta_i^r \le \frac{1}{\sqrt{1 + \max\{M_i^1, \dots M_i^r\}}}$$

Therefore, by Lemma 5:

$$\sum_{i=1}^n \eta_i^R \mathbb{E}\big[\|\boldsymbol{h}_i^R - \boldsymbol{h}_i^\star\|^2 \,\big|\, \{\eta_i^r\}_{i,r}\big] \le D^2 + D^2 \sum_{r=1}^{R-1} \max\left\{0, \max_{1\le i\le n}\Big(\frac{1}{\sqrt{1+\max\{M_i^1, \dots M_i^r\}}}\Big) - 2m\right\}$$
$$+ M \sum_{r=1}^{R-1} \max_{1\le i\le n}\left\{\frac{1}{\eta_i^{r+1}}\right\} + 2D\sqrt{n}\,\Delta(R-1). \tag{29}$$

Next, the definition of $\eta_i^{r+1}$ implies:

$$\sum_{r=1}^{R-1} \max_{1\le i\le n}\left\{\frac{1}{\eta_i^{r+1}}\right\} \le \sqrt{1 + \max_{1\le i\le n, 1\le r\le R} M_i^r} \sum_{r=1}^{R-1} \frac{1}{r+1} \le \log(R+1)\sqrt{1 + \max_{1\le i\le n, 1\le r\le R} M_i^r}. \tag{30}$$

Plugging equation 30 into equation 29 yields:

$$\sum_{i=1}^n \eta_i^R \mathbb{E}\big[\|\boldsymbol{h}_i^R - \boldsymbol{h}_i^\star\|^2 \,\big|\, \{\eta_i^r\}_{i,r}\big] \le D^2 + D^2 \sum_{r=1}^{R-1} \max\left\{0, \max_{1\le i\le n}\Big(\frac{1}{\sqrt{1+\max\{M_i^1, \dots M_i^r\}}}\Big) - 2m\right\}$$
$$+ M\log(R+1)\sqrt{1 + \max_{1\le i\le n, 1\le r\le R} M_i^r} + 2D\sqrt{n}\,\Delta(R-1). \tag{31}$$

Moreover, by the definition of $\eta_i^R$:

$$\eta_i^R \ge \frac{R}{\sqrt{1 + \max\{M_i^1, \dots, M_i^R\}}} \ge \frac{R}{\sqrt{1 + \max_{1\le j\le n, 1\le r\le R} M_j^r}},$$

which implies:

$$\sum_{i=1}^n \eta_i^R \mathbb{E}\big[\|\boldsymbol{h}_i^R - \boldsymbol{h}_i^\star\|^2 \,\big|\, \{\eta_i^r\}_{i,r}\big] \ge \frac{R}{\sqrt{1 + \max_{1\le j\le n, 1\le r\le R} M_j^r}}\, \mathbb{E}\big[\|\boldsymbol{h}^R - \boldsymbol{h}^\star\|^2 \,\big|\, \{\eta_i^r\}_{i,r}\big].$$

Combining this with equation 31 and multiplying both sides by $\sqrt{1 + \max_{j,r} M_j^r}$ gives:

$$R\,\mathbb{E}\big[\|\boldsymbol{h}^R - \boldsymbol{h}^\star\|^2 \,\big|\, \{\eta_i^r\}_{i,r}\big] \le D^2\sqrt{1 + \max_{j,r} M_j^r} + M\log(R+1)\big(1 + \max_{j,r} M_j^r\big)$$
$$+ D^2\sqrt{1 + \max_{j,r} M_j^r} \sum_{r=1}^{R-1} \max\left\{0, \max_{1\le i\le n}\Big(\frac{1}{\sqrt{1+\max\{M_i^1, \dots M_i^r\}}}\Big) - 2m\right\}$$
$$+ 2D\sqrt{n}\,\Delta(R-1)\sqrt{1 + \max_{j,r} M_j^r}. \tag{32}$$

Taking expectations of both sides, define the three terms $I, II, III$ as:

$$I := \mathbb{E}\left[\sqrt{1 + \max_{j,r} M_j^r}\right], \qquad II := \mathbb{E}\left[1 + \max_{j,r} M_j^r\right],$$

and

$$III := \mathbb{E}\left[\sum_{r=1}^{R-1} \max\left\{0, \max_{1\le i\le n}\Big(\eta_i^{r+1} - \eta_i^r - 2m\Big)\right\}\sqrt{1 + \max_{j,r} M_j^r}\right].$$

Then equation 32 becomes

$$R\,\mathbb{E}\big[\|\boldsymbol{h}^R - \boldsymbol{h}^\star\|^2\big] \le D^2 I + M\log(R+1)\,II + D^2\,III + 2D\sqrt{n}\,\Delta(R-1)\,I. \tag{33}$$

It remains to bound the terms $I, II$, and $III$. This step is identical to the argument in Jordan et al. (2024), and so we defer the details to Jordan et al. (2024). The result then follows from plugging the respective bounds into equation 33. $\qquad\square$

### B.3    Proof of Theorem 4

To prove Theorem 4, our approach will be to run Algorithm 2 on suitable discretizations of the type space. Hence, we first establish the following theorem, which bounds last-iterate convergence to BNE in Bayesian game instances where agents have *finitely many* types.

**Theorem 7.** *Suppose for all agents $i$, $|\Theta_i| < \infty$. Suppose all agents use Algorithm 2 to decide their strategy in each round $r$, where at every round $r$, the end-of-round feedback is as specified in Assumption 2. Then, letting $k = \max_{i \in [n]} |\Theta_i|$, the final iterate strategies $\boldsymbol{h}_1^R, \ldots, \boldsymbol{h}_n^R$ are an $\epsilon$-approximate BNE, for some $\epsilon$ that satisfies the following in expectation over the algorithm's randomness:*

$$\mathbb{E}[\epsilon'] = O\left(\text{poly}(n, T, p_0, B, S, U', \alpha_{\max}, \beta_{\max}, 1/(\kappa\hat{\kappa})) \cdot \left(\sqrt{k}\frac{\log^{3/2}(R)}{\sqrt{R}} + (\Delta_\alpha + \Delta_\beta)\left(1 - \frac{1}{R}\right)\log^2(R)\right)\right)$$

*where $S, U', \kappa, \hat{\kappa}$ are constants such that $\sup_{\boldsymbol{s}} \|\boldsymbol{s}\| \leq S$, $\forall i \ \sup_{h_i \in \mathcal{H}_i} \|\nabla f_i(\boldsymbol{h}_i(\theta_i))\| \leq U'$, $\forall r \in [R] \ \hat{\alpha}_i^r \geq \kappa, \hat{\beta}_i^r \geq \hat{\kappa}$.*

*Proof.* We define the game $\mathcal{G}$ where each agent $i$ chooses a strategy map $\boldsymbol{h}_i \in \mathcal{H}_i$ and suffers cost:

$$\ell_i(\boldsymbol{h}_i, \boldsymbol{h}_{-i}; P) = \mathbb{E}_{\theta, \boldsymbol{\lambda} \sim P}[c_i(\boldsymbol{h}_i(\theta_i), \boldsymbol{h}_{-i}(\theta_{-i}), \boldsymbol{\lambda})]$$

We verify the conditions of Theorem 5 on this game $\mathcal{G}$. Since for all type profiles $\theta$, $c_i(\boldsymbol{h}_i(\theta_i), \boldsymbol{h}_{-i}(\theta_{-i}), \boldsymbol{\lambda})$ is continuous in $(\boldsymbol{h}_i(\theta_i), \boldsymbol{h}_{-i}(\theta_{-i}))$ and continuously differentiable in $\boldsymbol{h}_i(\theta_i)$, we have that $\ell_i(\boldsymbol{h}_i, \boldsymbol{h}_{-i}; P)$ is continuous in $(\boldsymbol{h}_i, \boldsymbol{h}_{-i})$ and continuously differentiable in $\boldsymbol{h}_i$. Under $(B, L)$ regularity, $\|\boldsymbol{h} - \boldsymbol{h}'\| \leq B\sqrt{nk}$ for all $\boldsymbol{h}, \boldsymbol{h}'$. Furthermore, by Lemma 4, $\mathcal{G}$ is $m$-strongly monotone, where $m$ is the strong monotonicity of Theorem 1.

To apply Theorem 5, it remains to establish the required conditions on agents' gradient feedback. Recall at each round $r \in [R]$, agent $i$ receives as feedback: $\tilde{\nabla}_{\boldsymbol{h}_i}^r = \nabla_{\boldsymbol{h}_i^r} \hat{c}_i(\boldsymbol{h}_i^r(\theta_i^r); \{p_t\}, \hat{\alpha}_i^r, \hat{\beta}_i^r)$. We first show that the feedback $\tilde{\nabla}_{\boldsymbol{h}_i}^r$ is close in expectation to the true population gradient feedback.

To do so, we formally derive the expression for the gradient. We first write the change in price as an expression of the true total aggregate demand. Fix a round $r \in [R]$ and agent $i \in [n]$. Fix $\alpha, \beta$. For each type profile $\theta$, we denote the external aggregate demand at time $t \in [T]$ by $d_{-i}^r(\theta) = \sum_{j \neq i} \boldsymbol{h}_j^r(\theta_j^r) - \boldsymbol{s}^r$, so that the true total aggregate demand is: $D_t^r(\theta) = h_{i,t}^r(\theta_i^r) + d_{-i,t}^r(\theta^r)$. Recall that $p_t^r = p_t^{w,r} + \beta D_t^r(\theta)$ and $p_t^{w,r} - p_{t-1}^{w,r} = \alpha D_t^r(\theta)$. Let $\Delta p_t^r := p_t^r - p_{t-1}^r$ denote the observed price differences. Combining these equations, we get the following recursive expression:

$$\Delta p_t^r = (\alpha + \beta) D_t^r(\theta) - \beta D_{t-1}^r(\theta)$$

Using this, we now derive an expression for the *estimated* total aggregate demand based on the realized prices $\{p_t^r\}_{t=0}^T$ and estimates $\hat{\alpha}_i^r, \hat{\beta}_i^r$ of $\alpha, \beta$ of the true market impact parameters the agent receives at the end of round $r$. Rewriting the above expression in terms of $\hat{\alpha}_i^r, \hat{\beta}_i^r$ and rearranging, agent $i$ constructs an estimated total aggregate demand sequence, where $\hat{D}_{i,0}^r := 0$ and for all $t = 1, \ldots, T$:

$$\hat{D}_{i,t}^r := \frac{\Delta p_t^r + \hat{\beta}_i^r \hat{D}_{i,t-1}^r}{\hat{\alpha}_i^r + \hat{\beta}_i^r}$$

Finally, define the estimate of the external aggregate demand at time $t$ as:

$$\hat{d}_{-i,t}^r := \hat{D}_{i,t}^r - h_{i,t}^r(\theta_i^r)$$

which subtracts the agent's own realized demand trajectory. Thus, the estimated gradient can be written:

$$\tilde{\nabla}_{\boldsymbol{h}_i}^r = p_0 \boldsymbol{1}_T + \hat{\alpha}_i^r W \boldsymbol{h}_i(\theta_i^r) + \hat{\alpha}_i^r W' \hat{d}_{-i}^r + \hat{\beta}_i^r (2\boldsymbol{h}_i(\theta_i) + \hat{d}_{-i}^r) - \nabla f_i(\boldsymbol{h}_i(\theta_i))$$

while the true gradient can be written as:

$$\nabla_{\boldsymbol{h}_i} c_i\big(\boldsymbol{h}_i(\theta_i), \boldsymbol{h}^r_{-i}(\theta_{-i}), \boldsymbol{\lambda}^r\big) = p_0 \mathbf{1}_T + \alpha^r\, W\, \boldsymbol{h}_i(\theta_i) + \alpha^r\, W'\, d^r_{-i}(\theta) + \beta^r\big(2\boldsymbol{h}_i(\theta_i) + d^r_{-i}(\theta)\big) - \nabla f_i\big(\boldsymbol{h}_i(\theta_i)\big).$$

where $W \in \mathbb{R}^{T \times T}$ is the matrix with $W_{tt} = 2$ for all $t \in [T]$ and $1$ everywhere else, and $W' \in \mathbb{R}^{T \times T}$ is the matrix with $W'_{ts} = 1$ for $s \leq t$ and $0$ everywhere else.

Since $\theta^r, \boldsymbol{\lambda}^r \sim P$ are sampled independently from the strategies chosen at round $r$, we have that $\mathbb{E}[\tilde{\nabla}^r_{\boldsymbol{h}_i}|\boldsymbol{h}^r] = \mathbb{E}[\tilde{\nabla}^r_{\boldsymbol{h}_i}]$ and thus:

$$\begin{aligned}
&\big\|\mathbb{E}[\tilde{\nabla}^r_{\boldsymbol{h}_i}|\boldsymbol{h}^r] - \nabla_{h_i}\ell_i(\boldsymbol{h}_i, \boldsymbol{h}^r_{-i})\big\| \\
&= \big\|\mathbb{E}[\tilde{\nabla}^r_{\boldsymbol{h}_i}] - \nabla_{h_i}\ell_i(\boldsymbol{h}_i, \boldsymbol{h}^r_{-i})\big\| \\
&= \big\|\mathbb{E}_{\theta^r, \boldsymbol{\lambda}^r \sim P}[\tilde{\nabla}^r_{\boldsymbol{h}_i}] - \nabla_{\boldsymbol{h}_i}\mathbb{E}_{\theta, \boldsymbol{\lambda} \sim P}\big[c_i\big(\boldsymbol{h}_i(\theta_i), \boldsymbol{h}^r_{-i}(\theta_{-i}), \boldsymbol{\lambda}\big)\big]\big\| \\
&= \Big\|\mathbb{E}_{\theta^r, \boldsymbol{\lambda}^r \sim P}\Big[\big(\hat{\alpha}^r_i - \alpha^r\big)\, W\, \boldsymbol{h}_i(\theta_i) + \big(\hat{\alpha}^r_i - \alpha^r\big)\, W'\, d^r_{-i}(\theta) + \big(\hat{\beta}^r_i - \beta^r\big)\big(2\boldsymbol{h}_i(\theta_i) + d^r_{-i}(\theta)\big) \\
&\qquad\qquad\quad + \hat{\alpha}^r_i\, W'\big(\hat{d}^r_{-i} - d^r_{-i}(\theta)\big) + \hat{\beta}^r_i\big(\hat{d}^r_{-i} - d^r_{-i}(\theta)\big)\Big]\Big\| \\
&\leq \underbrace{\Big\|\mathbb{E}_{\theta^r, \boldsymbol{\lambda}^r \sim P}\Big[\big(\hat{\alpha}^r_i - \alpha^r\big)\big(W\, \boldsymbol{h}_i(\theta_i) + W'\, d^r_{-i}(\theta)\big) + \big(\hat{\beta}^r_i - \beta^r\big)\big(2\boldsymbol{h}_i(\theta_i) + d^r_{-i}(\theta)\big)\Big]\Big\|}_{=:\, T_1} \\
&\quad + \underbrace{\Big\|\mathbb{E}_{\theta^r, \boldsymbol{\lambda}^r \sim P}\Big[\hat{\alpha}^r_i\, W'\big(\hat{d}^r_{-i} - d^r_{-i}(\theta)\big) + \hat{\beta}^r_i\big(\hat{d}^r_{-i} - d^r_{-i}(\theta)\big)\Big]\Big\|}_{=:\, T_2} \qquad\text{(by triangle inequality)}
\end{aligned}$$

We can bound $T1$ as:

$$\begin{aligned}
T1 &\leq (TB + T((n-1)B + S))\,\mathbb{E}_{\theta^r, \boldsymbol{\lambda}^r \sim P}[|\hat{\alpha}^r_i - \alpha^r|] + ((n+1)B + S)\mathbb{E}_{\theta^r, \boldsymbol{\lambda}^r \sim P}[|\hat{\beta}^r_i - \beta|] \\
&\qquad\qquad\qquad\qquad\qquad\text{(by triangle inequality and boundedness assumptions)} \\
&\leq \Delta_\alpha\,(TB + T((n-1)B + S)) + \Delta_\beta((n+1)B + S) \qquad\text{(by Assumption 2)}
\end{aligned}$$

Next we bound $T2$, which is the error in the gradient arising from the estimated aggregate demand trajectory. For $t = 0, ..., T$, let the per-time reconstruction error $e_t := \hat{D}^r_{i,t} - D^r_t(\theta)$. Then, for each $t = 1, ..., T$, we have:

$$\begin{aligned}
e_t &= \frac{\Delta p^r_t + \hat{\beta}^r_i \hat{D}^r_{i,t-1}}{\hat{\alpha}^r_i + \hat{\beta}^r_i} - \frac{\Delta p^r_t + \beta^r D^r_{t-1}(\theta)}{\alpha^r + \beta^r} \\
&\qquad\qquad\qquad\text{(plugging in the recursive equation for total aggregate demand)} \\
&= \frac{\hat{\beta}^r_i}{\hat{\alpha}^r_i + \hat{\beta}^r_i}\big(\hat{D}^r_{i,t-1} - D^r_{t-1}(\theta)\big) + \Big(\frac{1}{\hat{\alpha}^r_i + \hat{\beta}^r_i} - \frac{1}{\alpha^r + \beta^r}\Big)\Delta p^r_t + \Big(\frac{\hat{\beta}^r_i}{\hat{\alpha}^r_i + \hat{\beta}^r_i} - \frac{\beta}{\alpha^r + \beta^r}\Big)D^r_{t-1}(\theta) \\
&= \frac{\hat{\beta}^r_i}{\hat{\alpha}^r_i + \hat{\beta}^r_i}\, e_{t-1} + \Big(\frac{1}{\hat{\alpha}^r_i + \hat{\beta}^r_i} - \frac{1}{\alpha^r + \beta^r}\Big)\Delta p^r_t + \Big(\frac{\hat{\beta}^r_i}{\hat{\alpha}^r_i + \hat{\beta}^r_i} - \frac{\beta}{\alpha^r + \beta^r}\Big)D^r_{t-1}(\theta)
\end{aligned}$$

To bound this expression, let $s := \alpha^r + \beta^r \geq \kappa > 0$ and $\hat{s} := \hat{\alpha}^r_i + \hat{\beta}^r_i \geq \hat{\kappa} > 0$. Let $D_{\max} := \sup_{i,r,t} |D^r_t(\theta)| \leq nB + S$ by our boundedness assumptions. Then, using the price recursion given by the true aggregate demand $\Delta p^r_t = sD^r_t(\theta) - \beta D^r_{t-1}(\theta)$ and $s \leq \alpha_{\max} + \beta_{\max}$, $\beta \leq \beta_{\max}$, we can bound:

$$|\Delta p^r_t| \leq (s + \beta)D_{\max} \leq (\alpha_{\max} + 2\beta_{\max})D_{\max} =: P_{\max}.$$

Next:

$$\Big|\frac{1}{\hat{s}} - \frac{1}{s}\Big| = \frac{|\hat{s} - s|}{\hat{s}\, s} \leq \frac{2}{\kappa\hat{\kappa}}|\hat{s} - s| \leq \frac{2}{\kappa\hat{\kappa}}\big(|\hat{\alpha}^r_i - \alpha| + |\hat{\beta}^r_i - \beta|\big)$$

and:

$$\left|\frac{\hat{\beta}_i^r}{\hat{s}} - \frac{\beta}{s}\right| = \left|\frac{\hat{\beta}_i^r - \beta}{\hat{s}} + \beta\left(\frac{1}{\hat{s}} - \frac{1}{s}\right)\right|$$

$$\leq \frac{2}{\kappa}|\hat{\beta}_i^r - \beta| + \beta_{\max} \cdot \frac{2}{\kappa\hat{\kappa}}\left(|\hat{\alpha}_i^r - \alpha| + |\hat{\beta}_i^r - \beta|\right).$$

Thus:

$$|e_t| \leq |e_{t-1}| + \left|\frac{1}{\hat{s}} - \frac{1}{s}\right| \cdot |\Delta p_t^r| + \left|\frac{\hat{\beta}_i^r}{\hat{s}} - \frac{\beta}{s}\right| \cdot |D_{t-1}^r(\theta)|$$

$$\leq |e_{t-1}| + \left(\frac{2P_{\max}}{\kappa\hat{\kappa}}\right)\left(|\hat{\alpha}_i^r - \alpha| + |\hat{\beta}_i^r - \beta|\right) + \left(\frac{2D_{\max}}{\kappa\hat{\kappa}}\right)|\hat{\beta}_i^r - \beta| + \left(\frac{2\beta_{\max}D_{\max}}{\kappa\hat{\kappa}}\right)\left(|\hat{\alpha}_i^r - \alpha| + |\hat{\beta}_i^r - \beta|\right)$$

Taking expectations and using $\mathbb{E}[|\hat{\alpha}_i^r - \alpha|] \leq \Delta_\alpha$ and $\mathbb{E}[|\hat{\beta}_i^r - \beta|] \leq \Delta_\beta$ gives

$$\mathbb{E}[|e_t|] \leq \mathbb{E}[|e_{t-1}|] + C_e\left(\Delta_\alpha + \Delta_\beta\right)$$

where $C_e = \text{poly}(n, B, S, \alpha_{max}, \beta_{max}, 1/\kappa\hat{\kappa})$. Since $e_0 = \hat{D}_{i,0}^r - D_0^r(\theta) = 0$ by definition, iterating gives us:

$$\max_{t\in[T]} \mathbb{E}[|e_t|] \leq T\,C_e(\Delta_\alpha + \Delta_\beta).$$

Therefore, we have that the deviation of the estimated external aggregate demand from the true external aggregate demand is:

$$\mathbb{E}\left[\|\hat{d}_{-i}^r - d_{-i}^r(\theta)\|\right] = \mathbb{E}\left[\|e\|\right] \leq \sqrt{T}\max_{t\in[T]}\mathbb{E}[|e_t|] \leq \sqrt{T}\left(T\,C_e(\Delta_\alpha + \Delta_\beta)\right).$$

and we can bound $T2$ as:

$$T_2 \leq \alpha_{\max}\mathbb{E}\left[\|W'(\hat{d}_{-i}^r - d_{-i}^r(\theta))\|\right] + \beta_{\max}\mathbb{E}\left[\|\hat{d}_{-i}^r - d_{-i}^r(\theta)\|\right]$$

$$\leq (\alpha_{\max}T + \beta_{\max})\mathbb{E}\left[\|\hat{d}_{-i}^r - d_{-i}^r(\theta)\|\right]$$

$$\leq (\alpha_{\max}T + \beta_{\max})\sqrt{T}\left(T\,C_e(\Delta_\alpha + \Delta_\beta)\right)$$

Combining the bounds on $T1$ and $T2$, the bias of the gradient can be bounded by $\Delta :=$ $\Delta_\alpha\left(TB + T((n-1)B + S)\right) + \Delta_\beta((n+1)B + S) + (\alpha_{\max}T + \beta_{\max})\sqrt{T}(TC_e(\Delta_\alpha + \Delta_\beta)) \leq (\Delta_\alpha + \Delta_b) \cdot$ $\text{poly}(n, T, B, S, \alpha_{max}, \beta_{max}, 1/(\kappa\hat{\kappa}))$.

We now move to bounding the norm of the gradient feedback. From the recursive reconstruction expression of the estimated demand, we have that for all $t = 1, ..., T$:

$$|\hat{D}_{i,t}^r| = \left|\frac{\Delta p_t^r + \hat{\beta}_i^r\,\hat{D}_{i,t-1}^r}{\hat{s}}\right| \leq \frac{|\Delta p_t^r|}{\hat{\kappa}} + \frac{\hat{\beta}_i^r}{\hat{\kappa}}|\hat{D}_{i,t-1}^r| \leq \frac{P_{\max}}{\hat{\kappa}} + \frac{\beta_{\max}}{\hat{\kappa}}|\hat{D}_{i,t-1}^r| \leq \frac{P_{\max}}{\hat{\kappa}} \cdot t \leq \frac{T\,P_{\max}}{\hat{\kappa}}$$

and therefore

$$\|\hat{D}_i^r\| \leq \sqrt{T}\max_{t\in[T]}|\hat{D}_{i,t}^r| \leq \frac{T^{3/2}P_{\max}}{\hat{\kappa}}.$$

Since $\|\hat{d}_{-i}^r\| \leq \|\hat{D}_i^r\| + \|\boldsymbol{h}_i^r(\theta_i^r)\|$, it follows that:

$$\|\hat{d}_{-i}^r\|_2 \leq \frac{T^{3/2}P_{\max}}{\hat{\kappa}} + \sqrt{T}B$$

So, we can derive:

$$\|\tilde{\nabla}_{h_i}^r\| \leq \|p_0\mathbf{1}_T\| + \hat{\alpha}_i^r\|W\boldsymbol{h}_i^r(\theta_i^r)\| + \hat{\alpha}_i^r\|W'\hat{d}_{-i}^r\| + \hat{\beta}_i^r\|2\boldsymbol{h}_i^r(\theta_i^r) + \hat{d}_{-i}^r\| + \|\nabla f_i(\boldsymbol{h}_i^r(\theta_i^r))\|$$

$$\leq |p_0|\sqrt{T} + \alpha_{\max}T\|\boldsymbol{h}_i^r(\theta_i^r)\| + \alpha_{\max}T\|\hat{d}_{-i}^r\| + \beta_{\max}\left(2\|\boldsymbol{h}_i^r(\theta_i^r)\| + \|\hat{d}_{-i}^r\|\right) + U'$$

$$\leq |p_0|\sqrt{T} + \alpha_{\max}T^{3/2}B + \alpha_{\max}T\left(\frac{T^{3/2}P_{\max}}{\hat{\kappa}} + \sqrt{T}\,B\right) + \beta_{\max}\left(2\sqrt{T}\,B + \frac{T^{3/2}P_{\max}}{\hat{\kappa}} + \sqrt{T}\,B\right) + U'$$

$$\leq \text{poly}(n, T, p_0, B, S, U', \alpha_{\max}, \beta_{\max}, 1/\hat{\kappa})$$

In particular, taking expectations yields $\mathbb{E}[\|\tilde{\nabla}_{h_i}^r\|^2|h^r]\mathbb{E}[\|\tilde{\nabla}_{h_i}^r\|^2] \leq \mathrm{poly}(n, T, p_0, B, S, U', \alpha_{\max}, \beta_{\max}, 1/\hat{\kappa})$.

Therefore, by Theorem 5, the final iterate produced by Algorithm 2 satisfies:

$$\mathbb{E}\left[\|h^R - h^*\|^2\right] \leq O\left(\frac{D^2 M(1 + \exp(1/m^2 \log R))\log(nR)\log^2(R)}{R} + \left(1 - \frac{1}{R}\right)\sqrt{n}\Delta\log(nR)\log(R)\right)$$

where $h^*$ is the Nash equilibrium of $\mathcal{G}$, and the expectation is taken over the randomness of the algorithm. Here, we have $D \leq B\sqrt{nk}$ and $M \leq \mathrm{poly}(n, T, p_0, B, S, U', \alpha_{\max}, \beta_{\max}, 1/\hat{\kappa})$.

Next we show that $\ell_i$ is Lipschitz in the $\ell_2$ norm, which will allow us to argue that since $h^R$ and $h^*$ are close in $\ell_2$ distance, they must also incur similar cost. Observe that for any $j \neq i$, for any $h_i, h_{-i}, \theta, \lambda$:

$$\|\nabla_{h_j}c_i(h_i(\theta_i), h_{-i}(\theta_{-i}), \lambda)\| = \|\alpha M^\top h_i(\theta_i) + \beta h_i(\theta_i)\| \leq (\alpha T + \beta)B$$

and so:

$$\sup_h \|\nabla_h c_i(h_i(\theta_i), h_{-i}(\theta_{-i}), \lambda)\|$$
$$\leq \underbrace{|p_0|\sqrt{T} + \alpha T B + \alpha T((n-1)B + S) + \beta((n+1)B + S) + U' + n(\alpha T + \beta)B}_{=:L'(p_0, \alpha, \beta)}$$

Therefore for all $h, h', \theta, \lambda$, $c_i$ is $L'(p_0, \alpha, \beta)$-Lipschitz in $h$, i.e.:

$$|c_i(h_i'(\theta_i), h_{-i}'(\theta_{-i}), \lambda) - c_i(h_i(\theta_i), h_{-i}(\theta_{-i}), \lambda)| \leq L'\|h' - h\|$$

Taking expectations, we have that $\ell_i$ is $L$-Lipschitz in $h$, i.e. for all $h, h'$:

$$\begin{aligned}|\ell_i(h_i', h_{-i}'; P) - \ell_i(h_i, h_{-i}; P)| &= |\mathbb{E}_{\theta, \lambda \sim P}[c_i(h_i'(\theta_i), h_{-i}'(\theta_{-i}), \lambda) - c_i(h_i(\theta_i), h_{-i}(\theta_{-i}), \lambda)]| \\ &\leq \mathbb{E}_{\theta, \lambda \sim P}[|c_i(h_i'(\theta_i), h_{-i}'(\theta_{-i}), \lambda) - c_i(h_i(\theta_i), h_{-i}(\theta_{-i}), \lambda)|] \\ &\leq L\|h' - h\|\end{aligned}$$

where $L \leq \max_{p_0, \alpha, \beta} L'(p_0, \alpha, \beta) = \mathrm{poly}(n, T, \alpha, \beta, p_0, B, S, U')$.

Thus, the cost evaluated at $h^R$ is close to the cost evaluated at $h^*$: $\mathbb{E}\left[\ell_i(h_i^*, h_{-i}^*; P) - \ell_i(h_i^R, h_{-i}^R; P)\right]$

$$\leq L \cdot \mathbb{E}\left[\|h^R - h^*\|\right] \hspace{3cm} \text{(by } L\text{-Lipschitzness)}$$
$$\leq L \cdot O\left(\sqrt{\frac{D^2 M(1 + \exp(1/m^2 \log R))\log(nR)\log^2(R)}{R}} + \left(1 - \frac{1}{R}\right)\sqrt{n}\Delta\log(nR)\log(R)\right)$$

In the second inequality, we use that fact that $\mathbb{E}\left[\|h^R - h^*\|\right]^2 \leq \mathbb{E}\left[\|h^R - h^*\|^2\right]$ by Jensen's inequality. Furthermore, since the entries of $\|h^R - h^*\|$ are non-negative, we also have that for any $h_i \in \mathcal{H}_i$:

$$\begin{aligned}\mathbb{E}\left[\ell_i(h_i, h_{-i}^R; P) - \ell_i(h_i, h_{-i}^*; P)\right] &\leq L \cdot \mathbb{E}\left[\|h_{-i}^R - h_{-i}^*\|\right] \hspace{1.5cm} \text{(by } L\text{-Lipschitzness)} \\ &\leq L \cdot \mathbb{E}\left[\|h^R - h^*\|\right] \\ &\leq L \cdot O\left(\sqrt{\frac{D^2 M(1 + \exp(1/m^2 \log R))\log(nR)\log^2(R)}{R}} + \left(1 - \frac{1}{R}\right)\sqrt{n}\Delta\log(nR)\log(R)\right)\end{aligned}$$

Combining the above, we can show that $\boldsymbol{h}^R$ is an approximate Nash equilibrium of $\mathcal{G}$. In particular, for any $i$, for any $\boldsymbol{h}_i \in \mathcal{H}_i$: $\mathbb{E}\left[\ell_i(\boldsymbol{h}_i^R, \boldsymbol{h}_{-i}^R; P) - \ell_i(\boldsymbol{h}_i, \boldsymbol{h}_{-i}^R; P)\right]$

$$\leq \mathbb{E}\left[\ell_i(\boldsymbol{h}_i^*, \boldsymbol{h}_{-i}^*; P) - \ell_i(\boldsymbol{h}_i, \boldsymbol{h}_{-i}^R; P)\right] +$$
$$L \cdot O\left(\sqrt{\frac{D^2 M(1 + \exp(1/m^2 \log R))\log(nR)\log^2(R)}{R}} + \left(1 - \frac{1}{R}\right)\sqrt{n}\Delta \log(nR)\log(R)\right)$$

$$\leq \mathbb{E}\left[\ell_i(\boldsymbol{h}_i^*, \boldsymbol{h}_{-i}^*; P) - \ell_i(\boldsymbol{h}_i, \boldsymbol{h}_{-i}^*; P)\right] +$$
$$2L \cdot O\left(\sqrt{\frac{D^2 M(1 + \exp(1/m^2 \log R))\log(nR)\log^2(R)}{R}} + \left(1 - \frac{1}{R}\right)\sqrt{n}\Delta \log(nR)\log(R)\right)$$

$$\leq 2L \cdot O\left(\sqrt{\frac{D^2 M(1 + \exp(1/m^2 \log R))\log(nR)\log^2(R)}{R}} + \left(1 - \frac{1}{R}\right)\sqrt{n}\Delta \log(nR)\log(R)\right)$$

(by the fact that $\boldsymbol{h}^*$ is a Nash equilibrium)

Thus, we can conclude that $\boldsymbol{h}^R$ is an approximate Bayesian Nash equilibrium. Specifically, for all $i$ and $\boldsymbol{h}_i$:

$$\mathbb{E}[\mathbb{E}_{\theta, \boldsymbol{\lambda} \sim P}[c_i(\boldsymbol{h}_i^R(\theta_i), \boldsymbol{h}_{-i}^R(\theta_{-i}), \boldsymbol{\lambda}) - c_i(\boldsymbol{h}_i(\theta_i), \boldsymbol{h}_{-i}^R(\theta_{-i}), \boldsymbol{\lambda})]]$$
$$\leq 2L \cdot O\left(\sqrt{\frac{D^2 M(1 + \exp(1/m^2 \log R))\log(nR)\log^2(R)}{R}} + \left(1 - \frac{1}{R}\right)\sqrt{n}\Delta \log(nR)\log(R)\right)$$
$$\leq O\left(\text{poly}(n, T, p_0, B, S, U', \alpha_{\max}, \beta_{\max}, 1/(\kappa\hat{\kappa})) \cdot \left(\sqrt{k}\frac{\log^{3/2}(R)}{\sqrt{R}} + (\Delta_\alpha + \Delta_\beta)\left(1 - \frac{1}{R}\right)\log^2(R)\right)\right)$$

$\square$

*Proof of Theorem 4.* We begin by defining a "discretized game" obtained by discretizing types over a covering of $\Theta_i$. Let $\delta = 1/\log R$. For each agent $i$, let $\hat{\Theta}_i$ be a $\delta$-net of $\Theta_i$. That is, for any $\theta_i \in \Theta_i$, there exists $\hat{\theta}_i \in \hat{\Theta}_i$ satisfying $\|\hat{\theta}_i - \theta_i\|_\infty \leq \delta$. For any $\theta_i \in \Theta_i$, we will write $\hat{\theta}_i$ to denote its nearest neighbor in $\hat{\Theta}_i$. Define $\hat{P}$ as the joint distribution over game types and discretized player types that is the pushforward of the distribution $P$ under the transformation $(\theta_1, ..., \theta_n, \lambda) \mapsto (\hat{\theta}_1, ..., \hat{\theta}_1, \lambda)$. For every agent $i$, let $\hat{\mathcal{H}}_i = \{\hat{\boldsymbol{h}}_i : \Theta_i \to \mathbb{R}^T | \hat{\boldsymbol{h}}_i(\theta_i) = \boldsymbol{h}_i(\hat{\theta}_i) \,\forall \boldsymbol{h}_i \in \mathcal{H}_i\}$ be the set of strategies that is piecewise constant over each "bin" induced by the discretization. Observe that we can represent every $\hat{\boldsymbol{h}}_i \in \mathbb{R}^{k_i \times T}$, as a $k_i \times T$-dimensional matrix, where $k_i = |\hat{\Theta}_i| \leq (1/\delta)^{m_i}$ (we can always obtain a $\delta$-net by discretizing each coordinate to multiples of $\delta$). Let $k = \max_i k_i$.

Now, Algorithm 1 runs Algorithm 2 over this discretized game. To invoke its guarantees, we first verify that the strategy space $\hat{\mathcal{H}}_i$ is convex. Indeed, observe that for any $\zeta \in [0, 1]$ and any $\hat{\boldsymbol{h}}_i, \hat{\boldsymbol{h}}_i' \in \hat{\mathcal{H}}_i$, $(1 - \zeta)\hat{\boldsymbol{h}}_i + \zeta\hat{\boldsymbol{h}}_i' \in G_i$, since $G_i$ is itself convex. Also, any convex combination of piecewise constant functions is also convex. Finally, since $\hat{\boldsymbol{h}}_i, \hat{\boldsymbol{h}}_i'$ are $L$-Lipschitz over $\hat{\Theta}_i$, for any $\hat{\theta}_i, \hat{\theta}_i'$:

$$\|(1 - \zeta)\hat{\boldsymbol{h}}_i(\hat{\theta}_i) + \zeta\hat{\boldsymbol{h}}_i'(\hat{\theta}_i) - (1 - \zeta)\hat{\boldsymbol{h}}_i(\hat{\theta}_i') - \zeta\hat{\boldsymbol{h}}_i'(\hat{\theta}_i')\|_\infty$$
$$\leq (1 - \zeta)\|\hat{\boldsymbol{h}}_i(\hat{\theta}_i) - \hat{\boldsymbol{h}}_i(\hat{\theta}_i')\|_\infty + \zeta\|\hat{\boldsymbol{h}}_i'(\hat{\theta}_i) - \hat{\boldsymbol{h}}_i'(\hat{\theta}_i')\|_\infty$$
$$\leq (1 - \zeta)L\|\hat{\theta}_i - \hat{\theta}_i'\|_\infty + \zeta L\|\hat{\theta}_i - \hat{\theta}_i'\|_\infty = L\|\hat{\theta}_i - \hat{\theta}_i'\|_\infty$$

Thus $(1 - \zeta)\hat{\boldsymbol{h}}_i + \zeta\hat{\boldsymbol{h}}_i' \in \hat{\mathcal{H}}_i$, proving convexity.

Therefore, we can invoke Theorem 7, which guarantees that the final iterates are approximate ex-ante BNE. Let $\hat{\boldsymbol{h}}_1^R, \ldots, \hat{\boldsymbol{h}}_n^R$ be the final iterate strategies produced by all agents running Algorithm 1. By Theorem 7,

$\hat{\boldsymbol{h}}_1^R, \ldots, \hat{\boldsymbol{h}}_n^R$ satisfy for all $i$, all $\hat{\boldsymbol{h}}_i' \in \hat{\mathcal{H}}_i$:

$$\mathbb{E}_{\hat{\theta}, \boldsymbol{\lambda} \sim \hat{P}}[c_i(\hat{\boldsymbol{h}}_i^R(\hat{\theta}_i); \hat{\boldsymbol{h}}_{-i}^R(\hat{\theta}_{-i}), \boldsymbol{\lambda})] \le \mathbb{E}_{\hat{\theta}, \boldsymbol{\lambda} \sim \hat{P}}[c_i(\hat{\boldsymbol{h}}_i'(\hat{\theta}_i); \hat{\boldsymbol{h}}_{-i}^R(\hat{\theta}_{-i}), \boldsymbol{\lambda})] + \epsilon' \tag{34}$$

where $\mathbb{E}[\epsilon'] = O\left(\text{poly}(n, T, p_0, B, S, U', \alpha_{\max}, \beta_{\max}, 1/(\kappa\hat{\kappa})) \cdot \left(\sqrt{k}\frac{\log^{3/2}(R)}{\sqrt{R}} + (\Delta_\alpha + \Delta_\beta)\left(1 - \frac{1}{R}\right)\log^2(R)\right)\right)$.
Note that this equilibrium guarantee is stated over the distribution $\hat{P}$, the underlying distribution induced by the discretized game.

We now lift this guarantee to an ex-ante BNE guarantee in the original, continuous-type game. Recall that for all agents $i$ and all $\hat{\theta}_i \in \hat{\Theta}_i$, $\hat{\boldsymbol{h}}_i$ is piecewise constant over all $\theta_i \in \Theta_i$ in the same bin as $\hat{\theta}_i$ — that is, $\hat{\boldsymbol{h}}_i(\theta_i) = \hat{\boldsymbol{h}}_i(\hat{\theta}_i)$ for all $\theta_i$ such that $\hat{\theta}_i$ is its nearest neighbor in $\hat{\Theta}_i$. Thus, since $\hat{P}$ is the pushforward distribution of $P$ obtained by discretizing all $\theta_i$ to $\hat{\theta}_i$, we have that, for the LHS of Eq. 34:

$$\mathbb{E}_{\hat{\theta}, \boldsymbol{\lambda} \sim \hat{P}}[c_i(\hat{\boldsymbol{h}}_i^R(\hat{\theta}_i); \hat{\boldsymbol{h}}_{-i}^R(\hat{\theta}_{-i}), \boldsymbol{\lambda})] = \mathbb{E}_{\theta, \boldsymbol{\lambda} \sim P}[c_i(\hat{\boldsymbol{h}}_i^R(\theta_i); \hat{\boldsymbol{h}}_{-i}^R(\theta_{-i}), \boldsymbol{\lambda})]$$

Similarly, for the RHS:

$$\mathbb{E}_{\hat{\theta}, \boldsymbol{\lambda} \sim \hat{P}}[c_i(\hat{\boldsymbol{h}}_i'(\hat{\theta}_i); \hat{\boldsymbol{h}}_{-i}^R(\hat{\theta}_{-i}), \boldsymbol{\lambda})] = \mathbb{E}_{\theta, \boldsymbol{\lambda} \sim P}[c_i(\hat{\boldsymbol{h}}_i'(\theta_i); \hat{\boldsymbol{h}}_{-i}^R(\theta_{-i}), \boldsymbol{\lambda})]$$

Thus, Eq. 34 guarantees that for any agent $i$, their final iterate strategy is $\epsilon$-optimal against any deviation to an alternative strategy in $\hat{\mathcal{H}}_i$. It remains to compare their costs under any deviation to an alternative strategy in $\mathcal{H}_i$.

We show that for all agents $i$, for any $\boldsymbol{h}_i \in \mathcal{H}_i$, there exists $\hat{\boldsymbol{h}}_i \in \hat{\mathcal{H}}_i$ such that their outputs on any $\theta_i$ are close, and thus agent $i$'s costs under the two strategies are close. Fixing a $\boldsymbol{h}_i \in \mathcal{H}_i$, define $\hat{\boldsymbol{h}}_i$: $\hat{\boldsymbol{h}}_i(\theta_i) = \boldsymbol{h}_i(\hat{\theta}_i)$. Then, by $L$-Lipschitzness, for all $\theta_i \in \Theta_i$:

$$\|\boldsymbol{h}_i(\theta_i) - \hat{\boldsymbol{h}}_i(\theta_i)\|_\infty = \|\boldsymbol{h}_i(\theta_i) - \boldsymbol{h}_i(\hat{\theta}_i)\|_\infty \le L\|\theta_i - \hat{\theta}_i\|_\infty \le L\delta$$

We then establish that $c_i$ is Lipschitz so that under these two strategies, the costs are similar. We can derive, for any $\theta, \boldsymbol{\lambda}$:

$$\sup_{\boldsymbol{h}_i} \|\nabla_{\boldsymbol{h}_i} c_i(\boldsymbol{h}_i(\theta_i), \boldsymbol{h}_{-i}(\theta_{-i}), \boldsymbol{\lambda})\| \le \underbrace{p_0\sqrt{T} + \alpha T B + \alpha T((n-1)B + S) + \beta((n+1)B + S) + U'}_{=:L_c(p_0, \alpha, \beta)}$$

Using $L_c(p_0, \alpha, \beta)$-Lipschitzness of $c_i$, we have that for all $\theta, \boldsymbol{\lambda}$:

$$|c_i(\boldsymbol{h}_i(\theta_i); \hat{\boldsymbol{h}}_{-i}^R(\theta_{-i}), \boldsymbol{\lambda}) - c_i(\hat{\boldsymbol{h}}_i(\theta_i); \hat{\boldsymbol{h}}_{-i}^R(\theta_{-i}), \boldsymbol{\lambda})|$$
$$\le L_c(p_0, \alpha, \beta)\|\boldsymbol{h}_i(\theta_i) - \hat{\boldsymbol{h}}_i(\theta_i)\|$$
$$\le L_c(p_0, \alpha, \beta)\|\boldsymbol{h}_i(\theta_i) - \hat{\boldsymbol{h}}_i(\theta_i)\|$$
$$\le L_c(p_0, \alpha, \beta)\sqrt{T}\|\boldsymbol{h}_i(\theta_i) - \hat{\boldsymbol{h}}_i(\theta_i)\|_\infty$$
$$\le L_c(p_0, \alpha, \beta)L\sqrt{T}\delta$$

Taking expectations, we have:

$$|\mathbb{E}_{\theta, \boldsymbol{\lambda} \sim P}[c_i(\boldsymbol{h}_i(\theta_i); \hat{\boldsymbol{h}}_{-i}^R(\theta_{-i}), \boldsymbol{\lambda})] - \mathbb{E}_{\theta, \boldsymbol{\lambda} \sim P}[c_i(\hat{\boldsymbol{h}}_i(\theta_i); \hat{\boldsymbol{h}}_{-i}^R(\theta_{-i}), \boldsymbol{\lambda})]| \le L_c L \sqrt{T}\delta$$

where $L_c = \max_{\boldsymbol{\lambda}} L_c(p_0, \alpha, \beta) = \text{poly}(n, T, \alpha, \beta, p_0, B, S, U')$.

Therefore, any deviation to a strategy in $\mathcal{H}_i$ cannot increase agent $i$'s cost by more than $L_c L \delta$. That is:

$$\forall \hat{\boldsymbol{h}}_i' \in \hat{\mathcal{H}}_i : \quad \mathbb{E}_{\theta, \boldsymbol{\lambda} \sim P}[c_i(\hat{\boldsymbol{h}}_i^R(\theta_i); \hat{\boldsymbol{h}}_{-i}^R(\theta_{-i}), \boldsymbol{\lambda})] \le \mathbb{E}_{\theta, \boldsymbol{\lambda} \sim P}[c_i(\hat{\boldsymbol{h}}_i'(\theta_i); \hat{\boldsymbol{h}}_{-i}^R(\theta_{-i}), \boldsymbol{\lambda})] + \epsilon$$
$$\implies \forall \boldsymbol{h}_i' \in \mathcal{H}_i : \quad \mathbb{E}_{\theta, \boldsymbol{\lambda} \sim P}[c_i(\hat{\boldsymbol{h}}_i^R(\theta_i); \hat{\boldsymbol{h}}_{-i}^R(\theta_{-i}), \boldsymbol{\lambda})] \le \mathbb{E}_{\theta, \boldsymbol{\lambda} \sim P}[c_i(\boldsymbol{h}_i'(\theta_i); \hat{\boldsymbol{h}}_{-i}^R(\theta_{-i}), \boldsymbol{\lambda})] + L_c L \sqrt{T}\delta + \epsilon$$

Thus we can conclude that the final iterate strategies are an $(L_c L\delta + \epsilon')$-approximate ex-ante BNE, where $L_c L\sqrt{T}\delta + \mathbb{E}[\epsilon']$

$$
= L\delta \cdot \text{poly}(n, T, \alpha, \beta, p_0, B, S, U')
$$

$$
+ O\left(\text{poly}(n, T, p_0, B, S, U', \alpha_{\max}, \beta_{\max}, \tfrac{1}{\kappa\hat{\kappa}}) \cdot \left(\sqrt{k}\frac{\log^{3/2}(R)}{\sqrt{R}} + (\Delta_\alpha + \Delta_\beta)\left(1 - \frac{1}{R}\right)\log^2(R)\right)\right)
$$

$$
= L\delta \cdot \text{poly}(n, T, \alpha, \beta, p_0, B, S, U')
$$

$$
+ O\left(\text{poly}(n, T, p_0, B, S, U', \alpha_{\max}, \beta_{\max}, \tfrac{1}{\kappa\hat{\kappa}}) \cdot \left((1/\delta)^{m/2}\frac{\log^{3/2}(R)}{\sqrt{R}} + (\Delta_\alpha + \Delta_\beta)\left(1 - \frac{1}{R}\right)\log^2(R)\right)\right)
$$

$$
= O\left(\frac{L \cdot \text{poly}(n, T, \alpha, \beta, p_0, B, S, U')}{\log R}\right.
$$

$$
+ \text{poly}(n, T, p_0, B, S, U', \alpha_{\max}, \beta_{\max}, \tfrac{1}{\kappa\hat{\kappa}}) \cdot \left(\frac{\log^{(m+3)/2}(R)}{\sqrt{R}} + (\Delta_\alpha + \Delta_\beta)\left(1 - \frac{1}{R}\right)\log^2(R)\right)
$$

$$
= O\left(\text{poly}(n, T, p_0, B, S, U', \alpha_{\max}, \beta_{\max}, \tfrac{1}{\kappa\hat{\kappa}}) \cdot \left(\frac{L}{\log R} + \frac{\log^{(m+3)/2}(R)}{\sqrt{R}} + (\Delta_\alpha + \Delta_\beta)\left(1 - \frac{1}{R}\right)\log^2(R)\right)\right)
$$

where the second-to-last step follows from our choice of $\delta$. $\qquad\square$

## C  Additional Discussion on the Model and its Practical Implications

### C.1  Derivation of Price Model

First, we discuss in more detail how our price dynamics in Assumption 1 relate to Walrasian price dynamics. As mentioned in Section 2, this model positions that (mean) prices evolve according to the continuous time differential equation

$$
dp_t = \alpha(\text{demand}_t - \text{supply}_t)dt,
$$

for some price-sensitivity factor $\alpha$. Given this, the dynamic model for $p_t^w$ can be viewed as a discretization of this process. In addition, allowing for noise in $s_t$, this turns it into a discretization of the corresponding stochastic differential equation

$$
dp_t = \alpha(\text{demand}_t - \text{supply}_t)dt + \sigma dW_t,
$$

for some noise process $dW_t$ (*e.g.* Brownian motion). The actual price that traders must pay differs from this Walrasian process by amount $\beta(\sum_{i=1}^n h_{i,t} + s_t)$. We can interpret this difference as a temporary (instantaneous) price adjustment from $p_t^w$ driven by the imbalance of supply and demand; when supply and demand are not balanced, the difference must be met by *market makers*, who provide liquidity. These market makers require some spread from $p_t^w$ in order to account for the risk they take by providing liquidity. For example, if demand outstrips supply at time $t$, the market makers will balance this by selling an equal amount at a slight premium; hence, the instantaneous market price $p_t$ will be slightly higher than $p_t^w$. We implicitly assume as part of Assumption 1 that this difference is linear in the imbalance $\sum_{i=1}^n h_{i,t} + s_t$, with coefficient $\beta$.

Alternatively, this model can also be justified from the literature on market impact. For example, the seminal model of Almgren & Chriss (2000), which is the basis for much of the classical theory on (non-strategic) optimal trade execution, posits almost exactly the same model for price impact from trade execution over time, except that they consider a slightly more general offset based on supply and demand imbalance of the kind $\psi(\sum_{i=1}^n h_{i,t} + s_t)$, for some concave function $\psi : \mathbb{R}^+ \to \mathbb{R}^+$. Therefore, our price model is equivalent to theirs in the case of $\psi(x) = \beta x$. We note that empirical research (see *e.g.* Almgren et al. (2005)) suggests power-law models of the kind $\psi(x) = \beta x^\gamma$ with $\gamma \approx 3/5$ to be well supported by real data. Such a model

would be more challenging to study under strategic interaction, as it could break strong monotonicity without some additional assumptions on $f_i$ and/or $G_i$; we leave the investigation of alternative price models like this to future work.

## C.2  Relation to Existing Models in the Literature

Here, we make some concrete comparisons of our model with the models used in the recent lines of work on position building under competition Chriss (2024b;c;a; 2025); Kearns & Shi (2025).

**Relation to Existing Discrete Time Model**  First, consider the special case of our model with no idiosyncratic utilities ($f_i = 0$ for all $i$), and no exogenous actions ($\boldsymbol{s} = 0$). In this case, we can re-formulate the objective for each agent as minimizing a cost function $c_i(\boldsymbol{h}_i, \boldsymbol{h}_{-i})$ given by the negative of the utility $u_i$, which if we unroll the autoregressive price definitions like in the proof of Lemma 2, we can easily verify is given by

$$c_i(\boldsymbol{h}_i; \boldsymbol{h}_{-i}) = \alpha \sum_{t=1}^{T} h_{i,t} \sum_{l=1}^{t} \sum_{j=1}^{n} h_{j,l} + \beta \sum_{t=1}^{T} h_{i,t} \sum_{j=1}^{n} h_{j,t} + \sum_{t=1}^{T} h_{i,t} p_0$$

Now, assume further that the constraints $G_i$ contains a constraint of the kind $\sum_{t=1}^{T} h_{i,t} = V_i$ for some fixed $V_i$. Then, the third term above can be ignored, as it is always equal to $p_0 V_i$ for any feasible $\boldsymbol{h}_i \in G_i$. Given this, and with some slight re-arranging of terms, we have that the cost structure is given by

$$c_i(\boldsymbol{h}_i; \boldsymbol{h}_{-i}) = \alpha \sum_{t=1}^{T} h_{i,t} \sum_{j=1}^{n} x_{j,t-1} + (\alpha + \beta) \sum_{t=1}^{T} h_{i,t} \sum_{j=1}^{n} h_{j,t} \, ,$$

where we define $x_{i,t} = \sum_{l=1}^{t} h_{i,l}$ as the cumulative position acquired by agent $i$ over the first $t$ time steps. This corresponds exactly to the kind of cost structure assumed in Kearns & Shi (2025), who considered a discrete time version of optimal position building, with cost function

$$c_i^{\mathrm{KS}}(\boldsymbol{h}_i; \boldsymbol{h}_{-i}) = \kappa \sum_{t=1}^{T} h_{i,t} \sum_{j=1}^{n} x_{j,t-1} + \sum_{t=1}^{T} h_{i,t} \sum_{j=1}^{n} h_{j,t} \, .$$

Following terminology for literature on optimal position building, they denote first term as the *permanent-impact cost*, and the second term as the *temporary-impact cost*, with permanent-impact coefficient $\kappa$, and unit temporary-impact coefficient (which is completely general up to normalization of cost). Therefore, if we normalize our cost by $\alpha + \beta$, we see that under the above model restrictions it recovers theirs with $\kappa = \alpha/(\alpha + \beta)$.

Although our model may seem less general given the above reduction, as they allow for any $\kappa \geq 0$ but ours only allows $\kappa \in [0, 1]$, we argue that this restriction does not have much or any material impact in practice. First, as discussed in Kearns & Shi (2025), if they decompose their cost into zero-sum and potential (*i.e.* congestion game-style cost) components, the coefficient in front of potential cost becomes negative when $\kappa > 2$. This implies that agents are rewarded rather than punished from congestion of their trading schedule, which therefore encourages agents to behave as aggressively as their constraints will allow (this is reflected *e.g.* in the unstable dynamics they observe when agents play no-regret with $\kappa > 2$). Given this, we would probably wish to restrict to $\kappa < 2$ in such a discrete model in practice. Second, and perhaps more importantly, to the extent that their model is justified as a discretization of the continuous time model discussed below, the convergence of this discretization as we make it more and more fine-grained only works if we let the ratio of temporary-impact-coefficient to permanent-impact-coefficient (*i.e.* $\kappa$) tend towards zero as $T \to \infty$. Therefore, no matter the target $\kappa$ value in the continuous-time cost $c_i^{\mathrm{NC}}$ defined below, the corresponding $\kappa$ in the discrete-time cost $c_i^{\mathrm{KS}}$ that approximates this will be less than 1 if the discretization is sufficiently fine-grained.

**Relation to Existing Continuous Time Model** The works by Chriss (2024b;c;a; 2025) consider a continuous-time version of this problem, where the strategies $\boldsymbol{h}_i$ are functions over some continuous time range (which they normalize to be $[0,1]$ without loss of generality). In this setting, we assume the strategies are defined by functions $\boldsymbol{h}_i : [0,1] \to \mathbb{R}$, where $\boldsymbol{h}_i(t)$ is their instantaneous trading rate at time $t$. We also define $\boldsymbol{x}_i$ implicitly in terms of $\boldsymbol{h}_i$ as the total accumulated position up to time $t$, which is mathematically given by: $\boldsymbol{x}_i(t) = \int_0^t \boldsymbol{h}_i(l)dl$. Then, the assumed cost structure is:

$$c_i^{\text{NC}}(\boldsymbol{h}_i; \boldsymbol{h}_{-i}) = \kappa \int_0^1 \boldsymbol{h}_i(t) \sum_{j=1}^n \boldsymbol{x}_i(t)dt + \int_0^1 \boldsymbol{h}_i(t) \sum_{j=1}^n \boldsymbol{h}_j(t)dt \,,$$

which is the continuous-time analogue of the cost structure based on decomposition into permanent-impact cost and temporary-impact cost mentioned above.[9]

Now, suppose we are given a problem instance of this continuous time model, with $\kappa$ given, and time normalized into range $[0,1]$. We can approximate this arbitrarily well with a discrete time model as $T \to \infty$, by letting the discrete time grid correspond to $\{\frac{1}{T}, \frac{2}{T}, \ldots 1\}$ in continuous time. Specifically, we can do this as follows: suppose we are given collection of continuous-time strategies $\boldsymbol{h}_1^c, \ldots, \boldsymbol{h}_n^c$, and define

$$\boldsymbol{x}_i^c(t) = \int_0^t \boldsymbol{h}_i^c(l)dl \qquad \text{(for continuous } t \in [0,1])$$

$$x_{i,t} = \boldsymbol{x}_i^c\left(\frac{t}{T}\right) \qquad \text{(for discrete } t \in [T])$$

$$h_{i,t} = x_{i,t} - x_{i,t-1} \qquad \text{(for discrete } t \in [T]) \,.$$

Then, our discrete-time cost structure in terms of these strategy vectors $\boldsymbol{h}_i$ will be given by

$$c_i^{\alpha,\beta,T}(\boldsymbol{h}_i; \boldsymbol{h}_{-i}) = \alpha \sum_{t=1}^T h_{i,t} \sum_{j=1}^n x_{j,t-1} + (\alpha+\beta) \sum_{t=1}^T h_{i,t} \sum_{j=1}^n h_{j,t}$$

$$= \alpha \sum_{t=1}^T \left\{ \boldsymbol{x}_i^c\left(\frac{t}{T}\right) - \boldsymbol{x}_i^c\left(\frac{t-1}{T}\right) \right\} \sum_{j=1}^n \boldsymbol{x}_j^c\left(\frac{t-1}{T}\right)$$

$$+ (\alpha+\beta) \sum_{t=1}^T \left\{ \boldsymbol{x}_i^c\left(\frac{t}{T}\right) - \boldsymbol{x}_i^c\left(\frac{t-1}{T}\right) \right\} \sum_{j=1}^n \left\{ \boldsymbol{x}_j^c\left(\frac{t}{T}\right) - \boldsymbol{x}_j^c\left(\frac{t-1}{T}\right) \right\}$$

$$= \alpha \sum_{t=1}^T \frac{1}{T} \boldsymbol{h}_i^c\left(\frac{t-\gamma_{i,t}}{T}\right) \sum_{j=1}^n \boldsymbol{x}_j^c\left(\frac{t-1}{T}\right) + (\alpha+\beta) \sum_{t=1}^T \frac{1}{T} \boldsymbol{h}_i^c\left(\frac{t-\gamma_{i,t}}{T}\right) \sum_{j=1}^n \frac{1}{T} \boldsymbol{h}_j^c\left(\frac{t-\gamma_{j,t}}{T}\right) \,,$$

where the final line follows from the mean-value theorem, where $\gamma_{i,t} \in (0,1)$ for all $i,t$. Therefore, if we consider a sequence of discrete problem instances with $\alpha = \kappa$ and $\beta = T$, we get: $\lim_{T\to\infty} c_i^{\kappa,T,T}(\boldsymbol{h}_i; \boldsymbol{h}_{-i})$

$$= \lim_{T\to\infty} \kappa \frac{1}{T} \sum_{t=1}^T \boldsymbol{h}_i^c\left(\frac{t-\gamma_{i,t}}{T}\right) \sum_{j=1}^n \boldsymbol{x}_j^c\left(\frac{t-1}{T}\right) + \lim_{T\to\infty} \left(\frac{\kappa+T}{T}\right) \frac{1}{T} \sum_{t=1}^T \boldsymbol{h}_i^c\left(\frac{t-\gamma_{i,t}}{T}\right) \sum_{j=1}^n \boldsymbol{h}_j^c\left(\frac{t-\gamma_{j,t}}{T}\right)$$

$$= \kappa \int_0^1 \boldsymbol{h}_i^c(t) \sum_{j=1}^n \boldsymbol{x}_i^c(t)dt + \int_0^1 \boldsymbol{h}_i^c(t) \sum_{j=1}^n \boldsymbol{h}_j^c(t)dt \,,$$

where first equality plugs in the above result with $\alpha = \kappa$ and $\beta = T$, and the second follows from product of limits and the definition of the Riemann integral. Therefore, under appropriate re-normalization of the ratio $\beta/\alpha$ as we make the discrete-time approximation more fine-grained, our model can approximate the existing continuous time cost structure considered in Chriss (2024b;c;a; 2025) arbitrarily well if we let $T \to \infty$. Therefore, our model on the above restriction on idiosyncratic utilities, constraints, and exogenous actions subsumes theirs up to a vanishing discretization error.

---

[9]Historically this continuous time model predates the discrete time version as it originated in the finance/economics literature. We present in opposite order since our model, being computational, is discrete-time.

# D Experiment Details

## D.1 Estimating Market Parameters:

We extract market data for the Canadian Dollar (CAD) to U.S. Dollar (USD) forex exchange from Dukascopy, a Swiss financial services firm, which provides tick-level data on the following columns: bid, ask, bid volume, and ask volume. We use the approach outlined in Section 5 to estimate the permanent and temporary impact coefficients $\alpha$ and $\beta$ from such basic data. To control for periodic effects, we built a training data set with tick data from 10am-11am, 11am-12pm, and 12pm-1pm (Eastern time) for the 9 Tuesdays between September 2, 2025 and November 4, 2025. Each of these hour-long intervals consist of roughly 6000 ticks. For the test set, we consider the same three hour time window on Tuesday, Nov 11.

We present the results estimating $\alpha, \beta$ in Figures 3 and 4. Note that we *expect* the data here to be extremely noisy. A very strong signal/correlation of these market coefficients would allow strong predictions on price and would thus be quickly discovered by market participants (*i.e.* would violate principle of no arbitrage).

## D.2 Details on the Simulation Setup:

Let $\alpha^* = 4.65e^{-7}$ and $\beta^* = 3.25e^{-6}$ be the parameters estimated from our regression analysis on real data. To set the type dependent parameters means $\mathbb{E}[\alpha|\theta_1, \theta_2]$ we choose $\mu_{\alpha,\theta_1,\theta_2} = \mathbb{E}[\alpha|\theta_1, \theta_2] = \alpha^* + \mathrm{Unif}(-1e^{-8}, 1e^{-8})$ and $\mu_{\beta,\theta_1,\theta_2} = \mathbb{E}[\beta|\theta_i] = \beta^* + \mathrm{Unif}(-1e^{-7}, 1e^{-7})$. Then conditioned on their realized type, $P(\alpha|\theta_1, \theta_2) \sim \mathcal{N}(\mu_{\alpha,\theta_1,\theta_2}, 1e^{-8})$ and $P(\beta|\theta_1, \theta_2) \sim \mathcal{N}(\mu_{\beta,\theta_i}, 1e^{-7})$. Conditioned on $\theta_1, \theta_2$, $\alpha$ and $\beta$ are independent in our instance generation process.

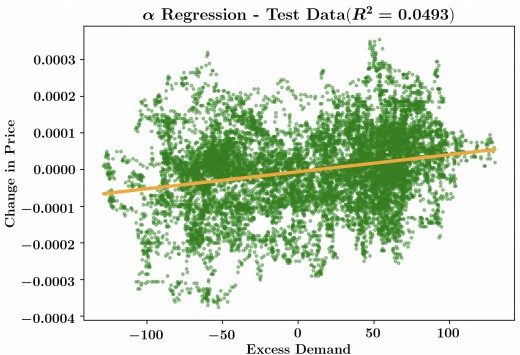
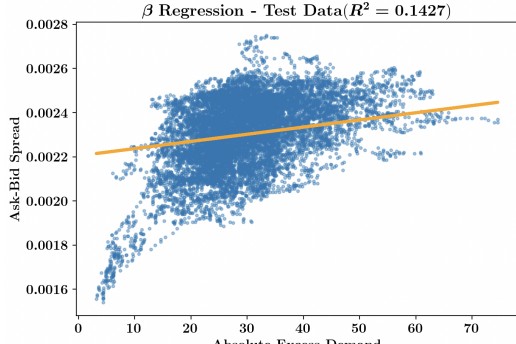

Figure 3: Regression performance on test data with $\alpha = 4.65e^{-7}$

Figure 4: Regression performance on test data with $\beta = 3.25e^{-6}$

## D.3 Sensitivity Analysis

To evaluate the robustness of the analytical Bayesian Nash Equilibrium and the last iterate result produced by Algorithm 1, we conduct a sensitivity analysis. First, we compute the confidence interval of our predicted $\alpha^*$ and $\beta^*$ parameters by using the standard approach for ordinary least squares regression. In figures 5 and 6 below, we fix one of these price parameters and vary the other over it's 95% confidence interval. The trajectory at the estimated $\alpha^*$ and $\beta^*$ are considered the baseline. While the simplest approach would be to plot, at each time $t$, the deviation from the baseline as a % of the baseline position at $t$, this suffers from a core issue: since all positions start at 0 at $t = 0$, we can see a blow up and numerical instability at early times. As such, we plot the deviation from the baseline as a % of the baseline position at the final time $T$. Further, since all agents types have some constraint or goal on their final position, this a meaningful anchor. We observe that in both the last iterate and the Bayesian Nash Equilibrium, the trajectories remain fairly consistent and stable across these perturbations, differing by no more than 1%.

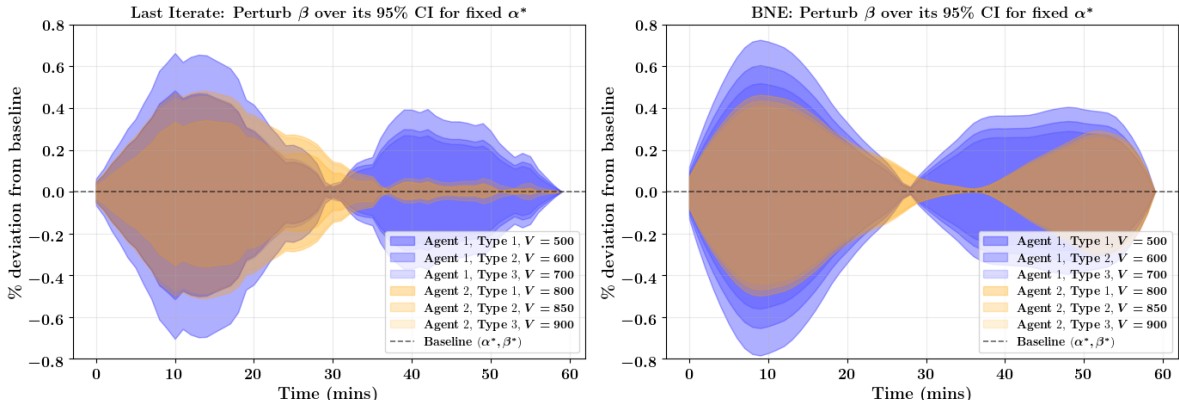

Figure 5: Varying $\beta$ over its 95% confidence interval $[3.10e^{-7}, 3.35e^{-7}]$ for fixed $\alpha^*$. Plotted is the deviation from baseline position as a % of the final baseline position. On the left is the effect on the last iterate from Algorithm 1. On the right is the effect on the empirical BNE.

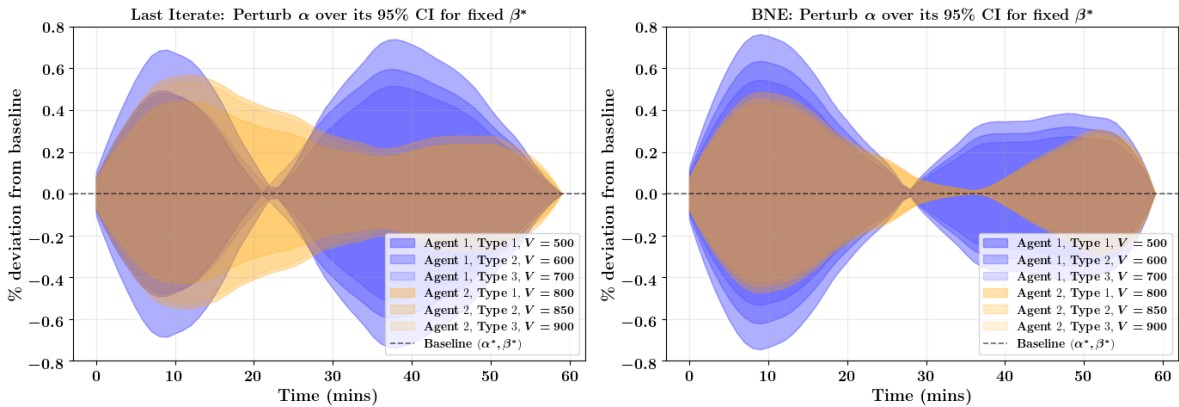

Figure 6: Varying $\alpha$ over its 95% confidence interval $[4.40e^{-7}, 4.9e^{-7}]$ for fixed $\beta^*$. Plotted is the deviation from baseline position as a % of the final baseline position. On the left is the effect on the last iterate from Algorithm 1. On the right is the effect on the empirical BNE.

# E   Comparisons and Connections to VWAP

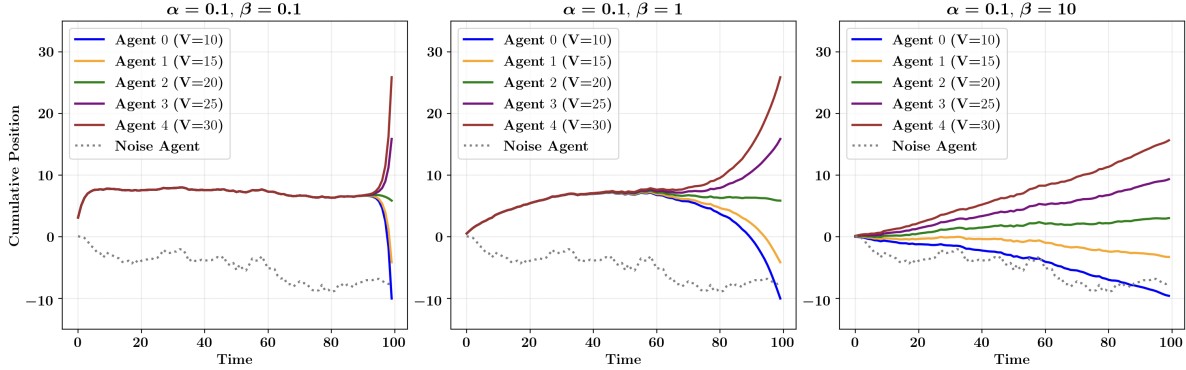

Figure 7: Cumulative position over time for 5 agents when all are being strategic and playing Nash Equilibrium strategies.

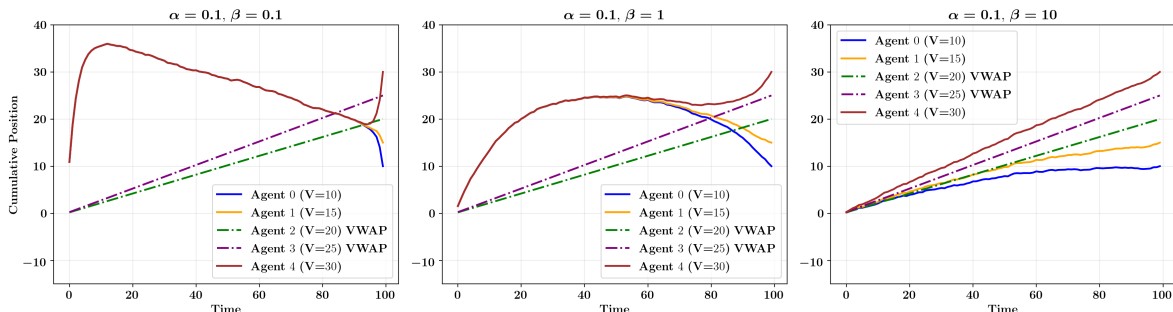

Figure 8: The strategies of the 5 players when agents 2 and 3 are play VWAP and the rest play the induced equilibrium of this setting.

Our model considers the dynamics of $n$ traders looking to execute a position within some markets governed by parameters $\alpha, \beta$, signifying the permanent and temporary impact of trading volume on price. We take a game-theoretic approach, where each trader's strategy is dictated by an equilibrium between all players. It is, however, instructive to compare such an approach with the widely used execution strategy known as *VWAP - Volume Weighted Average Price.*

The VWAP strategy is simple and doesn't require strategic consideration: at any given time interval, each agent trades proportional to the historical volume of trade that occurred at that interval. Larger trades are placed when there is expected to be large volume in the market, and smaller-sized trades are placed when the market volume is low. While our model does not explicitly model the historical market volume (except in empirical evaluations where $s$ is the aggregate market demand), this can be easily remedied by reinterpreting our model's time dimension. That is, instead of treating each time step $[t, t+1]$ as a fixed period of wall-clock time, we instead interpret it as *volume weighted time.* That is, based on past historical market data, we dilate each interval (shrink or expand) such that an equal amount of volume is traded within each interval. This may mean, for instance, that one interval corresponds to wall-clock time [9:30, 9:35] (high volume at the start of the day) and another to wall-clock time [11:00, 12:00] (lower volume at mid-day). Indeed, the price impact model (Almgren & Chriss, 2000) that our work is based on implicitly already assumes that time is defined as "volume-weighted time" exactly like this (see *e.g.* Almgren et al. (2005) for a detailed discussion of this issue.) Changing time measurement does not change any of our results.

In volume-weighted time, the VWAP strategy is simple: trade a constant amount at each interval. So, a trader building a position $V_i$ simply executes $V_i/T$ at each interval. A core question thus emerges: *How does the VWAP strategy compare to the equilibrium strategy and what are the dynamics of a market that includes both equilibrium traders and VWAP traders?*

We explore this question in the context of a simple synthetic example in the complete information setting. Suppose we have 5 traders who want to build (hard constraint) positions of $[10, 15, 20, 25, 30]$. They each have a linear final position utility $f_i(\boldsymbol{h}_i) = r_i \sum_t h_{i,t}$ for some reserve price $r_i$ (we use reserve values of $[4, 5, 6, 7, 8]$ respectively). We set $p_0 = 2$, $\alpha = 0.1$, vary $\beta \in \{0.1, 1, 10\}$, and set the exogenous player $s$ to be randomly sampled from a gaussian distribution. Then, in Figure 8, we plot the joint strategies where agents 2 and 3 follow VWAP, and the remaining agents follow the corresponding complete-information 3 player Nash Equilibrium given agents 2 and 3 following VWAP.

Even though agents $0, 1, 4$ are strategic in both settings (Figures 7 and 8), the fact that agents 2 and 3 have now shifted to a VWAP strategy changes the equilibrium strategy of these 3. Note, however, that for the large $\beta = 10$ setting, the agent behaviors do not change much (both for those who deviate and those who don't). This is intuitive since when the temporary impact is large, agents generally want to spread out their trades regardless of other factors.

We next ask, how does this shift affect the cost (i.e. negative utility) incurred by all agents? In Figure 10, we see that the agents who switch from being strategic to playing VWAP pay a *higher* cost for doing so. However, as $\beta$ becomes larger, this becomes less consequential, as the ratio tends to 1. The effect on the remaining three players, however, is the opposite. These players end up paying a *lower* cost when agents (2,3) are following VWAP versus when they are strategic. This suggests that players who switch from being strategic to playing VWAP end up paying a higher cost at VWAP. Agents who are always strategic can exploit those who follow VWAP. As $\beta$ becomes large, however, these effects become small.

Our results suggest a transition to VWAP to strategic (or vice versa) can have both a positive or negative impact depending on the player. How does this affect *total welfare*, *i.e.* the (negative) sum of all costs incurred by agents? In Figure 9, we plot the cumulative cost as a function of $\beta$. We plot 6 curves, where the $k^{th}$ curve corresponds to a subset of $k$ agents playing the VWAP strategy (we in fact consider all combinations of $k$ agents playing VWAP and take the average). Interestingly, we observe that the cumulative cost is almost always lower when a subset of agents are playing VWAP as compared to the all-strategic cumulative cost. This suggests that VWAP strategies could have better social welfare properties than all-strategic. We believe that this is an interesting finding that should be considered by market designers and regulators.

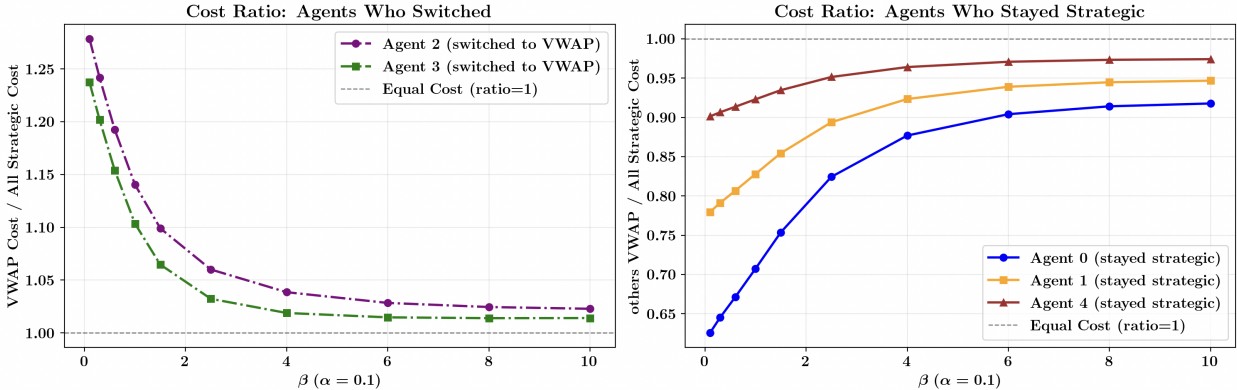

Figure 9: On the left is the ratio of cost between playing VWAP and playing strategically for agents 2 and 3. On the right is the ratio of costs for the remaining 3 agents (0,1,4) between when agents 2,3 were playing VWAP and when agents 2 and 3 were strategic.

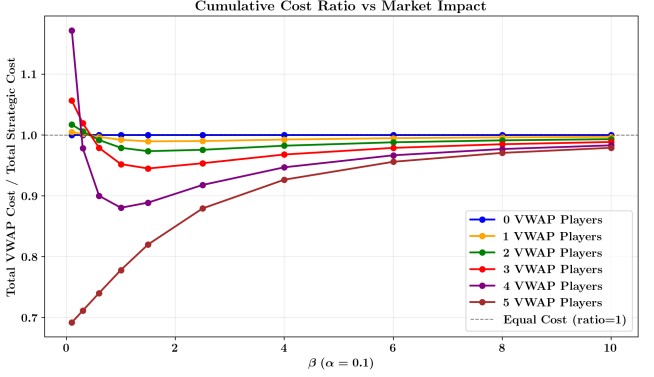

Figure 10: Ratio of cumulative costs between a subset of agents playing VWAP and all agents being strategic.

