# OpenReview forum: "Learning to Strategically Acquire Resources in Competition"
_TMLR — Accepted by TMLR_

### Review · Reviewer_xtGF · 2026-03-29

**Summary Of Contributions:**

The paper claims four main contributions.
1) it proposes a general Bayesian-game model of competitive resource acquisition with endogenous price impact, allowing convex action constraints, concave value functions, and incomplete information.
2) in the common-prior setting, it claims existence, uniqueness, deterministic structure, and efficient computability of the Bayesian Nash equilibrium, together with price-of-anarchy results.
3) in the no-common-prior setting, it claims a learning result under price-trajectory feedback and estimated market parameters, yielding last-iterate convergence to an approximate BNE.
4) it claims empirical support through simulations grounded in real FX data.

**Audience:**

Yes

**Audience Explanation:**

I think at least part of the TMLR audience would be interested in this paper.

It sits at the intersection of game theory and multi-agent learning.
It is framed not only around financial trading, but also around more general resource-allocation settings such as compute markets.

It might be a somewhat niche paper, but it is still clearly within the scope of TMLR.

**Broader Impact Concerns:**

I do not have major additional broader-impact concerns. The paper already includes a Broader Impact section, and I agree with its message.

**Claims And Evidence:**

Yes

**Claims Explanation:**

I think the main theoretical claims are basically supported: the model is fairly understandable and the equilibrium and learning results are provided with proofs. I did not notice an obvious fatal gap in the main arguments.

That said, I do think the paper oversells a few things a bit. For example, the price model is not really “canonical”, especially since the paper itself mentions other standard alternatives. Also, the experiments are not as strong as the abstract makes them sound: the market parameters are estimated from real FX data, but the agent types, preferences, and constraints are synthetic, and the online experiment assumes no additional observation bias in the market parameters, which avoids the hardest case stressed by the theory.

So overall I’d say *yes* for the core theory, but with some reservations about the way a few claims are presented.

**Requested Changes:**

My main request is that the paper should tone down some of the overselling and be more precise about what is actually established and what is only suggested. In particular, I would encourage the authors to use more cautious language about the empirical section and about how much it really validates the theory.
I would also suggest being more careful when describing the price model as canonical, and when making broad claims about the practical relevance of the framework: as the same authors later acknowledge, alternative standard price-impact models exist, and here agents commit ex ante to full trajectories without adapting within the $T$-step horizon.

---

> ### Author Response · Authors · 2026-05-06
> **Author Response**
>
> We thank the reviewer for the careful reading and for confirming the core theoretical claims. As in any theoretical work, we agree that our results hold under some simplifications. To that end, we followed the reviewer’s advice and made the following changes:
>
> - **On the price model:** We have replaced "canonical" with "standard", with Section 2 acknowledging the model is stylized relative to limit-order-book microstructure. Section 6 includes a more detailed discussion on this limitation and future related directions.
> - **On the empirical claims:** We have softened "empirically validating our theory" to "illustrating our theoretical results" and updated other phrasings in that spirit. We agree the experiments use synthetic agent types and constraints, and the language now reflects this. Similarly we move away from stressing the immediate practical relevance, noting that this is a primarily theory driven work.
> - **On the theoretical claims:** We more precisely outline the conditions and limitations of our theoretical results, including an explicit query complexity framing of solving for BNE, and a discussion on the delta-net based approach we use (Section 4) and its limitations (Section 6).

---

### Review · Reviewer_ztTF · 2026-04-03

**Summary Of Contributions:**

This paper studies a game in which multiple agents acquire divisible goods over multiple time steps under a certain price dynamics.
The authors consider a setting where each agent has a private type, and provide theorems on the properties of Bayesian Nash equilibria (BNE) as well as methods for their learning and computation.
Specifically, they establish the existence and uniqueness of BNE, show that the BNE can be computed in the complete information setting (where the prior is given), and provide learning dynamics for computing BNE through repeated play even when the prior is not known in advance.
They also provide an analysis of the Price of Anarchy.

Key strengths:
- The paper formulates a natural and general problem setting that encompasses existing models, and then broadens the discussion from basic properties of the game to advanced and wide-ranging topics such as the Price of Anarchy and doubly optimal learning dynamics. This suggests the paper would be of interest to a broad audience.
- Most of the main results are presented with readable and careful proofs, and I did not find any apparent mathematical errors.
- The effectiveness of the proposed framework is validated not only theoretically but also through experiments on real data.

Key weaknesses:
- Many of the technical components appear to be combinations of existing results, and it is somewhat unclear where the technical novelty lies. For instance, the learning dynamics and the analysis methods rely heavily on Jordan et al. (2024). While it is mentioned that the analysis techniques from prior work cannot be directly applied, a clear and concrete explanation of what new techniques were developed to address these difficulties would strengthen the paper.
- The claim that BNE is computable in the common prior setting was not entirely clear to me in terms of what it concretely means (see Requested Changes).

**Audience:**

Yes

**Audience Explanation:**

The paper formulates a natural and general problem setting that encompasses existing models, and broadens the discussion from basic properties of the game to advanced and wide-ranging topics such as the Price of Anarchy and doubly optimal learning dynamics. This suggests the paper would be of interest to a broad audience within TMLR.

**Claims And Evidence:**

Yes

**Claims Explanation:**

The theoretical claims are supported by detailed proofs, most of which are readable and carefully written. I did not find any apparent mathematical errors. The effectiveness of the proposed framework is further validated through experiments on real data.

**Requested Changes:**

- In Section 2, the paper claims to "generalize and improve on past settings." I would appreciate a more explicit explanation of which specific prior works are generalized and in what respects. While this may be inferable from a careful reading of the text, I was unable to form a clear picture of how the various prior works relate to the present work just from reading the paper. A summary table, for example, might be helpful.

- Regarding the claim on page 5 that the BNE is "computable," I would like the authors to explicitly clarify what computational model is assumed and with respect to what parameter the running time is linear. Specifically, in order for a meaningful discussion of computational complexity, one needs to specify: (1) how the prior is accessed (e.g., whether one can draw an arbitrary number of i.i.d. samples from the prior, or whether an explicit representation of the prior is given), and (2) how the functions $f_i$ are accessed (e.g., as black-box functions, via a first-order oracle, or through some explicit representation). Furthermore, when claiming "linear time," it was unclear to me with respect to which parameter this linearity holds. The standard convention is linear time in the input length, but in settings like this one, where instances are defined by distributions and functions, what constitutes the input length is non-trivial and case-dependent, and therefore needs to be specified.

Minor comments:
- Definition 3:
  - 1. Is the intended quantification "for all $\theta_i$ and for all $h_i \in \mathcal{H}_i$"?
  - 2. I understand $\mathcal{H}_i$ to be a set of mappings $\Theta_i \to \mathbb{R}^T$, but I was unclear on what it means for this set to be "closed with respect to the $L_2$ norm." In particular, when $\Theta_i$ is an infinite set, how is the $L_2$ norm defined?

---

> ### Author Response · Authors · 2026-05-06
> **Author Response**
>
> Thank you for the helpful comments and suggestions which will improve the paper. We have updated Section 3 to elaborate more precisely on the complexity of computing equilibrium, Section 4 to clarify/highlight our technical contributions, and Section 2 to give a high-level summary on how work builds on and generalizes past works (Table 1). In detail and specific to the points you raised:
>
> ### **Clarifying technical novelty of Section 4:**
> While our theory builds heavily on Jordan et al., the new technical work here is to adapt and extend their AdaOGD framework to our setting of Bayesian resource acquisition/position building games with realistic market feedback. This adaptation requires extending both the algorithm and the analysis in two key ways. First, whereas Jordan et al. study finite-dimensional action spaces, our agents choose type-contingent strategy maps, potentially over continuous type spaces. We therefore adapt the algorithm to optimize over a discretized Bayesian game, prove that the relevant convexity and monotonicity properties are preserved, and show how convergence to BNE in the discretized game can be lifted back to approximate BNE convergence in the original continuous-type game. Second, whereas Jordan et al. assume unbiased stochastic gradient observations, our agents compute gradients from prices and estimated market-impact parameters, which introduces bias. We extend the last-iterate convergence argument to handle these biased gradient estimates, yielding the additional estimation-error term in our convergence guarantee. We updated the writing in Section 4 to explicitly explain these two extensions and the technical obstacles they address.
>
> ### **Computability of BNE:**
> This thoughtful comment prompted us to revise the relevant statements in the paper and be more precise. We have updated Corollary 1 and its proof in Appendix A, and added a follow-up paragraph providing a more granular bound for discrete type spaces. To summarize, complexity results available in this setting (computation of equilibria/optimization) are most naturally stated as a *query complexity* result rather than a bit-level time complexity result in the classical TCS sense. The reason is that the operator $F$ defining the variational inequality is given implicitly through (i) the prior over types and market parameters, and (ii) the value functions $f_i$. A clean time-complexity statement requires committing to an explicit representation of both, which would substantially restrict the generality of the model. The query-complexity framing, counting calls to an oracle for $F$ (or for more granularity, to $f_i$), is the standard convention in the variational inequality and learning-in-games literature precisely for this reason, and is what we have adopted in the revision. With this in mind, our revisions are as follows:
>
> - We have rewritten the corollary to state the precise query-complexity claim: assuming oracle access to the variational inequality operator $F$, the extragradient algorithm computes an $\varepsilon$-approximate BNE in $O((L/c) \log(1/\varepsilon))$ oracle calls, where $c$ is the strong-monotonicity constant and $L$ is the Lipschitz constant of $F$ established in Theorem 1. While the existence of this oracle may be unrealistic for continuous-valued type spaces, access to exact priors for such type spaces is already unrealistic, so this is a moot point (this is also why we study a more realistic model in Section 4 without access to priors). Instead, this computational result is useful as it ensures efficient computation of BNE in approximate or counterfactual models of reality for the purpose of analysis.
>
> - In the paragraph following the corollary proof,, we provide a more granular bound for the case of discrete types. Letting $M = \max_i |\Theta_i|$ (the size of the largest type space), an $\varepsilon$-approximate BNE can be computed in $O((nMT)^2 (L/c) \log(1/\varepsilon))$ arithmetic operations and queries to the gradient of the value functions $f$.
>
> ### **Regarding Definition 3**
> - Yes, the first quantification is for all $\theta_i \in \Theta_i$ and all $h_i \in \mathcal{H}_i$. We have clarified this in the updated manuscript.
> - The strategy space $\mathcal{H}_i$ is a closed subset of the Hilbert space of square-integrable mappings with the inner product defined as: $\mathbb{E} \langle h_i(\theta_i), h'_i(\theta_i) \rangle$. We also clarified this in the manuscript.

---

### Review · Reviewer_hDUm · 2026-04-22

**Summary Of Contributions:**

This paper proposes a highly realistic dynamic market model for competitive resource acquisition, which captures many realistic factors occurring in the real market. In this paper, the authors start from complete information to incomplete information. They give formal definitions of agents type, market type, assumptions and interaction process mathematically, which speak clearly of such games. Besides, in section 3, they analyze existence and uniqueness of BNE with common prior. They provide positive(Upper bound PoA when all agents only buy and build a position) and negative results(cannot upper bound PoA without specific limiations) respectively. Considering the existence and uniqueness of BNE, they design the online-learning algorithm to help agents learn environment’s parameters without common prior. They prove the last iterative-strategies form an \varepsilon-BNE and upper bound the \varepsilon. In the analysis of the algorithm, they gives two solutions to solve existing limitations. I think their methods are inspiring and natural. Finally, they give the numerical experiments based on read data and results work.

**Audience:**

Yes

**Audience Explanation:**

The findings of this paper will be of interest to a substantial portion of TMLR's audience, particularly researchers working at the intersection of multi-agent systems, online learning, and game theory.

**Broader Impact Concerns:**

The authors have included a dedicated "Broader Impact" statement in Section 6.2. They correctly identify that their primary contribution is theoretical and methodological, and that any potential risks are indirect and tied strictly to how downstream practitioners might deploy these algorithms in real-world financial or cloud compute market

**Claims And Evidence:**

Yes

**Claims Explanation:**

Theoretical Evidence: The mathematical proofs supporting the existence, uniqueness, and computability of the Bayesian Nash Equilibrium (BNE) in Section 3 are roughly clearly structured. The authors expertly handle the transition to the no-prior setting in Section 4 by extending doubly-optimal learning dynamics. Their approach to overcoming the challenges of continuous type spaces (via $\delta$-net discretization) and biased gradient feedback is mathematically sound. The underlying assumptions(such as (B,L)-regularity for strategy spaces and the standard linear price impact model) are reasonable within the context of optimal execution literature.

Empirical Evidence: The authors validate their claims using both synthetic data and real-world CAD/USD tick-level forex data from Dukascopy.

**Requested Changes:**

Critical changes:
1. In Section 4, Algorithm 1 uses a $\delta$-net to discretize the continuous type space $\Theta_i$​. Because the size of this discretized space scales exponentially with the type dimension $m_i​$, the approach faces a severe computational bottleneck for high-dimensional inputs. Please explicitly acknowledge this scalability limitation in the Discussion (Section 6), clarifying that the method is primarily practical for low-dimensional type spaces. A brief mention of future directions, such as using function approximation instead of strict discretization, would strengthen the paper.
1. The linear regressions used to estimate $\alpha$ and $\beta$ (Appendix D) have very low $R^2$ values (0.0493 and 0.1427). While noisy tick-level data is expected, running simulations based solely on these specific point estimates makes the empirical validation slightly less convincing. Please consider to add a brief sensitivity analysis: perturb α and β within their confidence intervals to demonstrate that Algorithm 1's convergence and the qualitative behavior of the BNE remain structurally robust to this estimation noise.

Minor/Recommended changes:
1. Precision in Appendix B.3: Some of the text explaining the bounds between the biased gradient estimate and the true population gradient in the proof of Theorem 4 could be tighter. Please review the phrasing in these expectation bounds to ensure a crystal-clear distinction between empirical estimates and true distributional parameters, which will make the long derivations easier to follow.
1. Early Disclaimer on Price Dynamics: You rightly acknowledge in Section 6 that the linear price impact model abstracts away complex limit order book (LOB) microstructures. However, to prevent readers with strong finance backgrounds from getting hung up on this early on, I recommend adding a quick one-sentence disclaimer in Section 2 right when Assumption 1 is introduced to set proper expectations.

---

> ### Author Response · Authors · 2026-05-06
> **Author Response**
>
> Thank you for the helpful comments; we appreciate your suggestions which will improve the paper. Regarding your comments:
>
> - **Critical Change 1:** We agree that the discretization approach introduces an important scalability limitation, since the size of the epsilon-net grows exponentially with the dimension of the type space. We expanded the discussion to Section 6.1 to explicitly acknowledge the issue this can cause in high-dimensional spaces, as well as discuss the function approximation approach you pointed out. However, as we note there, in practice this exponential dependence on dimension may be overly pessimistic, as much past work in ML shows that learning complexity really depends on intrinsic dimension, which may be much lower than the actual type space dimension. In other words, the number of discrete points needed to cover the type space well for our algorithm may be much lower than a naive analysis would suggest.
> - **Critical Change 2:** In the revised manuscript, we implemented your helpful suggestion. The appendix now includes a sensitivity evaluation of the Bayesian Nash Equilibrium and the last iterate result produced by Algorithm 1 (Appendix D.3). Specifically, we vary $\alpha$ within its 95% confidence interval (for the estimated $\beta$) and plot the deviation in trajectory (for both BNE and last iterate) as compared to the estimated value. We do a similar analysis varying $\beta$ for the estimated $\alpha$. We observe that neither trajectory deviated by more than 1% of the baseline position at the final time $T$.
> - **Minor Change 1:** Thank you for pointing this out. We have updated the proof of Theorem 4 with some clarifying text. These updates are intended to make the derivation easier to follow and to clarify precisely how price observations and estimated market parameters are used to compute a biased gradient, as well as how the biased gradient estimate differs from the true population gradient.
> - **Minor Change 2:** This is a great suggestion and we have included a small disclaimer in Section 2 about our model vis-a-vis more detailed limit order book models.

---

### Author Response · Authors · 2026-05-06
**General Author Response**

We sincerely thank all reviewers for their time, effort, and helpful comments on our paper. We have incorporated the suggestions and changes mentioned by the reviewers in our updated manuscript, with all changes displayed in blue font. Please see the individualized responses for more details.

---

### Decision · Action_Editor_fiNv · 2026-06-01

**Recommendation:** Accept as is

**Audience:**

Yes

**Audience Explanation:**

The paper will be of interest to the portion of TMLR's audience working at the intersection of multi-agent systems, online learning, mechanism design and game theory.

**Claims And Evidence:**

Yes

**Claims Explanation:**

All reviewers agreed that the claims made in the submission are supported by careful and rigorous (if not always fully novel) mathematical proofs.

The reviews prompted a revision that includes more precise and transparent statements about the scalability and numerical complexity of the proposed approach, as requested. More broadly, the authors have clarified the merits and limitations of their work.